# Multimodal collective swimming of magnetically articulated modular nanocomposite robots

Sukyoung Won [1,2,10], Hee Eun Lee[3,10], Young Shik Cho[4], Kijun Yang [1,2], Jeong Eun Park[1,2], Seung Jae Yang [5] ✉ & Jeong Jae Wie[6,7,8,9] ✉

Magnetically responsive composites can impart maneuverability to miniaturized robots. However, collective actuation of these composite robots has rarely been achieved, although conducting cooperative tasks is a promising strategy for accomplishing difficult missions with a single robot. Here, we report multimodal collective swimming of ternary-nanocomposite-based magnetic robots capable of on-demand switching between rectilinear translational swimming and rotational swimming. The nanocomposite robots comprise a stiff yet lightweight carbon nanotube yarn (CNTY) framework surrounded by a magnetic polymer composite, which mimics the hierarchical architecture of musculoskeletal systems, yielding magnetically articulated multiple robots with an agile above-water swimmability (~180 body lengths per second) and modularity. The multiple robots with multimodal swimming facilitate the generation and regulation of vortices, enabling novel vortex-induced transportation of thousands of floating microparticles and heavy semi-submerged cargos. The controllable collective actuation of these biomimetic nanocomposite robots can lead to versatile robotic functions, including microplastic removal, microfluidic vortex control, and transportation of pharmaceuticals.

Collective behavior of active matter is widely prevalent and ranges from cell migration[1] to insect swarms[2] and bird flocks[3]. Numerous living creatures organize to complete complicated tasks, navigate through harsh environments, and defend against predators. Recently, nature-inspired robotic collectives, *e.g.* kilobot swarms[4,5] and drone flocks[6], have been devised. Multiple mechanical robots equipped with on-board batteries can be organized through sensor-driven communication. Meanwhile, in miniaturized robotics, external-stimuli-responsive particle swarms that can be operated by a magnetic field[7], light[8], electric field[9], and ultrasound[10] have been introduced to avoid on-board batteries. In particular, magnetic actuation has attracted considerable attention for developing miniaturized robotic systems owing to the real-time spatiotemporal controllability and high penetrability of external magnetic fields[11,12]. Time-varying magnetic fields[13–15] including rotating, oscillating, and precessing offer programmable magnetomotility of microrobots through magnetic torque

[1]The Research Institute of Industrial Science, Hanyang University, Seoul 04763, Republic of Korea. [2]Program in Environmental and Polymer Engineering, Inha University, Incheon 22212, Republic of Korea. [3]Green Product Solution Center, SK Innovation, Daejeon 34124, Republic of Korea. [4]Department of Materials Science & Engineering and Research Institute of Advanced Materials (RIAM), Seoul National University, Seoul 08826, Republic of Korea. [5]Advanced Nano-hybrids Laboratory, Department of Chemistry and Chemical Engineering, Education and Research Center for Smart Energy and Materials, Inha University, Incheon 22212, Republic of Korea. [6]Department of Organic and Nano Engineering, Hanyang University, Seoul 04763, Republic of Korea. [7]Human-Tech Convergence Program, Hanyang University, Seoul 04763, Republic of Korea. [8]Department of Chemical Engineering, Hanyang University, Seoul 04763, Republic of Korea. [9]Institute of Nano Science and Technology, Hanyang University, Seoul 04763, Republic of Korea. [10]These authors contributed equally: Sukyoung Won, Hee Eun Lee. ✉e-mail: sjyang@inha.ac.kr; jjwie@hanyang.ac.kr

occurring in a magnetic body. Reconfigurable swarms of magnetic microparticles have been realized by varying the axes and frequencies of the alternating magnetic fields[16,17]. However, most studies on collective behavior have focused on colloidal particle systems[18]. Although particle swarms demonstrate micro-cargo transportation capabilities, they face challenges in the transportation of multitudinous cargos. When the amount of cargo increases, long-distance transport and location control become difficult to achieve in a limited time duration.

Geometric changes in miniaturized robots are anticipated to improve actuation efficiencies of the robots for cargo transportation. Polymer-composite-based robots are particularly promising since diverse structures, composed of multi-material, can be constructed through polymer processing techniques[19–23]. For example, polymeric robots in capsule[24] or cross[25] shape could load and release a miniscule cargo to arbitrary destinations. However, cargo transportation has primarily focused on a single cargo smaller than the size of polymeric robots. Recently, tens of robots have cooperated to carry an object that was larger and heavier than the individual robots[26], yet further challenges remain to deliver multiple cargos rapidly and simultaneously. Controllable collective actuation of multiple robots is required to achieve functionalities beyond those offered by single-robot systems.

In this study, we presented musculoskeletal-system-mimicking nanocomposite robots capable of agile multimodal above-water swimming. The nanocomposite robots were designed with the aim of generating a water vortex at the air-water interface and transporting multiple cargos via the vortex. Vorticity was controlled by multimodal collective swimming of multiple nanocomposite robots. Thousands of floating microbeads were collected or confined in space through vorticity control by on-demand multimodal collective swimming. Furthermore, the nanocomposite robots transported a heavy semi-submerged cargo that was intentionally prevented from buoying. The multimodal collective swimming of these biomimetic nanocomposite robots demonstrates their potential for overcoming the challenges of limited interactive motions of multiple polymeric robots.

## Results

### Biomimetic hierarchical structure of robots

For a swimming robot to be capable of vortex generation at the air-water interface as well as cargo transportation, a lightweight robot body was essential. Lightweightness of robots can impart agility and above-water swimmability to the robots. A stiff body frame was also required to overcome surface tension and drag force in swimming even though the two variables of lightweightness and rigidity are considered trade-offs according to Ashby plots[27] showing Young's modulus versus density. We further conceived a magnetic robot with high aspect ratio (AR) and collective actuation of the high-AR robots as a more effective strategy for generating the vortex. When the high-AR robots were magnetically assembled to have large hydrodynamic volume, the magnetically assembled robots were expected to exert a high rotational force and induce a large-magnitude vortex during rotational swimming. Magnetic assembly of the multiple robots required larger magnetic attractive force among robots than drag force to preserve the assembled structure even during rapid swimming. To enhance magnetic attractive forces among robots, we considered a core-shell-like robot by binding highly concentrated magnetic particles only on the robot surface, instead of embedding particles inside the robot since the interparticle magnetic force drastically increases by decreasing interparticle distances[28,29]. The robot design that met these requirements was inspired by brachial anatomy (Fig. 1a). Movements of the human arm are produced by forces generated from skeletal muscles surrounding the stiff long bones. Along the interior toward the end of the long bones, the spongy bone shows a hierarchical composition of collagen fiber bundles followed by a porous network while the bone marrow fills in the bone. The porosity of spongy bone offers lightweightness to the long bones. Surrounding

skeletal muscles are bounded by the connective tissue known as deep fascia. We introduced a biomimetic hierarchical nano/microstructure with nanoporous carbon nanotube yarn (CNTY) framework, polydimethylsiloxane (PDMS), and magnetic microparticles, emulating the spongy bone, connective tissue, and skeletal muscle, respectively (Fig. 1b and c). First, the spongy-bone-mimicking CNTY framework was prepared using multi-walled carbon nanotube (MWNT) fibers synthesized through floating catalyst chemical vapor deposition (Supplementary Fig. 1). The synthesized 36-μm-thick MWNT fiber was composed of ten-walled 14 nm-thick MWNTs (Fig. 1d). The MWNT fiber provided specific strength of 1.19 N tex$^{-1}$ (equivalent to 1.19 GPa g$^{-1}$ cm$^3$) and specific modulus of 46.9 N tex$^{-1}$ (Supplementary Fig. 2). For lightweight nanoporous structure, high densification was intentionally restrained in the MWNT fiber, confirmed by Raman spectra (Supplementary Fig. 3). To build the stiff yet lightweight nano-framework, 100 MWNT fibers were twisted to form the 340-μm-thick CNTY (Supplementary Fig. 4).

Then, a dip-coating process was implemented to bind highly concentrated magnetic particles on the robot surface. The nanoporous CNTY framework was immersed in a mixture of PDMS prepolymer and magnetic particles, followed by drainage of excessive mixture (Supplementary Fig. 5). During the dip-coating process, the nanoporous CNTY framework operated as a membrane to permeate the liquid-phased PDMS prepolymer through capillary forces[30], whereas the 5-μm-sized magnetic particles[31] were excluded from the core region, as shown in Fig. 1b. The magnetic particles at the CNTY surface were bound through thermal curing of PDMS via hydrosilylation polymerization. The PDMS acted as a binder for the magnetic particles in a fashion similar to the deep fascia and mimicked the binding of muscle fibers. Absence of the PDMS binder could hinder the adhesion of magnetic particles onto the CNTY framework due to the weak cohesive energy between the magnetic particles. By spatially selective coating of the magnetic particles onto the surface of the nanoporous CNTY framework, the coating thickness of the magnetic composite layer could be varied with the concentration of magnetic particles dispersed in the PDMS prepolymer (Supplementary Fig. 6–8).

A uniform coating of the magnetic composite layer was achieved at particle concentrations of 10, 20, and 30 vol.%. To comprehensively probe the effects of particle concentration on the hierarchical nano/microstructures, the rheological behavior of the prepolymer mixtures was investigated by small-amplitude oscillatory shear (Supplementary Fig. 9, 10, Supplementary Note 1). The complex viscosity of the PDMS prepolymer was measured to be 18 Pa·s at an angular frequency of 3 rad s$^{-1}$. The rheological analysis was essential to avoid a scenario such as sagging of a low-viscosity mixture under gravity, which prevents the formation of magnetic composite layers during the dip-coating process. At particle concentrations of 10, 20, and 30 vol.%, the complex viscosities reached 90, 371, and 897 Pa·s, yielding 14-, 21-, and 60-μm-thick magnetic composite layers, respectively, after thermal curing. The saturation magnetization ($M_s$) values of the ternary nanocomposites were measured to be 69, 118, and 152 emu g$^{-1}$, respectively, in accordance with the increased loading of the magnetic particles (Supplementary Fig. 11). The drastic increase in complex viscosity was observed between particle concentrations of 20 vol.% and 30 vol.%, indicating the percolation of randomly distributed spherical particles. At magnetic particle loadings >30 vol.%, the ensuing high complex viscosities interrupted the removal of excessive prepolymer mixtures via gravity-driven drainage, resulting in a non-uniform thick coating of the magnetic composite layers. Therefore, CNTY robots were prepared using particle concentrations of 10, 20, and 30 vol.% with respect to the PDMS prepolymer. The geometry of the CNTY robots was regulated to yield different ARs such as 2.5, 6, and 13. The high-AR CNTY robots were facilely obtained owing to the dip-coating process of CNTYs. Each CNTY robot is denoted by its AR and the particle concentration (for example, AR-2.5/30-vol.%). The dimensions,

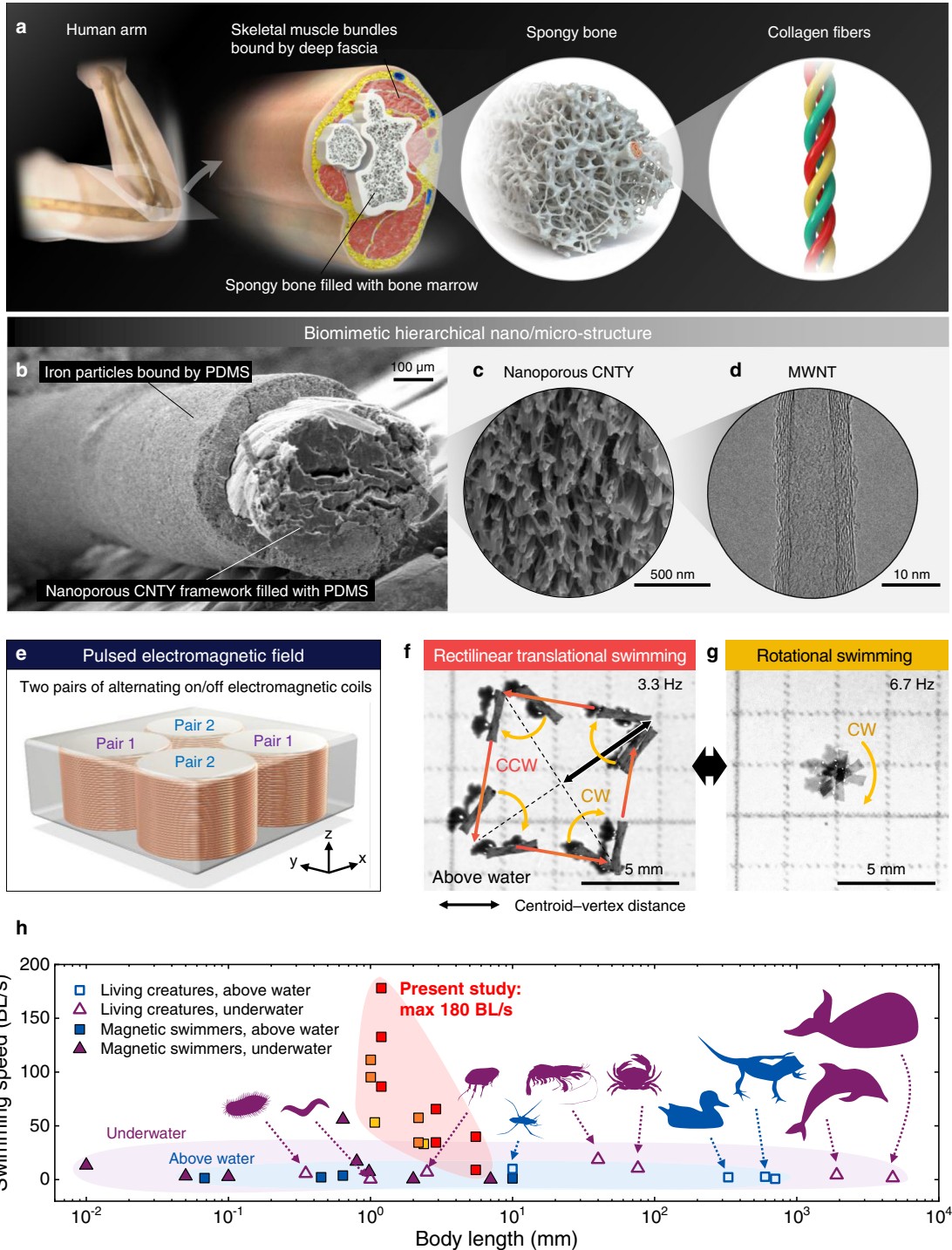

**Fig. 1 | Biomimetic nanocomposite robots capable of above-water swimming.**
**a** Schematic illustration of human musculoskeletal system. **b** Biomimetic hierarchical nano/microstructure. **c** Nanoporous CNTY framework prepared by twisting 100 MWNT fibers. **d** A single MWNT. The cross-sectional views shown in (**b**) and (**c**) were obtained by scanning electron microscopy (SEM), and the top-down view shown in (**d**) was obtained by transmission electron microscopy (TEM). **e** Schematic illustration showing the induction of magnetic rotation by the pulsed electromagnetic field. **f, g** Overlaid images representing bimodal above-water swimming of CNTY robots in (**f**) rectilinear translational swimming and (**g**) rotational swimming modes. **h** Swimming speeds of CNTY robots, living swimmers, and artificial swimmers. The yellow, orange, and red squares represent 10, 20, and 30 vol.% CNTY robots, respectively.

body masses, and $M_s$ values of the CNTY robots are listed in Supplementary Table 1.

## Agile multimodal swimming

The designed CNTY robot swam above-water under a pulsed quadrupolar electromagnetic field (<13 mT) (Fig. 1e, Supplementary

Movie 1). Two electromagnetic coils were embedded pairwise inside the device, and alternative switching of the coil pairs in the on/off states induced a clockwise (CW) rotation of the pulsed electromagnetic field. For example, north (N) and south (S) magnetic poles were sequentially induced in the (i) first and third, (ii) fourth and second, (iii) third and first, and (iv) second and fourth quadrant positions.

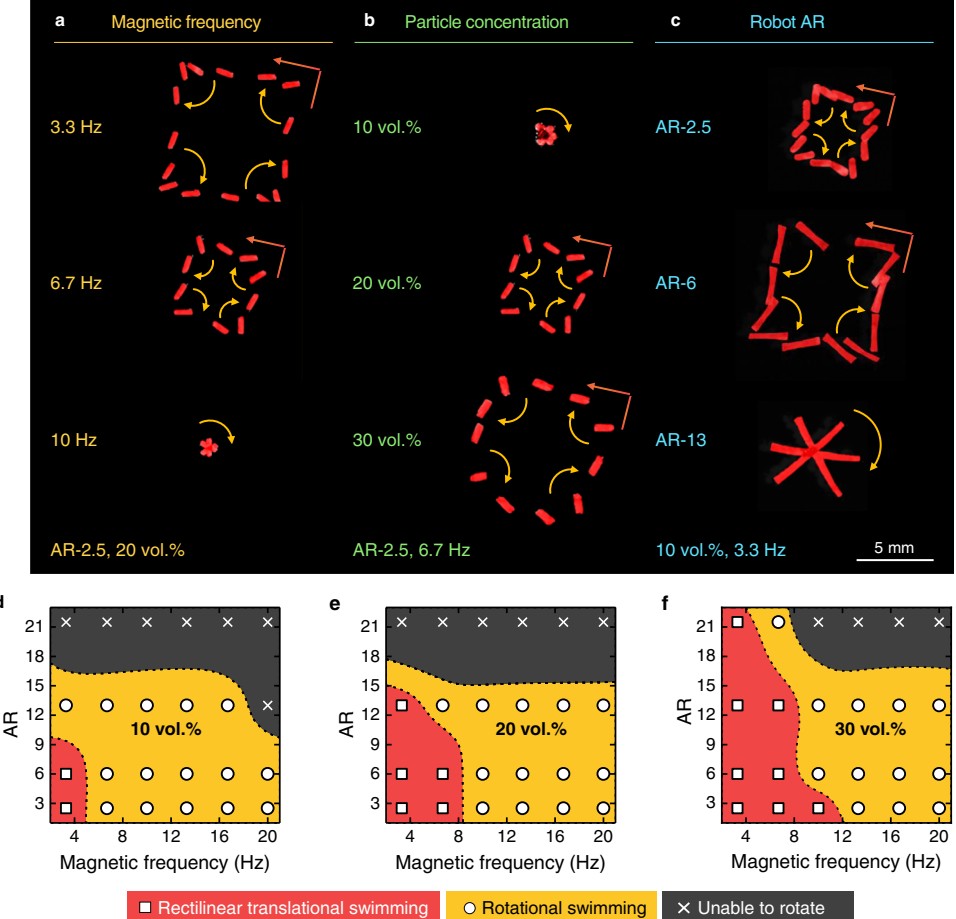

**Fig. 2 | Manipulation of a single CNTY robot. a–c** Overlaid images of a single CNTY robot swimming under varying (**a**) magnetic frequency, (**b**) particle concentration, and (**c**) robot AR conditions. The robots are shown in red to distinguish their shadows reflected on water. **d–f** Panoptic swimming modes of the CNTY robot at particle concentrations of (**d**) 10, (**e**) 20, and (**f**) 30 vol.% in PDMS prepolymer.

During the transition state in these four steps, the CNTY robot performed counterclockwise (CCW) rectilinear translational swimming (Supplementary Fig. 12) via the magnitude gradients of the magnetic field[32]. When the subsequent coil pair switched on, a CW rotation of the CNTY robot was initiated at the vertex of the swimming trajectory owing to the magnetic torque applied to the robot. Therefore, a rectilinear pathway was observed, as shown in Fig. 1f. The centroid–vertex distance in the AR-6/10-vol.% robot was 4 mm at a magnetic frequency of 3.3 Hz. This magnetic frequency represents the rotation frequency of the pulsed electromagnetic field. The actuation mode of the AR-6/10-vol.% robot could be switched from CCW rectilinear translational swimming to CW rotational swimming by increasing the magnetic frequency from 3.3 to 6.7 Hz (Fig. 1g, Supplementary Fig. 13). The increased magnetic frequency prohibited rectilinear translational swimming at the adopted time scale and immediately diverted the trajectory of the CNTY robot. Consequently, the AR-6/10-vol.% robot rotated in the CW direction at the center of the substrate. The bimodal above-water swimming could be distinguished by the presence or absence of phase difference in the time-resolved harmonic waves in the *x*-axis and *y*-axis coordinates of the CNTY robots (Supplementary Note 2, Supplementary Fig. 14, 15).

During above-water swimming, the rotation frequency of the CNTY robot was synchronized to the magnetic frequency of the source (Supplementary Fig. 16), contributing to highly agile rectilinear translational swimming. The swimming speed of the CNTY robotic system was compared to those of previously reported living swimmers and miniaturized artificial magnetic swimmers (Fig. 1h, Supplementary

Table 2, 3). The swimming speed of the CNTY robots could be selected according to the magnetic frequency, particle concentration, and AR. Representatively, the AR-2.5/30-vol.% CNTY robot demonstrated an average swimming speed of up to 212 mm s$^{-1}$, which was two times that of the water striders. When the body length (BL) was used to normalize these data for enabling comparison with other swimmers, the CNTY robot was found to show an average swimming speed of 180 BL s$^{-1}$ (Supplementary Fig. 17). Moreover, the ternary-nanocomposite-based CNTY robots exhibited an above-water swimming speed 47 times faster than that of the binary-nanocomposite-based robot (spinbot) previously developed by our group[26], which was actuated by continuous rotation of two linearly connected permanent magnets. The agility of the CNTY robots can be attributed to not only the stiff yet lightweight biomimetic structures but also the synchronized rotation frequency of the CNTY robots with the magnetic frequency of the pulsed electromagnetic field. In addition to the AR-2.5/30-vol.% CNTY robot, AR-6/30-vol.% and AR-13/30-vol.% CNTY robots demonstrated agile above-water swimmability up to 66 and 40 BL s$^{-1}$, respectively.

Three variables affecting the swimming modes—external magnetic frequency, particle concentration, and robot AR—were scrutinized (Fig. 2, Supplementary Fig. 18, Supplementary Movie 2). Magnetic force-driven rectilinear translational swimming is correlated with the inertial force of the CNTY robot and the drag force applied to it. First, an increase in magnetic frequency led to fast swimming speed and consequently caused high shear stress of water (Fig. 2a). When the magnetic frequency was switched from 3.3 to 6.7 Hz, the centroid–vertex distance in the AR-2.5/20-vol.% robot decreased;

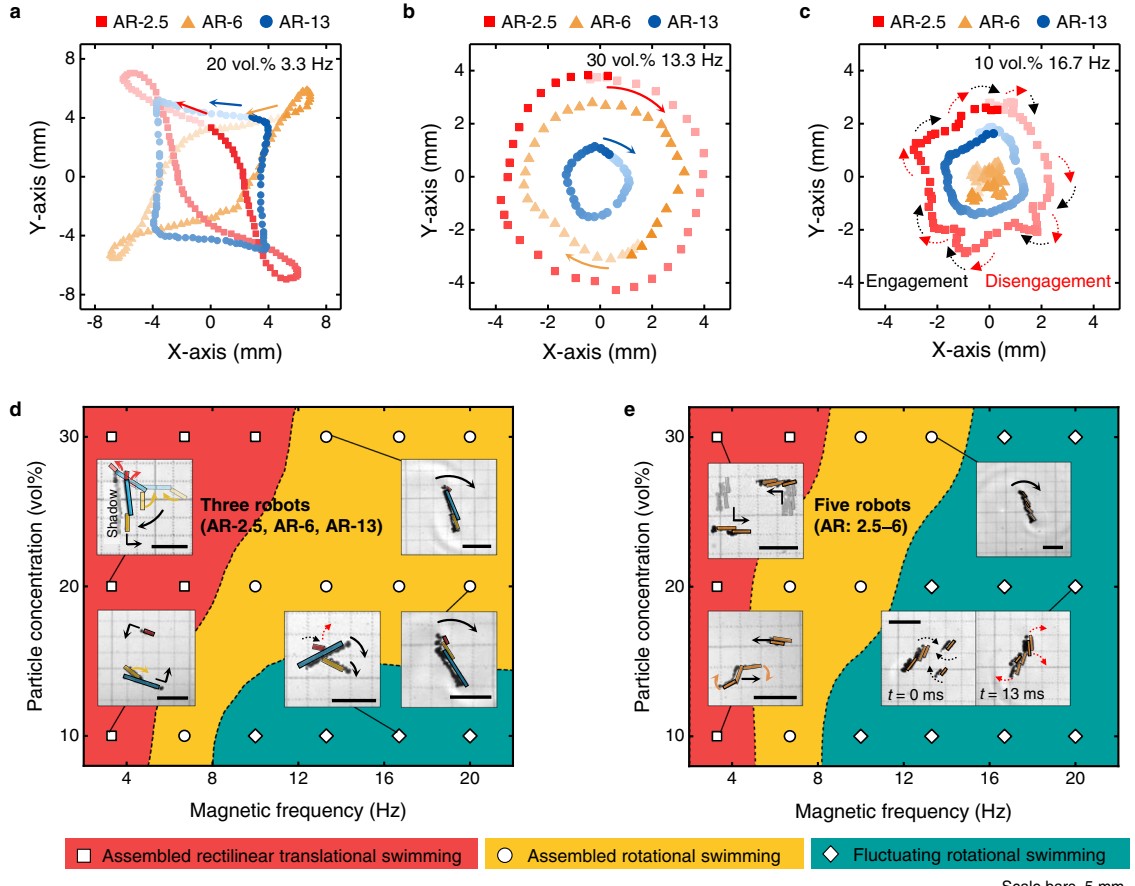

**Fig. 3 | Manipulation of multiple CNTY robots. a–c** Simultaneous actuation of three (AR-2.5, AR-6, and AR-13) CNTY robots. The *x*-axis and *y*-axis coordinates represent (**a**) assembled rectilinear translational swimming, (**b**) assembled rotational swimming, and (**c**) fluctuating rotational swimming. **d, e** Phase diagrams of collective swimming modes obtained via manipulation of multiple CNTY robots: Swimming of (**d**) three (AR-2.5, AR-6, and AR-13) and (**e**) five (AR: 2.5–6.0) CNTY robots.

rotational swimming eventually appeared at 10 Hz. Second, a higher particle concentration increased the centroid–vertex distance because a larger inertial force was induced in the CNTY robot with increased $M_s$ and mass (Fig. 2b). The effects of AR entailed a complex swimming behavior due to the interdependency of the BL, $M_s$, mass-to-inertia force, and drag force (Fig. 2c). However, the longer CNTY robots were found to require higher $M_s$ for increasing the inertial forces (Fig. 2d–f). The 30 vol.% CNTY robot was capable of bimodal swimming up to an AR of 21.5 while previous artificial magnetic swimmers allowed above-water magnetomotility up to an AR of 3.3 (see Supplementary Table 3).

**Magnetic organization**

Multiple CNTY robots were magnetically assembled and adaptively organized according to varied magnetic frequencies to perform multi-modal collective swimming (Supplementary Fig. 19, Supplementary Movie 3). The multimodal collective swimming included assembled rectilinear translational swimming, assembled rotational swimming, and fluctuating rotational swimming (Supplementary Movie 4). The swimming modes were investigated for manipulating three CNTY robots—AR-2.5, AR-6, and AR-13 (Fig. 3a–d). Figure 3a shows the assembled rectilinear translational swimming of the AR-2.5/20-vol.%, AR-6/20-vol.%, and AR-13/20-vol.% robots at 3.3 Hz. These CNTY robots were organized into a chain-like configuration, essentially the AR-21.5/20-vol.% configuration, due to dipole-dipole interactions[33]. It is worth noting that the magnetic modular assembly achieved assembled rectilinear translational swimming, whereas a single AR-21.5/20-vol.% robot was nonmotile due to the high drag force. The collective swimming can be attributed to the magnetically interacting dynamic joints between the CNTY robots, that is, soft joints. During rotation at the vertex of swimming trajectory, the magnetic modular assembly experienced viscous torque. Drag force was perpendicularly applied to the axially assembled robots, attenuating the magnetic attractive force among the soft joints. Magnetic attractive force was recovered during subsequent rectilinear translational swimming. As a result, the soft joints exhibited rotational degrees of freedom and periodically undulated to endure drag force. The trajectories of AR-2.5/20-vol.% (red squares) and AR-6/20-vol.% (yellow triangles) show two intersection points, implying dissipation and recovery of the magnetic attractive force among the soft joints during the assembled rectilinear translational swimming. The influence of dissipative magnetic interactions was confirmed using the deconvoluted harmonic sine functions of the trajectory (Supplementary Fig. 20). When the magnetic attractive force increased upon the use of 30 vol.% CNTY robots with high $M_s$, the periodic undulation of soft joints disappeared in the assembled rectilinear translational swimming of multiple CNTY robots.

When drag force became dominant over the inertia force of swimming robots, swimming mode was switched from assembled rectilinear translational swimming to assembled rotational swimming, that is, an increase in magnetic frequency or a decrease in particle concentration. During the assembled rotational swimming of the magnetic modular assembly, the *y*-axis coordinate of each robot was represented by a single sine waveform (Fig. 3b, Supplementary Fig. 21). The magnetic modular assembly of the 30 vol.% CNTY robots preserved the assembled rotational swimming up to 20 Hz. However, the 10 vol.% CNTY robots with a low $M_s$ disassembled even at a lower magnetic frequency of 10 Hz due to weak magnetic attractive force.

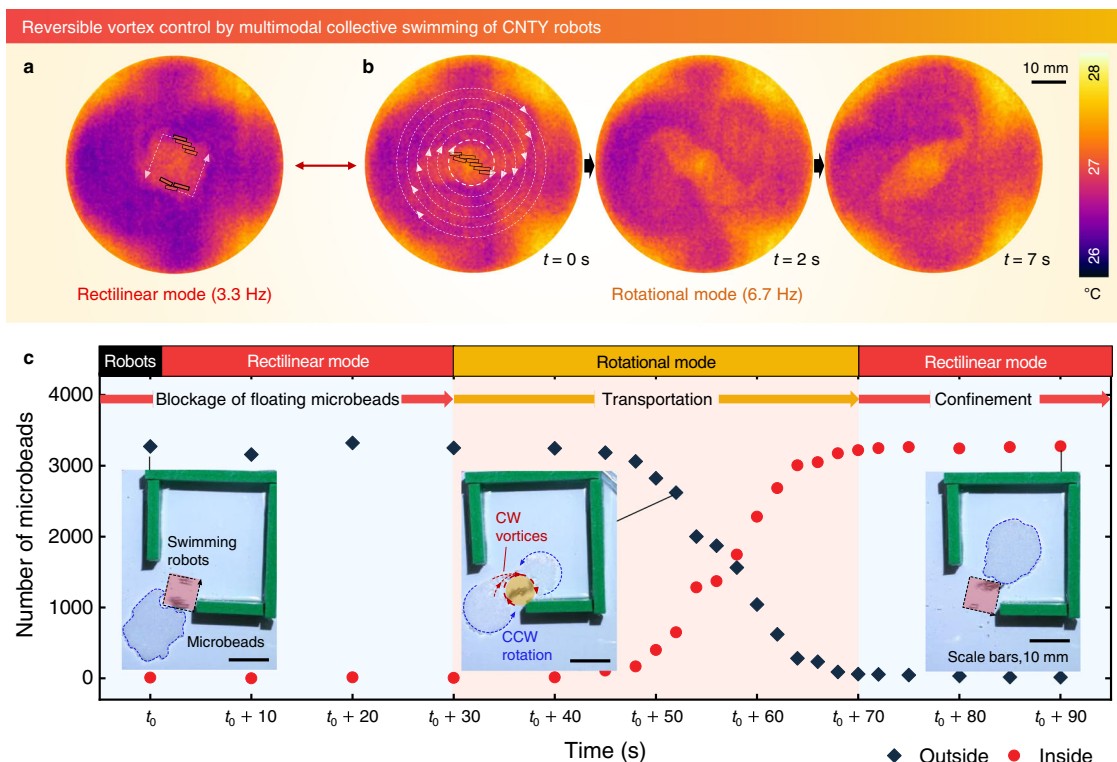

**Fig. 4 | Vortex control achieved by collective swimming of 10 vol.% CNTY robots. a, b** Vortices in a fluid generated via swimming of seven modular robots in the (**a**) rectilinear and (**b**) rotational modes. The overlaid schematics show swimming 10 vol.% CNTY robots and vortex lines. **c** Blockage, transportation, and confinement of > 3,000 floating microbeads with a diameter of 255 μm achieved through application of the appropriate swimming modes shown in (**a**) and (**b**).

Instead of assembled rotational swimming, the 10 vol.% CNTY robots showed fluctuating rotational swimming (Fig. 3c, Supplementary Fig. 22), which featured engagement and disengagement of multiple CNTY robots. The breakage of magnetic modular assembly occurred when drag force overcame magnetic attractive force[34] among soft joints. The repetition of engagement and disengagement led to a fluctuation in the distance between the AR-2.5/10-vol.% and AR-13/10-vol.% robots.

The swimming modes of the AR-2.5-, AR-6-, and AR-13-based CNTY robots are summarized in the panoptic representation of the acquired data in Fig. 3d. Interestingly, the simultaneous actuation of multiple robots improved the motility of an ineffectual robot (Supplementary Fig. 23). A single AR-13/10-vol.% robot exhibited only rotational swimming from 3.3 to 16.7 Hz. However, rectilinear translational swimming was enabled for this robot at 3.3 Hz with the assistance of the AR-6/10-vol.% robot. The AR-6/10-vol.% robot swam forward with periodic undulation to perform rectilinear translational swimming along with the magnetically linked AR-13/10-vol.% robot. At an increased magnetic frequency of 20 Hz, rotational swimming was induced for the originally nonmotile AR-13/10-vol.% robot through rotations with the AR-2.5/10-vol.% and AR-6/10-vol.% robots. Multimodal collective swimming was also observed in the case of five similar-sized coexisting CNTY robots instead of ineffectual AR-13 robots (Fig. 3e). Fluctuating rotational swimming was observed under the conditions of high magnetic frequency and low particle concentration, where drag force became dominant over magnetic attractive force applied to soft joints of the CNTY robots. Unlike in the scenario involving the accompanying AR-13 robots, the magnetic modular assembly with lower ARs could disassemble due to drag force applied to the soft joints, reducing the maximum magnetic frequency at which the assembled rectilinear translational swimming or assembled rotational swimming was permitted.

## Vortex generation

The magnetic modular assembly of multiple CNTY robots regulated the magnitude, angular velocity, and chiral direction of the fluid vortices owing to their adaptable magnetic organization (Figs. 4, 5). To visualize these vortices, the temperature of the water surface was analyzed by exposing the CNTY robots to near-infrared (NIR) light at 0.5 W cm⁻² for 3 min. Because the swimming CNTY robots radiated absorbed energy, the temperature gradient in the swirling water was observed using a forward-looking infrared (FLIR) thermal imager. For seven 10 vol.% CNTY robots, the rotational mode at 6.7 Hz led to a more significant water vortex than the rectilinear mode at 3.3 Hz (Fig. 4a, b). The spiral line on the water surface was characterized by a Rankine vortex[35], which featured a forced vortex generated at the center upon rotation of the CNTY robots (thick lines) and a surrounding free vortex (thin lines). The reversible transition between the rectilinear and rotational modes enabled the block or transport of 3,350 floating microbeads with a diameter ($D$) of 255 μm by the 10 vol.% CNTY robots to a desired location (Fig. 4c, Supplementary Fig. 24, Supplementary Movie 5). Initially, the agile CNTY robots with assembled rectilinear swimming prevented the microbeads from approaching the entrance of the divided space (Supplementary Fig. 25). Upon initiating the rotational mode, the CW free vortex dragged the microbeads toward the rotating CNTY robots. Subsequently, the forced vortex carried the microbeads into the confined space, whereas the gradients in the vortex velocity continuously transported the external microbeads. Similarly, the application of the rectilinear or rotational modes enabled confinement or release of thousands of microbeads in a devided space (Supplementary Fig. 26-a). However, a single CNTY robot was unable to carry thousands of microbeads (Supplementary Fig. 26-b), confirming the importance of collective swimming. Assembled high-AR robots with large hydrodynamic volume facilitated generation of water vortices and consequently was able to transport thousands of floating microbeads.

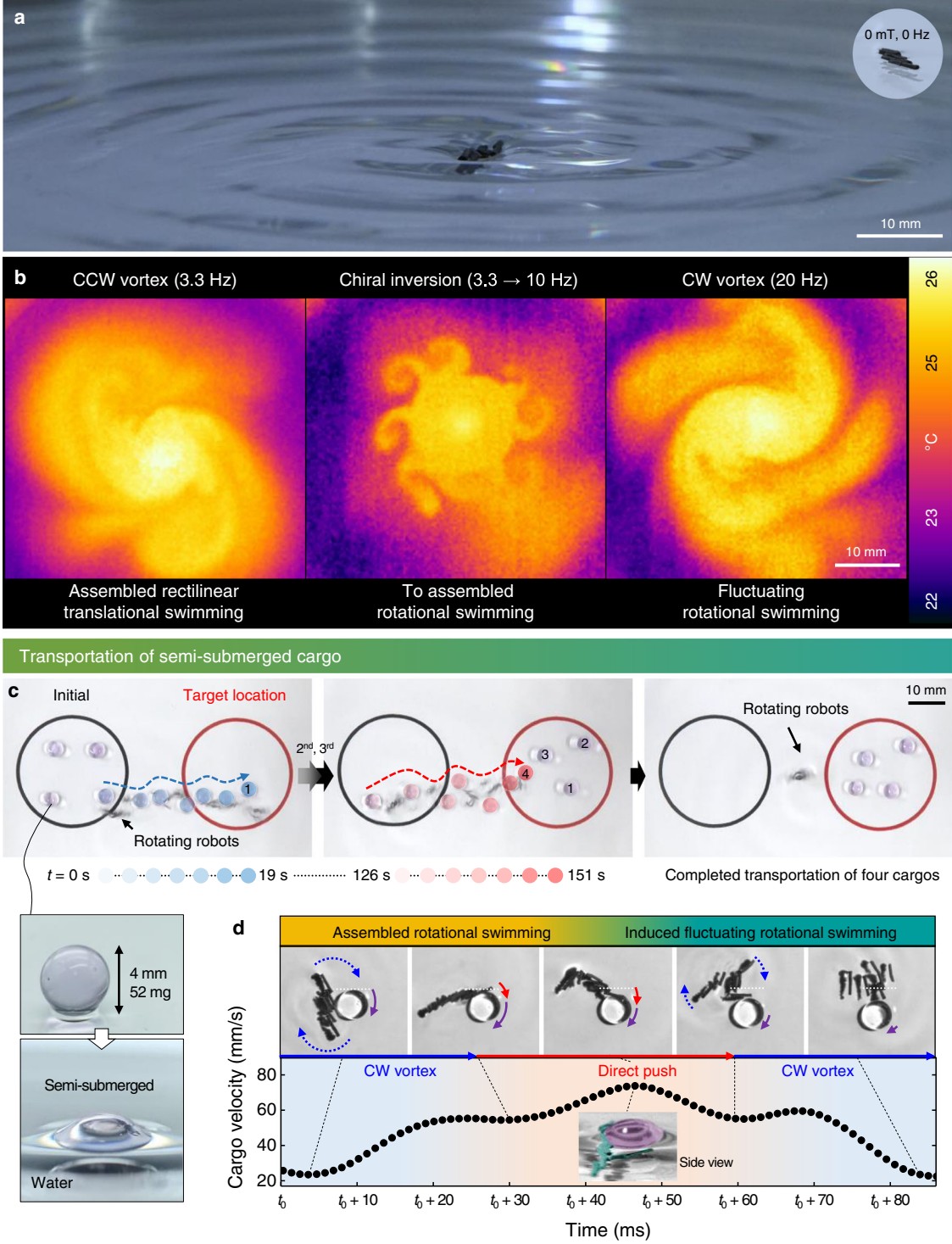

**Fig. 5 | Intensified vortices generated by swimming 30 vol.% CNTY robots.**
**a** Surface morphology of water modified by assembled rotational swimming of seven 30 vol.% CNTY robots at 13.3 Hz. The inset shows the free surface of water in the absence of the magnetic field. **b** Large-magnitude vortex generation by seven 30 vol.% CNTY robots and controllability of the vortex directions in the collective swimming modes. **c**, **d** Transportation of four semi-submerged cargos with a diameter of 4 mm. **c** Time-lapse images of cargo transport by seven 30 vol.% CNTY robots at 13.3 Hz. **d** Transported cargo by both CW vortex generation on water and direct pushing by six 30 vol.% CNTY robots.

The water vortex was intensified by increasing the particle concentration from 10 to 30 vol.%. The 30 vol.% CNTY robots were highly attracted to the electromagnetic field, which can be attributed to their high $M_s$ (see Supplementary Fig. 11), and generated a paraboloid water surface (Fig. 5a, Supplementary Fig. 27, Supplementary Movie 6). Large-magnitude vortices were observed in all three swimming modes (assembled rectilinear swimming, assembled rotational swimming, and fluctuating rotational swimming) of seven 30 vol.% CNTY robots owing to the modified surface morphology of water (Fig. 5b, Supplementary Movie 7). When the rectilinearly swimming modular robots were reorganized for achieving assembled rotational swimming, chiral inversion of the vortex occurred, resulting in a complex CCW and CW vortex in the fluid. By means of the large-magnitude vortex, multiple 30 vol.% CNTY robots transported semi-submerged spherical cargos ($D = 4$ mm)

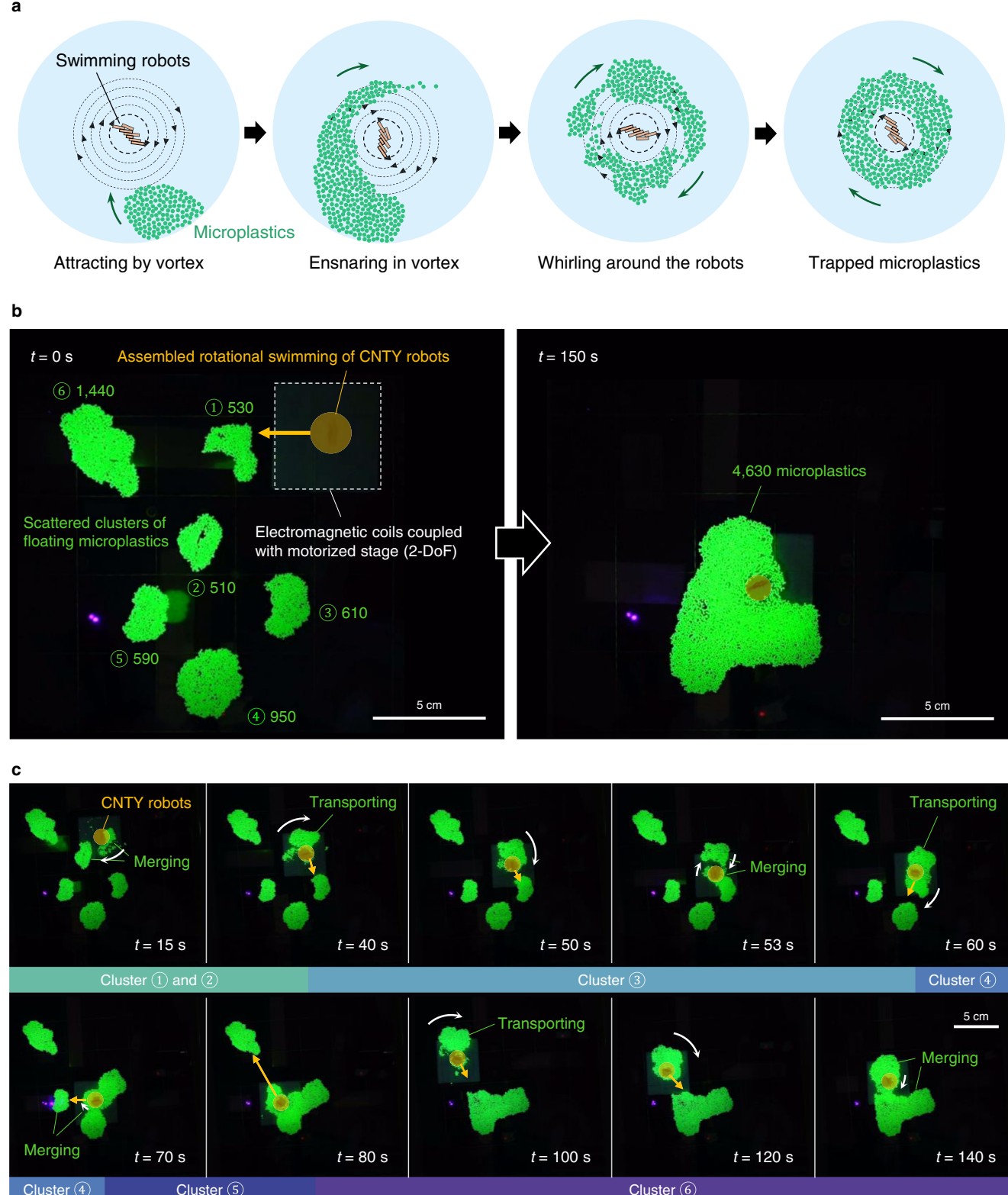

**Fig. 6 | Collection of thousands of microplastics in an open space. a** Schematic illustration gathering microplastics by assembled rotational swimming of CNTY robots. **b** Before ($t = 0$ s) and after ($t = 150$ s) collecting six scattered clusters of floating microplastics. The CNTY robots were steered by manipulating the motorized stage coupled with electromagnetic coils. **c** Time-lapsed images showing floating microplastics being collected. The diameter of microplastic was 850 μm. Seven 30 vol.% CNTY robots were manipulated at 5 Hz.

with a weight of 52 mg, which was 82 times heavier than an individual CNTY robot, at 13.3 Hz (Fig. 5c, Supplementary Movie 8). The CNTY robots were steered by slowly moving the position of the water container. Notably, the semi-submerged cargos were more difficult to be transported because of the prevention of buoyancy-induced transportation. Slow-motion analysis confirmed that in addition to the shear stress transfer via direct pushing of the CNTY robots, the CW Rankine vortex in the fluid assisted in transporting the semi-submerged spherical cargos (Fig. 5d, Supplementary Fig. 28). Although the assembled modular robots were immediately disengaged after pushing the cargo, they spontaneously reorganized to perform assembled rotational swimming. Multiple CNTY robots successfully realized the delivery of four semi-submerged spherical cargos to a target location.

To improve transportation precision of various cargos (Supplementary Movie 9), we devised a motorized stage having two degrees of freedom (2-DoF)[36] coupled with the electromagnetic coils. Through manipulating the location of the motorized stage instead of the water container, the spherical cargo with rolling resistance could be delivered with more delicacy to its designated position (Supplementary Fig. 29). The transportation capability was demonstrated even for semi-submerged or underwater cargos with sliding resistance by above-water collective swimming or underwater collective swimming, respectively (Supplementary Fig. 30).

Furthermore, seven 30 vol.% CNTY robots accomplished the collection of thousands of microplastics in an open space by generating a large-magnitude vortex (Fig. 6, Supplementary Movie 10). When the CNTY robots were adjacent to a cluster of floating microplastics, the microplastics became ensnared in the CW vortex (Fig. 6a). Under manipulation of the motorized stage, CNTY robots transported and merged the scattered clusters of floating polyethylene microbeads ($D = 850 \ \mu m$) using assembled rotational swimming (Fig. 6b, c). A total number of 4630 microplastics was successfully collected within 150 s.

## Discussion

Collective actuation of CNTY robots shows tremendous potential for expanding the functional scope of carbon-based nanomaterials developed into artificial muscles[37], supercapacitors[38], sensors[39], soft actuators[40], and three-dimensional (3D) microstructures[41]. The biomimetic hierarchical structure of these robots assisted in realizing a stiff yet lightweight long framework, imparting agile above-water swimmability of high-AR CNTY robots and thereby inducing vortices. A vortex-induced transportation strategy was introduced to enable the biomimetic nanocomposite robots to rapidly deliver thousands of miniscule cargos, which is an arduous task for gripping devices. Considering that tethered[42] or untethered[43,44] manipulation of miniaturized grippers has been implemented for delicate pick-and-place control of micro-scale objects, the transportation of numerous small objects requires the development of alternative strategies. The CNTY robots designed in this study could rapidly transport and confine thousands of microbeads into specific spaces owing to the multimodal collective swimming of their magnetic modular assembly which facilitated vortex generation and vorticity control. The vortex was intensified by collective swimming of the CNTY robots with increased thickness of the magnetic composite layer; this allowed transportation of millimeter-scale cargos that were 52 times heavier than an individual robot and collection of thousands of microplastics that were scattered at the air-water interface. The collective actuation of biomimetic nanocomposite robots is anticipated to provide versatile robotic applications for microplastic removal[45,46], vortex control in microfluidic platforms[47], and transportation of pharmaceuticals[48,49]. Moreover, because the magnetic modular assembly of CNTY robots performs on-demand multimodal collective swimming, this proof-of-concept of miniaturized modular robots suggests that more complex robotic tasks can be accomplished through the physical intelligence[50] of robot swarms.

## Methods

### Synthesis of carbon nanotube yarns

The CNTY was prepared using a floating catalyst chemical vapor deposition method[51]. Methane (Singin Gastech Co., Ltd.; 99.999%), thiophene (Sigma Aldrich; ≥99%), and ferrocene (Sigma Aldrich; 98%) were used as the carbon source, promoter, and catalyst precursor for MWNT synthesis at 1200 °C, respectively. Synthesized MWNT fibers were constantly drawn into threads in the reactor by passing through a tap-water bath followed by an acetone bath (Daejung Chemical and Metals). A thread of MWNT fibers was directly prepared on a roller at a winding speed of 5 m min⁻¹. The 100 fibers of MWNTs were twisted to form a yarn, forming a helical feature and a porous interior between the MWNT fibers.

### Dip-coating of carbon nanotube yarns

PDMS prepolymer was mixed with a crosslinking agent (Sylgard 184, Dow Corning) at a 10:1 ratio. Iron particles (CC grade, BASF) were dispersed in the PDMS prepolymer at concentrations ranging from 0 to 40 vol.%. The PDMS–iron-particle mixtures were degassed in a vacuum chamber. Six-centimeter-long CNTY specimens were dip-coated horizontally in the PDMS–iron-particle mixtures for a duration of 1 min. The dip-coated CNTY samples were drained vertically at room temperature for 10 min. The ternary nanocomposites were fully cured in an oven at 130 °C for 3 h, followed by cutting of the resultant specimens into predetermined ARs (2.5, 6, and 13) for preparing the CNTY robots.

### Magnetic actuation analysis

The CNTY robots were actuated at the air–water interface in a container filled with 6.5 mm of water under a pulsed electromagnetic field. The maximum magnitude of the electromagnetic field was measured to be 13 mT. The swimming modes of CNTY robots were changed by switching the magnetic frequency. The swimming modes were analyzed using a high-speed camera (Phantom, Micro C110, Vision Research), and the swimming trajectories were tracked using the center of mass of each CNTY robot. The vortices were visualized by analyzing the temperature of the water surface after exposing the CNTY robots to NIR light (0.5 W cm⁻²; UIM-250, UNIX) for 3 min. The heat radiation from the CNTY robots was examined using an FLIR imager (FLIR E40, Teledyne FLIR LCC). Multiple modular CNTY robots were prepared at ARs of 2.5–6 for the vortex visualization and cargo transport demonstration in Figs. 4–6. In Figs. 4 and 6, the total number of floating microbeads was calculated by measuring the area of a single microbead, the area of clustered floating microbeads, and average coverage of floating microbeads with consideration to the unoccupied space among the microbeads (see Supplementary Fig. 24). The deployed number of CNTY robots for cargo transport was minimized for actuation efficiency.

## Data availability

The data corresponding to this study are available from the first author and corresponding authors upon request.

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

## Acknowledgements

The authors acknowledge support from the Asian Office of Aerospace Research and Development (AOARD; FA2386-18-1-4103). This research was partially supported by a grant from the National Research Foundation (NRF) funded by the Korea government (NRF-2022R1A2C2002911)

and the Korea Institute of Science and Technology Open Research Program.

## Author contributions

J.J.W. conceived the project. S.W., H.E.L., Y.S.C., and K.Y. performed the experiments. S.W., H.E.L., Y.S.C., K.Y., J.E.P. analyzed the data. S.W wrote the paper, J.J.W. and S.J.Y. revised the paper and all authors provided feedback.

## Competing interests

The authors declare no competing interests.
