## [Peer Review File · Nature Communications]

Multimodal Collective Swimming of Magnetically Articulated Modular Nanocomposite RobotsREVIEWER COMMENTS

Reviewer #1 (Remarks to the Author):

This manuscript studies the multimodal collective swimming behavior of polymer-composite-based magnetic robots in detail. The structure of the nanocomposite robots mimicking the human musculoskeletal system is enlightening for designing other swimming robots. It is interesting that the swimming velocity of the robot can reach about 180 body lengths per second. The multimodal locomotion behaviors of both single and collective agents are presented. Factors that influence the swimming mode are fully investigated. Overall, this work is well organized, but some revisions are required. The comments and suggestions are listed as follows:

1. Compared with colloidal particle systems consisting of numerous colloidal particles, polymeric robots can minimize the number of robots required. Is this one of the advantages of the polymeric robot system? Is there a more specific explanation for this?
2. The second paragraph of the main body states that one of the significance of this work is “controllable collective actuation of multiple polymer-composite-based robots”. However, other studies, such as references 17, 19, 21, 22 of this manuscript, have investigated the collective behavior or self-assembly of multiple polymer robots. The novelty of this work about collective behavior compared with those studies shall be illustrated more specifically.
3. For a single nanocomposite robot, the maximum swimming velocity can reach about 180 body lengths per second, which is very amazing for the magnetic field-driven small agents. However, I think it is not “the fastest swimming velocity among the artificial swimmers”. For instance, the light-driven nanomotor reported previously can reach a speed up to 950 body lengths/s (J. Am. Chem. Soc. 2016, 138, 20, 6492–6497). Moreover, I don't think it make that much sense to compare this external field/gradient driven miniaturized swimmer with bigger battery-driven robots, as shown in Supplementary Table 2. Therefore, it is suggested to compare with other magnetic-field-driven small-scale swimmers only.
4. The rectilinear translational swimming of the robot seems to be a non-uniform speed process. Calculation of Reynolds number and characterization on the variation of velocity during the swimming process are needed. Furthermore, does the maximum speed (180 body lengths per second) refer to the average speed of the entire process or the speed of a certain moment?
5. In the five (AR: 2.5–6.0) CNTY robots' scenario, the magnetic CNTY robots are disassembled owing to the viscous forces, which is different from the scenario involving the accompanying AR-13 robots. What exactly is the difference in viscous forces between the two scenarios?
6. The collective CNTY robots in rectilinear mode can also generate a water vortex like that in rotational mode (although less significant). So, in the demonstration of transporting 830 floating microbeads (Fig. 4b and Supplementary Movie 5), why the rectilinear mode can prevent the microbeads from approaching the entrance?
7. In the transportation of semi-submerged cargo (Fig. 5c and Supplementary Movie 8), it seems that the collective robot cannot precisely transport the cargo to a specific point. Dose it because the separation of the robots and cargo is hard to control?
8. In the transportation of both floating microbeads and semi-submerged cargo, the position of the collective robots is changed by moving the water container. It seems that the general way to change the relative position of the rotational robots is moving the entire surrounding environment. However, in many scenarios, it is difficult to move the environments. How to solve this problem? Moreover, how can this robotic system be used for drug delivery?
9. What does the triangle symbol in Fig. 2e mean? Should it probably be a square symbol?
10. The color and symbols are not matched in Fig. 3e, which is easy to cause confusion.

Please correct it. The second sentence of Supplementary Table 2 (“The vol.% notation ...”) should belong to Supplementary Table 1.

Reviewer #2 (Remarks to the Author):

Summary:

The paper describes a methodology of targeted drug delivery where superparamagnetic beads are aerosolized and deployed using a nebulizer. Once released using a nebulizer, the authors apply rotational magnetic fields to induce rolling motion of the bead agglomerates formed in an aqueous film, described as microwheels, which can translate to a desired spot. As an application, the authors show a 3-D printed lung model whereby they attempt targeted delivery of these beads to branches using combined application of a permanent magnet (to create a bolus), and thereon rotational magnetic field (for subsequent localization).

General comments:

The manuscript describes CNTY robots, nanocomposite robots that mimic the musculoskeletal system, and hold the ability for multimodal collective motion and manipulation at air-water interface. These robots comprise of light and stiff network of MWCNTs and are coated with a magnetic microparticles using PDMS as a binder material. Further, the motion of these CNTY robots is demonstrated for a single robot and multi-robot systems. Finally, some applications are presented where collective motion of CNTY robots are used for cargo manipulation. Overall, the manuscript is well written and has an extensive description of fabrication and characterization of CNTY robots. However, there are several concerns regarding the swimming motion of these robots, particularly with regards to collective motion of multiple such robots. Moreover, while many applications for targeted manipulation are presented, the inability of these CNTY robots to move underwater does not justify the proposed application of targeted drug delivery as described by the authors. These above mentioned considerations have been summarized below as technical (scientific concerns) and detailed (addresses readability) comments.

Technical comments:

- Overall, my main criticism of CNTY robots is that they could only operate on the air-water interface. How do you address applications such as drug delivery as cited in the abstract/outlook? A majority of drug delivery applications require therapeutic agents to be submerged in water.
- Regarding the proposed application to manipulate semi-submerged objects, can authors propose a method where such CNTY robots can be made less buoyant so that they submerge in water instead? Please justify.
- The Introductory paragraphs describes the state-of-the-art well and in a succinct manner albeit has scope for some more clarifications:
 - o “Reconfigurable swarms of magnetic microparticles have been recently realized by varying the axes and frequencies of an alternating magnetic field”, motion differentiation in microrobots is also observed for different field precession, reference below:
Mohanty, S., Jin, Q., Furtado, G. P., Ghosh, A., Pahapale, G., Khalil, I. S., ... & Misra, S. (2020). Bidirectional Propulsion of Arc-Shaped Microswimmers Driven by Precessing Magnetic Fields. *Advanced Intelligent Systems*, 2(9), 2000064.

o “Although particle swarms demonstrate micro-cargo transportation capabilities, numerous robots are required to achieve them; moreover, the long-distance transport of multitudinous cargos in a short duration is difficult”, Why is it necessary to rely on swarms of microrobots e.g. Ren et al. also demonstrates such manipulation reliably albeit one at a time:

Ren, L., Nama, N., McNeill, J. M., Soto, F., Yan, Z., Liu, W., ... & Mallouk, T. E. (2019). 3D steerable, acoustically powered microswimmers for single-particle manipulation. *Science advances*, 5(10), eaax3084.

- In the description of synthesis procedure of CNTY robots, it could be clarified explicitly that MWCNTs were used to prepare the yarn. It is only specified in Supplementary Information (SI) and may be useful to defend the physical properties described in the main body of the manuscript.

- Why are the specific aspect ratios of CNTY robots - 2.5,6,13 and 20 chosen for the final experiments?

- Please mention the magnitude of magnetic fields used for the two swimming modes presented in subsequent swimming modes of CNTY robots.

- Regarding the experiments shown in magnetic organization:

How do you ensure that assembly of higher number of CNTY robots (e.g. case of 5 or more robots shown in the movies) does not break itself at high frequencies magnetic fields? My main concern is that the assembly and disassembly of multiple CNTY robots is not very clear. Some basic assumptions/quantifications must be clarified:

o At what frequency is the disassembly of robots triggered?

o What number of robots are optimal for collective swimming?

- Regarding the fluctuating rotational swimming mode discussed in the manuscript:

o It seems that in this case (SI Movie 3) the assembly of CNTY robots breaks at high frequencies?

o What is the utility of fluctuating rotational swimming?

o Besides, defining the range where CNTY robots disassemble, it is not clear how this mode contributes to the utility of collective CNTY motion?

- Regarding the vortex generation experiments:

o Although the demonstration looks promising, the direction in which payload of microbeads can be carried looks subjective to the presence of an obstacle. Case in point, beads go into the confined area when CNTY robots are rotating next to the bottom green edge whereas they come outside the confined area when CNTY robots are rotating next to the top green edge. How could CNTY robots manipulate objects in absence of any edges or obstacles? Please justify.

o In SI Movie 8, the motion of CNTY carrying cargo is triggered by manual movement of the container, hence such manipulation cannot take place in absence of an external stimulus.

How do authors explain this application for inaccessible or confined workspaces? Secondly, the rolling motion spherical cargo assisted in the manipulation which may not otherwise work for irregular/non-spherical geometries.

- In SI Fig. 7 - at what applied magnetic field does saturation M_s occur?

Detailed comments:

- Within Section: Biomimetic hierarchical structure of robots:

o “The human arm exerts a strong force in proportion to the thickness of muscle bundles”, this claim sounds vague, could you please provide numbers here?

- Within Section: Agile multimodal swimming:

o “CW rotational swimming by increasing the magnetic frequency to 6.7 Hz.”, please refer SI Fig. 8, and first part of SI Movie 2a.

o “When the magnetic frequency was switched from 3.3 Hz to 6.7 Hz, the centroid–vertex

distance”, define centroid-vertex distance. I assume this is related to the square-shaped trajectory, but the term is not well defined in any of the figures.

- Within Section: Magnetic organization,

- o “the viscous force applied to the robots causes dissipation of the magnetic attractive force in the soft joints”, the sentence is unclear.

- o In SI Fig 15, is it deconvoluted sine curves of only one AR i.e., 2.5 case?

- Fig. 5 caption: 13.3Hz - description in SI Fig. 20 suggest this to be camera imaging frequency. Please clarify.

- Please check the angular frequency described in SI Note 1, they have inconsistent units.

Reviewer #1

This manuscript studies the multimodal collective swimming behavior of polymer-composite-based magnetic robots in detail. The structure of the nanocomposite robots mimicking the human musculoskeletal system is enlightening for designing other swimming robots. It is interesting that the swimming velocity of the robot can reach about 180 body lengths per second. The multimodal locomotion behaviors of both single and collective agents are presented. Factors that influence the swimming mode are fully investigated. Overall, this work is well organized, but some revisions are required.

Response to comment

We thank the reviewer for the positive evaluation and valuable comments on our manuscript.

Q1) Compared with colloidal particle systems consisting of numerous colloidal particles, polymeric robots can minimize the number of robots required. Is this one of the advantages of the polymeric robot system? Is there a more specific explanation for this?

Response to comment

A1) Polymeric robots are promising in robotics since diverse structures composed of multi-material can be constructed through various polymer processing techniques. Polymer-composite-based robots can be fabricated in arbitrary complex shapes such as meshes, needles, jellyfishes, cilia, and helices by patterning, printing, molding, or thermal fixation of photo-curable, thermo-curable, and thermo-processible polymers. In this study, we fabricated ternary hierarchical nanocomposites by dip-coating and thermo-curing processes with the purpose to realize magnetic robots that can generate a water vortex and transport multiple cargos via the vortex.

In designing the magnetic robot, a lightweight robot body was essential to impart agile above-water swimmability. However, importantly, a stiff body frame was also required to overcome surface tension and drag force for swimming. Considering the well-known Ashby plots (modulus vs density), rigidity and lightweightness are in trade-off relationships. For example, robots with high Young's modulus can result in considerable vortex generation. However, increase in Young's modulus indicates that high density materials are necessary according to the Ashby plots. Furthermore, metal-based particle robots can cause submergence in water due to the high density. Therefore, we employed carbon nanotube yarn (CNTY) as rigid yet lightweight nanoporous framework as well as permeable template for dip-coating of thermo-curable magnetic polymer composites to address this tradeoff relationship.

Moreover, colloidal particles suffer from limited geometric and compositional variation. The low aspect ratios of particle robots are because the colloids are synthesized in a direction that minimizes the surface energy of each particle. However, we conceived a magnetic robot with high aspect ratio (AR) and their magnetic assembly as a more effective strategy for generating the vortex. As the magnetically assembled high-AR robots had a large hydrodynamic volume, rotational swimming of the magnetically assembled robots was expected to exert high rotational force and induce a large-magnitude vortex. To preserve the assembled structure even during the rapid swimming, magnetic attractive force among robots should exceed drag force. To enhance interrobot magnetic attractions, we designed core-shell-like robot by binding highly concentrated magnetic particles only on the robot surface, instead

of embedding the particles inside the robot since magnetic force between the particles drastically increases by decreasing the interparticle distances (A. Mehdizadeh et al., *Acta Mechanica Sinica*, **26**, 921–929, 2010).

To meet the requirements, we introduced a biomimetic hierarchical nano/microstructure using nanoporous CNTY framework, PDMS, and magnetic microparticles, which emulate the spongy bone, connective tissue, and skeletal muscle, respectively. First, stiff yet lightweight nano-framework was prepared by synthesizing multi-walled carbon nanotube (MWNT), followed by twisting the MWNT to form the nanoporous CNTY framework (newly added Supplementary Fig. 1, 2, and 3). Then, dip-coating process was implemented using a mixture of PDMS prepolymer and magnetic microparticles to bind highly concentrated magnetic particles on the robot surface. The nanoporous CNTY framework was immersed in the mixture of PDMS prepolymer and magnetic microparticle, followed by drainage of excessive mixture. Because the high concentration of magnetic particles could cause a drastic increase in the viscosity of the composite mixture, we handled the composite mixture by utilizing the prepolymer. Moreover, during the dip-coating process, the nanoporous CNTY framework operated as a membrane to permeate the liquid-phase PDMS prepolymer through capillary forces, whereas the 5- μm -sized magnetic microparticles were excluded from the core region. The magnetic microparticles at the CNTY surface were bound through thermal curing of PDMS via hydrosilylation polymerization. The PDMS acted as a binder for the magnetic particles in a fashion similar to the deep fascia which binds muscle fibers. Absence of the PDMS binder could hinder the adhesion of magnetic particles onto the CNTY framework due to the weak cohesive energy between the magnetic particles. We also could varied the coating thickness of the magnetic composite layer by changing the concentration of magnetic particles (Supplementary Fig. 8).

As a result, the AR-2.5/30-vol.% robot accomplished swimming velocity of 180 BL s^{-1} . We previously reported polymer-composite-based robot with AR-2, showing only above-water swimming velocity of 3.8 BL s^{-1} (S. Won et al., *Nature Communications*, **10**, 4751, 2019). The low swimming velocity and low-AR could not generate vortex. Maximum AR was 4 for magnetic actuation of the polymer-composite-based robots. Furthermore, for the high-AR CNTY robots of AR-6/30-vol.% and AR-13/30-vol.% CNTY robots, the maximum swimming velocity was measured to be 66 BL s^{-1} and 40 BL s^{-1} respectively.

We also would like to highlight polymer-composite-based robot systems in terms of the actuation efficiency by minimizing the number of robots. Only with a single robot, cargos could be transported to arbitrary destinations (L. Ren et al., *Science Advances*, **5**, eaax3084, 2019; C. Li et al, *Science Robotics*, **5**, eabb9822, 2020). However, these demonstrations were the cases of a single cargo smaller than the robot body length. Although we recently reported that tens of robots could cooperate to carry a larger and heavier cargo than the individual robots (S. Won et al., *Nature Communications*, **10**, 4751, 2019), further challenge remains to deliver numerous cargos simultaneously and rapidly. In this study, the number of CNTY robots was seven for cargo transportation of thousands of floating microbeads and semi-submerged millimeter-scale bead 52 times heavier than the CNTY robot.

As it seems that our intended design principle and significance of work were not fully delivered from the previous manuscript, we extensively revised the manuscript as follows:

1) Introduction in manuscript (page 2)

Previous texts

Bioinspired polymeric structures are anticipated to improve the actuation efficiencies of miniaturized robots while minimizing the number of robots required. The locomotion of aquatic and terrestrial living creatures is a noteworthy source for realizing miniaturized polymer-composite-based robots^{16,17}. Appropriate robotic architectures have been constructed using photo-curable^{18,19}, thermo-curable^{20,21}, and thermo-processable²² polymers. Moreover, magnetic materials have been randomly dispersed²³, anisotropically aligned²⁴, or selectively deposited²⁵ in these polymeric structures to realize magnetic actuation. However, studies on polymer-composite-based magnetic robots have primarily described the independent locomotion of a single robot. Controllable collective actuation of multiple polymer-composite-based robots is required to achieve functionalities beyond those offered by single-robot systems.

Revised texts

Geometric changes in miniaturized robots are anticipated to improve actuation efficiencies of the robots for cargo transportation while minimizing the number of robots required. Polymeric robots are particularly promising since diverse structures, composed of multi-material, can be constructed through polymer processing techniques. For example, polymer-composite-based robots can be fabricated into meshes¹⁹, needles²⁰, jellyfishes²¹, cilia²², and helices²³ by patterning, printing, molding, or thermal fixation of photo-curable, thermo-curable, and thermo-processible polymers. A single polymeric robot in capsule²⁴ or cross²⁵ shape could transport a miniscule cargo to arbitrary destinations. To control these polymeric robots magnetically, a magnetic metal-layer was deposited on the capsule-shaped photoresist²⁴, and magnetic nanowires were embedded in the cross-shaped hydrogel with anisotropic alignment²⁵. However, cargo transportation has primarily focused on a single cargo smaller than the body length of polymeric robots. Recently, tens of robots cooperated to carry an object that was larger and heavier than the individual robots²⁶, yet further challenges remain to deliver multiple cargos rapidly and simultaneously. Controllable collective actuation of multiple polymer-composite-based robots is required to achieve functionalities beyond those offered by single-robot systems.

Newly added references

19. Nguyen, K. T. *et al.* A Magnetically guided self-rolled microrobot for targeted drug delivery, real-time x-ray imaging, and microrobot retrieval. *Adv. Healthc. Mater.* **10**, 2001681 (2021).
20. Lee, S. *et al.* A needle-type microrobot for targeted drug delivery by affixing to a microtissue. *Adv. Healthc. Mater.* **9**, 1901697 (2020).
21. Ren, Z., Hu, W., Dong, X. & Sitti, M. Multi-functional soft-bodied jellyfish-like swimming. *Nat. Commun.* **10**, 2703 (2019).
22. Gu, H. *et al.* Magnetic cilia carpets with programmable metachronal waves. *Nat. Commun.* **11**, 2637 (2020).
23. Won, S., Je, H., Kim, S. & Wie, J. J. Agile underwater swimming of magnetic polymeric microrobots in viscous solutions. *Adv. Intell. Syst.* **4**, 2100269 (2022).

2) Results in manuscript (page 3-4)

Previous texts

Biomimetic hierarchical nano/microstructures were prepared using a nanoporous carbon nanotube yarn (CNTY) framework, polydimethylsiloxane (PDMS), and magnetic microparticles, which emulate the spongy bone, connective tissue, and skeletal muscle, respectively, in brachial anatomy (Fig. 1a and b-(i)). The human arm exerts a strong force in proportion to the thickness of muscle bundles by means of the skeletal muscle surrounding lightweight yet stiff bones. The spongy bone component of the long compact bones articulated in the elbow comprises hierarchically arranged collagen fibers and porous networks, and remains lightweight despite containing a packed bone marrow. Carbon nanotubes (CNTs) with an exceptional stiffness (Young's modulus ~ 1 TPa) and low density²⁶⁻²⁸ were twisted to construct the spongy-bone-mimicking CNTY framework. One hundred strands of 7-nm-thick CNTs were twisted to prepare a 340- μm -thick nanoporous CNTY framework (Fig. 1b-(ii) and (iii)).

Revised texts

For a magnetic robot to be capable of vortex generation at the air-water interface as well as cargo transportation, a lightweight robot body was essential to impart agile above-water swimmability. A stiff body frame was also required to overcome surface tension and drag force in swimming even though the two variables of lightweightness and rigidity are considered trade-offs according to Ashby plots²⁷ showing Young's modulus versus density. We further conceived a magnetic robot with high aspect ratio (AR) and collective actuation of the high-AR robots as a more effective strategy for generating the vortex. When the high-AR robots assembled magnetically to have large hydrodynamic volume, rotational swimming of the magnetically assembled robots was expected to exert high rotational force and induce a large-magnitude vortex. Magnetic assembly of the multiple robots required larger magnetic attractive force among robots than drag force to preserve the assembled structure even through rapid swimming. To enhance inter-robot magnetic attractions, we considered a core-shell-like robot by binding highly concentrated magnetic particles only on the robot surface, instead of embedding particles inside the robot since the magnetic force between particles drastically increases by decreasing interparticle distances^{28,29}. The robot design that met these requirements was inspired by brachial anatomy (Fig. 1a, b-(i)). Movements of the human arm are produced by forces generated from skeletal muscles surrounding the stiff long bones. Along the interior toward the end of the long bones, the spongy bone shows a hierarchical composition of collagen fiber bundles followed by a porous network while the bone marrow fills in the bone. The porosity of spongy bone offers lightness to the long bones while surrounding skeletal muscles are bounded by the connective tissue known as deep fascia. We introduced a biomimetic hierarchical nano/microstructure with nanoporous carbon nanotube yarn (CNTY) framework, polydimethylsiloxane (PDMS), and magnetic microparticles, emulating the spongy bone, connective tissue, and skeletal muscle, respectively. First, the spongy-bone-mimicking nano-framework was prepared using multi-walled carbon nanotube (MWNT) fibers synthesized through floating catalyst chemical vapor deposition (Supplementary Fig. 1). The synthesized

36- μm -thick MWNT fiber was composed of ten-walled 14 nm-thick MWNTs. The MWNT fiber provided specific strength of 1.19 N tex^{-1} (equivalent to $1.19 \text{ GPa g}^{-1} \text{ cm}^3$) and specific modulus of 46.9 N tex^{-1} (Supplementary Fig. 2). For lightweight nanoporous structure, high densification was intentionally restrained in the MWNT fiber, confirmed by Raman spectra (Supplementary Fig. 3, Fig. 1b-(ii), (iii)). To build the stiff yet lightweight nano-framework, 100 MWNT fibers were twisted to form the 340- μm -thick CNTY (Supplementary Fig. 4).

Then, a dip-coating process was implemented using a mixture of PDMS prepolymer and iron microparticles to bind highly concentrated magnetic particles on the robot surface. The nanoporous CNTY framework was immersed in the mixture of PDMS prepolymer and magnetic microparticles, followed by drainage of excessive mixture (Supplementary Fig. 5). During the dip-coating process, the nanoporous CNTY framework operated as a membrane to permeate the liquid-phase PDMS prepolymer through capillary forces³⁰, whereas the 5- μm -sized iron microparticles³¹ were excluded from the core region, as shown in Fig. 1b-(i). The magnetic microparticles at the CNTY surface were bound through thermal curing of PDMS via hydrosilylation polymerization. The PDMS acted as a binder for the magnetic particles in a fashion similar to the deep fascia and mimicked the binding of muscle fibers. Absence of the PDMS binder could hinder the adhesion of magnetic particles onto the CNTY framework due to the weak cohesive energy between the magnetic particles. By spatially selective coating of the magnetic microparticles onto the surface of the nanoporous CNTY framework, the coating thickness of the magnetic composite layer could be varied with the concentration of magnetic particles dispersed in the PDMS prepolymer (Supplementary Fig. 6, 7, 8).

3) Supplementary Fig. 1 in Supplementary Information (page 5)

Newly added figure

Supplementary Fig. 1 Physical properties of multi-walled carbon nanotube (MWNT). Scattering electron microscope (SEM) images of 36-μm-thick single MWNT fiber in (a) low-magnification (top-down view) and (b) high-magnification (radial section view). (c) Transmission electron microscope (TEM) image (top-down view) of a ten-walled 14 nm-thick MWNT composing of the MWNT fiber. (d) Obtained the MWNT fiber with high purity, confirmed by thermogravimetric analyzer (TGA) with a heating rate of 10 °C min⁻¹ in an air atmosphere. The weight loss of the MWNT fiber was 86% and residual weight was 6%.

4) Supplementary Fig. 2 in Supplementary Information (page 6)
Newly added figure

Supplementary Fig. 2 Stress-strain curve of the single MWNT fiber. Tensile test was performed at 3 mm min^{-1} and gauge length was 1 cm. Specific strength and specific modulus reached up to $1.19 \pm 0.09 \text{ N tex}^{-1}$ (equivalent to 1.19 GPa/SG) and $46.9 \pm 4.5 \text{ N tex}^{-1}$, respectively ($n= 10$). SG denotes specific gravity, the density of the materials divided by the density of water. The bulk density of MWNT fiber was 2.6 g cm^{-3} .

5) *Supplementary Fig. 3 in Supplementary Information (page 7)*

Newly added figure

Supplementary Fig. 3 Raman spectra (inset: polarized Raman spectra) of synthesized MWNT. The intensity ratio of the D-band to G-band (I_D/I_G) in the Raman spectra, which states the crystallinity of CNT, was approximately 0.43 ± 0.11 ($n= 5$). The G peak intensity ratio ($I_{G||}/I_{G\perp}$) of the parallel direction to the perpendicular direction along to the CNTY axis in the polarized Raman spectrum, which states the orientation of CNT, was 2.54 ± 0.77 ($n= 5$). For lightweight nanoporous structure, high densification was intentionally restrained in

the MWNT fiber, confirmed by the crystallinity and orientation^{23,24}.

6) Results in manuscript (page 5)

Previous texts

The geometry of the CNTY robots was regulated to yield different aspect ratios (ARs) of 2.5, 6, 13, and 21.5. Each CNTY robot is denoted by its AR and the concentration of magnetic particles in the PDMS prepolymer (for example, AR-2.5/30-vol.%).

Newly added texts

The geometry of the CNTY robots was regulated to yield different aspect ratios (ARs) such as 2.5, 6, and 13. High-AR CNTY robots were facily obtained owing to the dip-coating process of CNTYs. Each CNTY robot is denoted by its AR and the concentration of magnetic particles in the PDMS prepolymer (for example, AR-2.5/30-vol.%).

7) Results in manuscript (page 6)

Previous texts

Moreover, the CNTY robot exhibited an above-water swimming velocity that was 47 times that of the polymer-composite-based robot (spinbot) previously developed by our group, which was actuated by two continuously rotating linearly connected permanent magnets²⁷. The agility of the CNTY robots can be attributed to not only the lightweight yet stiff biomimetic structures but also the synchronized rotation frequency of the CNTY robots with the magnetic frequency of the pulsed electromagnetic field.

Newly added texts

Moreover, the ternary-nanocomposite-based CNTY robots exhibited an above-water swimming speed 47 times faster than that of the binary-nanocomposite-based robot (spinbot) previously developed by our group, which was actuated by two continuously rotating linearly connected permanent magnets²⁶. The agility of the CNTY robots can be attributed to not only the stiff yet lightweight biomimetic structures but also the synchronized rotation frequency of the CNTY robots with the magnetic frequency of the pulsed electromagnetic field. In addition to the AR-2.5, AR-6/30-vol.% and AR-13/30-vol.% CNTY robots demonstrated agile above-water swimmability up to 66 BL s^{-1} and 40 BL s^{-1} , respectively.

8) Results in manuscript (page 6)

Previous texts

The 30 vol.% CNTY robot was capable of bimodal swimming up to an AR of 21.5.

Revised texts

The 30 vol.% CNTY robot was capable of bimodal swimming up to an AR of 21.5 while previous artificial magnetic swimmers allowed above-water magnetomotility up to an AR of 3.3 (see Supplementary Table 3).

Q2) The second paragraph of the main body states that one of the significance of this work is “controllable collective actuation of multiple polymer-composite-based robots”. However, other studies, such as references 17, 19, 21, 22 of this manuscript, have investigated the collective behavior or self-assembly of multiple polymer robots. The novelty of this work about collective behavior compared with those studies shall be illustrated more specifically.

Response to comment

A2) We thank the reviewer for the constructive comment. We have revised Introduction more comprehensively to introduce the polymer-composite-based robots and address the limitations of cargo transportation in single or multi-robot systems of the polymeric robots. As we mentioned in the response to comment (Q1), the object of the previous references was to describe examples of manifold polymeric structure which can be constructed by polymer processing of photo-curable, thermo-curable, or thermo-processible polymers.

- Introduction in revised manuscript (page 2)

For example, polymer-composite-based robots can be fabricated into meshes¹⁹, needles²⁰, jellyfishes²¹, cilia²², and helices²³ by patterning, printing, molding, or thermal fixation of photo-curable, thermo-curable, and thermo-processible polymers. A single polymeric robot in capsule²⁴ or cross²⁵ shape could transport a miniscule cargo to arbitrary destinations. To control these polymeric robots magnetically, a magnetic metal-layer was deposited on the capsule-shaped photoresist²⁴, and magnetic nanowires were embedded in the cross-shaped hydrogel with anisotropic alignment²⁵. However, cargo transportation has primarily focused on a single cargo smaller than the body length of polymeric robots. Recently, tens of robots cooperated to carry an object that was larger and heavier than the individual robots²⁶, yet further challenges remain to deliver multiple cargos rapidly and simultaneously.

Herein, we would like to emphasize that multimodal collective swimming of multiple CNTY robots achieved rapid transportation capability of a number of cargos, which is differentiated from the previously mentioned references as below. We also deleted references that may be misunderstood with respect to actuation of multiple robots and their collective behavior.

(1) Previous reference 17 (current 22): actuation of a single robot system

Gu, H. *et al.* Magnetic cilia carpets with programmable metachronal waves. *Nat. Commun.* 11, 2637 (2020).

→ **Multiple artificial cilia were anchored at the polymeric substrate and individual cilium was difficult to move arbitrarily.** In this paper, collective behavior referred to the metachronal wave by deformation of artificial cilia which were anchored to the non-magnetic polymeric body. Meanwhile, the entire body of the single polymeric robot was composed of the cilia and non-magnetic polymeric substrate. Only the single polymeric robot demonstrated locomotion of crawling or rolling to propel forward. In our manuscript,

collective behavior of CNTY robots refers to the manipulation of multiple robots for achieving complex tasks which is difficult to perform with a single robot. For example, the multiple CNTY robots enabled transportation of thousands of floating microplastics in an open space when the position of CNTY robots was arbitrary moved by controlling the location of magnetic device underneath the CNTY robots (newly added Fig. 6).

- Newly added Fig.6 in manuscript (page 17)

collected. The diameter of microplastic was 850 μm . Seven 30 vol.% CNTY robots were manipulated at 5 Hz.

(2) Previous reference 19 (current deleted): non-actuatable polymeric architectures

Cho, W. *et al.* Programmable building blocks via internal stress engineering for 3D collective assembly. *Adv. Mater. Technol.* 5, 2000758 (2020).

→ **The polymeric blocks were static and did not demonstrate locomotion.** In this paper, the collective assembly referred to hierarchical structures of static polymeric blocks. By patterning photo-curable polymers, the multi-layered hierarchical structure was fabricated. We have removed the reference to avoid misunderstanding the polymeric architecture was actuatable.

(3) Previous reference 21 (current deleted): actuation of anchored micropillar arrays

Park, J. E. *et al.* Enhancement of magneto-mechanical actuation of micropillar arrays by anisotropic stress distribution. *Small* 16, 2003179 (2020).

→ **The micropillar array did not demonstrate an arbitrary movement required for robot locomotion.** The micropillars were anchored to the substrate, similar to cilia, for bending and twisting motions. In our manuscript, the multiple CNTY robots could be steered arbitrarily at the air-water interfaces, performing vortex-induced cargo transportation.

(4) Previous reference 22 (current 26): low swimming speed of low-AR robots

Won, S., Kim, S., Park, J. E., Jeon, J. & Wie, J. J. On-demand orbital maneuver of multiple soft robots via hierarchical magnetomotility. *Nat. Commun.* 10, 4751 (2019).

→ **The low-AR robots with low swimming speed could not generate water vortex.** Above-water swimming speed of the AR-2 robot reached only 3.8 BL s^{-1} and high-AR robots (> 4) were unable to swim. However, agile above-water swimmability of high AR-robots is essential to generate vortex and transport multiple cargos rapidly. Although tens of the robots performed collective behavior to carry a larger and heavier object than the individual robots, the robots were difficult to block or transport numerous cargos due to slow and unimodal above-water swimming of low-AR robots. To address the limited capability of cargo transportation, in this manuscript, we prepared the biomimetic stiff yet lightweight robots with high ARs. For AR-2, AR-6, and AR-13 CNTY robots, the maximum swimming speed was measured to be 180 BL s^{-1} , 66 BL s^{-1} , and 40 BL s^{-1} , respectively. In our manuscript, agile above-water swimming of the magnetically assembled high-AR robots facilitated the generation of water vortex. Moreover, multimodal swimming of the multiple robots provided controllability of vorticity, thereby blocking, transporting, and confining the thousands of microbeads.

Q3) For a single nanocomposite robot, the maximum swimming velocity can reach about 180 body lengths per second, which is very amazing for the magnetic field-driven small agents. However, I think it is not “the fastest swimming velocity among the artificial swimmers”. For instance, the light-driven nanomotor reported previously can reach a speed up to 950 body lengths/s (J. Am. Chem. Soc. 2016, 138, 20, 6492–6497). Moreover, I don't think it make that

much sense to compare this external field/gradient driven miniaturized swimmer with bigger battery-driven robots, as shown in Supplementary Table 2. Therefore, it is suggested to compare with other magnetic-field-driven small-scale swimmers only.

Response to comment

A3) We thank the reviewer for the insightful comment. As suggested by the reviewer, we now have compared the swimming speed of magnetic robots only as shown in Fig. 1e and Supplementary table 2 and 3. The body mass was revised to be body length since the mass information rarely reported as well as we would like to note the agile swimmability of CNTY robots despite high aspect ratios. As shown in the Supplementary table 3, the previous magnetic robots exhibited only a few body lengths per second ($0.55\sim 17 \text{ BL s}^{-1}$) in the single robot systems and their aspect ratios were ranged from 1 to 5. Swimming speed of AR-2.5, AR-6, and AR-13 CNTY robots reached 180, 66, and 40 BL s^{-1} , respectively. Particle swarming system was excluded as collective actuation of assembled particle robots affects the swimming speed. We analyzed the swimming speed for single magnetic robot systems.

1) Results in manuscript (page 5-6)

Previous texts

During above-water swimming, the rotation frequency of the CNTY robot was synchronized to the magnetic frequency of the source (Supplementary Fig. 12), which implied that highly agile rectilinear translational swimming could be achieved. The maximum swimming velocity of the designed robotic system was compared to those of living swimmers and miniaturized artificial swimmers in untethered actuation systems (Fig. 1e). The swimming velocity of the CNTY robots could be selected according to the magnetic frequency, particle concentration, and AR. Representatively, the AR-2.5/30-vol.% CNTY robot demonstrated a swimming velocity of up to 212 mm s^{-1} , which was two times that of the water striders. When the body length (BL) was used to normalize these data for enabling comparison with other swimmers, the CNTY robot was found to show a speed of 180 BL s^{-1} , which is the fastest swimming velocity among the artificial swimmers reported to date, to the best of our knowledge. Moreover, the CNTY robot exhibited an above-water swimming velocity that was 47 times that of the magnetic soft robot (spinbot) previously developed by our group, which was actuated by two continuously rotating linearly connected permanent magnets²². The agility of the CNTY robots can be attributed to not only the lightweight yet stiff biomimetic structures but also the synchronized rotation frequency of the CNTY robots with the magnetic frequency of the pulsed electromagnetic field.

Revised texts

During above-water swimming, the rotation frequency of the CNTY robot was synchronized to the magnetic frequency of the source (Supplementary Fig. 16), which implied that highly agile rectilinear translational swimming could be achieved. The **swimming speed** of the designed robotic system was compared to those of living swimmers and **miniaturized artificial magnetic swimmers** (Fig. 1e, Supplementary Table 2 and 3). The **swimming speed** of the CNTY robots could be selected according to the magnetic frequency, particle concentration, and AR. Representatively, the AR-2.5/30-vol.% CNTY robot

demonstrated an average swimming speed of up to 212 mm s^{-1} , which was two times that of the water striders. When the body length (BL) was used to normalize these data for enabling comparison with other swimmers, the CNTY robot was found to show an average swimming speed of 180 BL s^{-1} (Supplementary Fig. 17). Moreover, the ternary-nanocomposite-based CNTY robots exhibited an above-water swimming speed 47 times faster than that of the binary-nanocomposite-based robot (spinbot) previously developed by our group, which was actuated by two continuously rotating linearly connected permanent magnets²⁶. The agility of the CNTY robots can be attributed to not only the stiff yet lightweight biomimetic structures but also the synchronized rotation frequency of the CNTY robots with the magnetic frequency of the pulsed electromagnetic field. In addition to the AR-2.5, AR-6/30-vol.% and AR-13/30-vol.% CNTY robots demonstrated agile above-water swimmability up to 66 BL s^{-1} and 40 BL s^{-1} , respectively.

2) Fig. 1e in manuscript (page 12)

Previous Figure

Revised Figure

3) Supplementary Table 2 and 3 in Supplementary Information (page 3–4)

Previous table

Supplementary Table 2. Swimming velocities of living swimmers and untethered miniaturized artificial swimmers.

Swimmer	Swimming type	Maximum swimming velocity (BL s ⁻¹ ; BL: body length)	Body mass (g)	Supplementary reference	
Living swimmers	Water striders (Gerridae)	Above water	10.0	1.0×10^{-2}	1
	Basilisk lizard (Basiliscus plumifron)	Above water	2.7	20	2
	Yellow-bellied sea snake (Pelamis platura)	Above water	0.75	47	3
	Mallard (Anas platyrhynchos)	Above water	2.2	1160	4
	Copepods (Metridia pacifica)	Underwater	7.0	5.1×10^{-5}	5
	Brown shrimp (Crangon crangon)	Underwater	18.8	2.3	6
	Ghost crab (Ocypode quadrata)	Underwater	11	25	7
	Bottlenose dolphin (Tursiops gilli)	Underwater	4.3	8.9×10^4	8
	Killer whale (Orcinus orca)	Underwater	1.7	1.6×10^6	9
Present study	Above water	180	2.6×10^{-3}	-	
Untethered artificial swimmers	Solvent-driven	Above water	52	7.9×10^{-4}	10
	Solvent-driven	Above water	64	4.1×10^{-2}	11
	Solvent-driven	Above water	10	0.12	12
	Light-driven	Underwater	5.4×10^{-3}	3.2×10^{-2}	13
	Light-driven	Underwater	0.3	4.8×10^{-3}	14
	Light-driven	Underwater	9.2×10^{-2}	1.0×10^{-2}	15
	Battery-driven	Underwater	0.25	16.2	16
	Battery-driven	Underwater	0.51	1.6×10^3	17
	Battery-driven	Underwater	0.45	315	18
	Magnetic-field-driven	Underwater	0.36	1.2×10^{-3}	19
Magnetic-field-driven	Above water	3.8	1.8×10^{-4}	20	
	Underwater	19	10^{-4}		

Revised table

Supplementary Table 2. Swimming speeds of living swimmers

Living swimmer	Swimming type	Body length (mm)	Swimming speed (BL s ⁻¹ ; BL: body length)	Supplementary reference
Water strider (Gerridae)	Above water	10	10.0	1
Mallard (Anas platyrhynchos)	Above water	330	2.2	2
Basilisk lizard (Basiliscus plumifron)	Above water	600	2.7	3
Yellow-bellied sea snake (Pelamis platura)	Above water	705	0.75	4
Ciliate (Paramecium)	Underwater	0.35	5.7	5
Roundworm (Caenorhabditis elegans)	Underwater	1	0.4	6
Copepod (Metridia pacifica)	Underwater	2.5	7.0	7
Brown shrimp (Crangon crangon)	Underwater	40	18.8	8
Ghost crab (Ocypode quadrata)	Underwater	76	11	9
Bottlenose dolphin (Tursiops gilli)	Underwater	1910	4.3	10
Toothed whale (odontocete cetaceans)	Underwater	4550	1.4	11

Supplementary Table 3. Swimming speeds of miniaturized artificial magnetic swimmers in single magnetic robot systems.

Artificial magnetic swimmer	Swimming type	Body length (mm)	Aspect ratio (body length per body width)	Maximum swimming speed (BL s ⁻¹ ; BL: body length)	Supplementary reference
Present study (musculoskeletal system-mimetic robot)	Above water	1.2	2.5	180	-
	Above water	2.8	6	66	-
	Above water	6	13	40	-
Elastically linked ferromagnet	Above water	0.068	1.4	1.4	12

Pentagram-shaped robot	Above water	0.45	1	2.2	13
Helical robot	Above water	0.6	2	4	14
Spermatozoid-like robot	Above water	10	3.3	1.1	15
Janus dimer particle	Underwater	0.01	2	13.3	16
Magnetotactic bacteria-mimetic robot	Underwater	0.05	1	3.2	17
Cuboid-shaped robot	Underwater	0.1	5	2.8	18
Helical robot	Underwater	0.6	2	56	14
Rectangular-shaped robot	Underwater	0.8	2	17	19
Spherical robot	Underwater	0.977	1	7.23	20
Helical robot	Underwater	2	2	0.55	21
Cross-shaped robot	Underwater	7	1	2	22

Newly added references

5. Ghanbari, A. Bioinspired reorientation strategies for application in micro/nanorobotic control. *J. Micro-Bio Robot.* **16**, 173–197 (2020).
6. Rezai, P., Siddiqui, A., Selvaganapathy, P. R. & Gupta, B. P. Electrotaxis of *Caenorhabditis elegans* in a microfluidic environment. *Lab Chip* **10**, 220–226 (2010).
12. Bryan, M. T. *et al.* Microscale magneto-elastic composite swimmers at the air-water and water-solid interfaces under a uniaxial field. *Phys. Rev. Appl.* **11**, 044019 (2019).
13. He, Y. *et al.* Design, analysis and experiments of a magnetic microrobot capable of locomotion and manipulation at water surfaces. *J. Micromechanics Microengineering* **29**, 025010 (2019).
15. Lum, G. Z. *et al.* Shape-programmable magnetic soft matter. *Proc. Natl. Acad. Sci. U. S. A.* **113**, E6007–E6015 (2016).
16. Yu, S. *et al.* Self-propelled janus microdimer swimmers under a rotating magnetic field. *Nanomaterials* **9**, 1672 (2019).
17. Xie, M. *et al.* Bioinspired soft microrobots with precise magneto-collective control for microvascular thrombolysis. *Adv. Mater.* **32**, 2000366 (2020).
18. Ji, S., Li, X., Chen, Q., Lv, P. & Duan, H. Enhanced locomotion of shape morphing microrobots by surface coating. *Adv. Intell. Syst.* **3**, 2000270 (2021).
19. Bi, C., Guix, M., Johnson, B. V., Jing, W. & Cappelleri, D. J. Design of microscale magnetic tumbling robots for locomotion in multiple environments and complex terrains. *Micromachines* **9**, 68 (2018).
20. Chen, W. *et al.* Triple-configurational magnetic robot for targeted drug delivery and sustained release. *ACS Appl. Mater. Interfaces* **13**, 45315–45324 (2021).

21. Hunter, E. E., Brink, E. W., Steager, E. B. & Kumar, V. 3D micromolding of small-scale biological robots. in *2018 International Conference on Manipulation, Automation and Robotics at Small Scales (MARSS)* 1–6 (IEEE, 2018). doi:10.1109/MARSS.2018.8481196
22. Li, C. *et al.* Fast and programmable locomotion of hydrogel-metal hybrids under light and magnetic fields. *Sci. Robot.* **5**, eabb9822 (2020).

Q4) The rectilinear translational swimming of the robot seems to be a non-uniform speed process. Calculation of Reynolds number and characterization on the variation of velocity during the swimming process are needed. Furthermore, does the maximum speed (180 body lengths per second) refer to the average speed of the entire process or the speed of a certain moment?

Response to comment

A4) For the maximum speed in Fig. 1e, we refer to the average swimming speed of the entire process; we calculated the average swimming speed using the total swimming distance of CNTY robots divided by the swimming period. To calculate Reynolds number and characterize the variation of velocity, we measured the instantaneous swimming speed of the CNTY robots during rectilinear translational swimming (newly added Supplementary Fig. 17). The CNTY robot showed fluctuation of instantaneous velocity since the pulsed electromagnetic field induced rectilinear translational motion and rotational motion for the robots simultaneously (Supplementary Fig. 12). For AR-2.5/30 vol%. CNTY robot, maximum instantaneous speed was measured to be 227 BL s^{-1} while the average swimming speed was $180 \pm 24 \text{ BL s}^{-1}$. To prevent the misunderstanding of swimming speed, we corrected the swimming velocity to the average swimming speed of the entire process using the total swimming distance of CNTY robots divided by the swimming period.

1) Supplementary Fig.17 in Supplementary Information (page 21)

Newly added Figure

Supplementary Fig. 17 Speed analysis of rectilinear translational swimming. (a) Overlaid image of the center of mass of AR-2.5/30 vol.% CNTY robot at 6.7 Hz. (b) Corresponding trajectory of (a). The robot was shown in red to distinguish the shadows reflected on water. The blue colors indicate position at the vertex in the trajectory. (c) Instantaneous speed of the CNTY robot. The CNTY robot showed fluctuation of instantaneous velocity since the pulsed electromagnetic field induced rectilinear translational motion and rotational motion for the robots (see Supplementary Fig. 12). Maximum instantaneous speed was 227 BL s^{-1} . We calculated an average swimming speed using the total swimming distance of CNTY robots divided by the swimming period. The average swimming speed of the AR-2.5/30 vol.% CNTY robot was $180 \pm 24 \text{ BL s}^{-1}$.

2) Results in manuscript (page 5-6)

Previous texts

Representatively, the AR-2.5/30-vol.% CNTY robot demonstrated a swimming velocity of up to 212 mm s^{-1} , which was two times that of the water striders. When the body length (BL) was used to normalize these data for enabling comparison with other swimmers, the CNTY robot was found to show a speed of 180 BL s^{-1} , which is the fastest swimming velocity among the artificial swimmers reported to date, to the best of our knowledge.

Revised texts

Representatively, the AR-2.5/30-vol.% CNTY robot demonstrated **an average swimming speed** of up to 212 mm s^{-1} , which was two times that of the water striders. When the body length (BL) was used to normalize these data for enabling comparison with other

swimmers, the CNTY robot was found to show an average swimming speed of 180 BL s^{-1} (Supplementary Fig. 17).

When the diameter of the CNTY robot is supposed to be the characteristic length, Reynolds number for the AR-2.5/30-vol.% CNTY robot at 6.7 Hz ranged between $Re \approx 75$ –140. However, rectilinear translational swimming included both translational motion and rotational motion as shown in Supplementary Fig. 12 and 17, indicating that the characteristic length was continuously changed. Moreover, the calculation of characteristic length could be inaccurate since cylindrical CNTY robot swims at the air-water interface. Due to this experimental difficulty, we could not discuss how dominant inertia drag force is over viscous drag force, thereby we would like not to discuss Reynolds number. Therefore, we have modified the viscous force to be drag force in the manuscript in order to address resistive force more comprehensively against inertia force of swimming CNTY robots.

Q5) In the five (AR: 2.5–6.0) CNTY robots' scenario, the magnetic CNTY robots are disassembled owing to the viscous forces, which is different from the scenario involving the accompanying AR-13 robots. What exactly is the difference in viscous forces between the two scenarios?

Response to comment

A5) In the five CNTY robot's scenario (AR: 2.5–6), the lower AR than 13 increased the number of soft joints as well as decreased the contact area among the robots, thereby, attractive magnetic dipole-dipole interactions decreased among the robots. Consequently, bending stiffness of the magnetically assembled five robots was reduced when compared to that of the assembled robots with ARs of 2.5, 6, and 13. Eventually, drag force became relatively more dominant over magnetic attractive force in soft joints of the CNTY robots, inducing disassembly of the multiple robots. Fluctuational rotational swimming appeared at lower magnetic frequency when compared to the five-robot system.

For example, we compared 20 vol.% CNTY robots at 20 Hz in the three-robot and five robot systems (Figure R1, revised Fig. 3d and e). For the three-robot system with the AR of 2.5, 6, and 13, the CNTY robots were laterally engaged with each other instead of head-to-tail assembly. Because lateral engagement among the robots increased magnetic attractive force, the magnetic modular assembly was preserved at 20 Hz. The magnetically assembled robots overcame drag force and was capable of assembled rotational swimming at the magnetic frequency. Meanwhile, Five robots with the lower AR than 13 not only increased the number of soft joints, but also decreased the contact area. Five-robot system included one AR-3, one AR-4, two AR-5, and one AR-6 CNTY robots. The assembled 20 vol.% CNTY robots were unable to overcome the drag force at 20 Hz, particularly due to the low magnetic attractive force in soft joints, resulting in fluctuating rotational swimming.

The breakage of the magnetic modular assembly is correlated with Mason number, representing the competition between drag force and magnetic attractive force in soft joints (C. E. Sing et al., *Proceedings of the National Academy of Sciences of the United States of America*, **107**, 535-540, 2010) and we also initially considered this analysis. However, assembly and disassembly behaviors of the CNTY robots are more complicated than the colloid particle systems due to following reasons. First, Mason number is derived from Stokes' law which assume spherical geometry of the particle whereas CNTY robots featured high AR cylindrical morphology. In addition, the CNTY robots swam with not only rotational movement but also translational movement whereas the spherical particles were typically considered to have a motion of rotating or unidirectional dragging. To avoid misleading with inaccurate numbers for governing parameters, we would like not to affirm the effects of Mason number in the manuscript specifically.

Due to complex collective swimming behavior of multiple CNTY robots and difficult quantification of cases inducing disassembly, swimming modes were qualitatively classified on the basis of the experimental data according to magnetic frequency and particle concentration. For assembled rectilinear translational swimming, dissipative magnetic interactions in soft joints was confirmed using deconvoluted harmonic sine functions of the trajectory (Supplementary Fig. 20). The assembled rectilinear translational swimming was a result of dominant inertia force and magnetic attractive force over drag force. Assembled rotational swimming appeared when drag force overcame inertia force of robot yet magnetic attractive force remained dominantly than the drag force. The trajectory of CNTY robots was fitted using the single sine waveform of y-axis coordinate without deconvolution of wave functions (Supplementary Fig. 21). For fluctuating rotational swimming, the distance between CNTY robots fluctuated due to the engagement and disengagement (Supplementary Fig. 22), implying that drag force is higher than both inertia force and magnetic attractive force.

- Supplementary Fig. 20, 21, and 22 in Supplementary Information (page 24–26)

Supplementary Fig. 21 Assembled rotational swimming of multiple CNTY robots. Representative assembled rotational swimming of the AR-2.5/30-vol.%, AR-6/30-vol.%, and AR-13/30-vol.% robots at a magnetic frequency of 13.3 Hz. (a) Time-lapse images. (b) y-axis coordinates of the AR-2.5/30-vol.% robot as a function of time. The assembled rotational swimming data were fitted using a sinusoidal waveform of the y-axis.

Supplementary Fig. 22 Fluctuating rotational swimming of multiple CNTY robots. Representative fluctuating rotational swimming of the AR-2.5/10-vol.%, AR-6/10-vol.%, and AR-13/10-vol.% robots at a magnetic frequency of 16.7 Hz. (a) Time-lapse images

showing engagement and disengagement during the fluctuating rotational swimming mode. (b) Trajectory of fluctuating rotational swimming of the AR-2.5/10-vol.% robot fitted using a sinusoidal waveform of the y-axis. (c) Fluctuation of the distance between the AR-2.5/10-vol.% and AR-13/10-vol.% robots that was modulated by engagement and disengagement, as shown in Fig. 3c.

We also have addressed the competition of drag force and magnetic force in soft joints affecting the multimodal collective swimming as follows:

2) Results in manuscript (page 7)

Previous texts

However, the 10 vol.% CNTY robots with low M_s were disassembled even at a lower magnetic frequency of 10 Hz owing to a weak magnetic attractive force. Fluctuating rotational swimming, which was observed instead of assembled rotational swimming (Fig. 3c, Supplementary Fig. 17), featured engagement and disengagement of multiple CNTY robots. The repetition of engagement and disengagement at 16.7 Hz led to a fluctuation in the distance between the AR-2.5/10-vol.% and AR-13/10-vol.% robots.

Revised texts

When drag force became dominant over the inertia force of swimming robots, swimming mode was switched from assembled rectilinear translational swimming to assembled rotational swimming, that is, an increase in magnetic frequency or a decrease in particle concentration. During the assembled rotational swimming of the magnetic modular assembly, the y-axis coordinate of each robot was represented by a single sine waveform (Fig. 3b, Supplementary Fig. 21). The magnetic modular assembly of the 30 vol.% CNTY robots preserved the assembled rotational swimming up to 20 Hz. However, the 10 vol.% CNTY robots with a low M_s disassembled even at a lower magnetic frequency of 10 Hz due to weak magnetic attractive force. Instead of assembled rotational swimming, the 10 vol.% CNTY robots showed fluctuating rotational swimming at 16.7 Hz (Fig. 3c, Supplementary Fig. 22), which featured engagement and disengagement of multiple CNTY robots. The breakage of magnetic modular assembly occurred when drag force overcame magnetic attractive force³⁴ among soft joints. The repetition of engagement and disengagement led to a fluctuation in the distance between the AR-2.5/10-vol.% and AR-13/10-vol.% robots.

Newly added reference

34. Sing, C. E., Schmid, L., Schneider, M. F., Franke, T. & Alexander-Katza, A. Controlled surface-induced flows from the motion of self-assembled colloidal walkers. *Proc. Natl. Acad. Sci. U. S. A.* **107**, 535–540 (2010).

3) Results in manuscript (page 8)

Previous texts

Unlike in the scenario involving the accompanying AR-13 robots, the magnetic modular CNTY robots could be disassembled owing to the viscous forces, reducing the maximum magnetic frequency at which the assembled rectilinear translational swimming or

assembled rotational swimming was permitted.

Revised texts

Fluctuating rotational swimming was observed under the conditions of high magnetic frequency and low particle concentration, where drag force became dominant over magnetic attractive force applied to soft joints of the CNTY robots. Unlike in the scenario involving the accompanying AR-13 robots, the magnetic modular assembly with lower ARs could disassemble due to drag force applied to the soft joints, reducing the maximum magnetic frequency at which the assembled rectilinear translational swimming or assembled rotational swimming was permitted.

Q6) The collective CNTY robots in rectilinear mode can also generate a water vortex like that in rotational mode (although less significant). So, in the demonstration of transporting 830 floating microbeads (Fig. 4b and Supplementary Movie 5), why the rectilinear mode can prevent the microbeads from approaching the entrance?

Response to comment

A6) After the CNTY robots rotated at the vertex of the trajectory, the CNTY robots initiated rectilinear translational swimming, which could physically block the microbeads from entering. We confirmed that the physical blockage by analyzing high-speed camera (newly added Supplementary Fig. 25). Importantly, the agility of rectilinear translational swimming contributed to this physical blockage. Over the course of 1 min, the CNTY robots successfully prevented the influx of floating microbeads.

1) Supplementary Fig.25 in Supplementary Information (page 29)

Newly added Figures

We would like to inform the reviewer that the total number of floating beads was miscalculated in the previous manuscript. In Fig. 4, the number of floating beads has been modified from 830 to 3,350. Previously, the total number of floating microbeads was calculated by measuring the area of a single microbead and the area of floating microbeads from the digital camera images. In that method, the low resolution of the digital image prevented accurate measurement of the area for a single microbead with a small diameter of 255 μm . The overestimated area of a single microbead caused the miscalculation. To determine the number of floating microbeads more precisely, we measured the area of a single particle and the coverage of floating microbeads from the optical microscope images (newly added Supplementary Fig. 24). After converting the microscope image of floating microbeads into a binary image, we measured the average coverage of floating microbeads (black area) per

counted number of floating microbeads. As light reflection and multi-focus issue of spheres hindered the image processing, the center of the microbead areas (white area) was excluded from the calculation of coverage area of microbead.

2) *Supplementary Fig.24 in Supplementary Information (page 28)*

Newly added Figure

Supplementary Fig. 24. Binary image of floating microbeads. The unoccupied space among the floating microbeads was considered when calculating the number of floating microbeads. First, we confirmed the optical microscope image of the floating microbeads. After converting the microscope image of floating microbeads into a binary image, we measured the average coverage of floating microbeads (black area) per counted number of floating microbeads. Black and white areas represented microbeads and non-coverage areas, respectively. As light reflection and multi-focus issue of spheres hindered the image processing, the center of the microbead areas (white area in the particle) was excluded from the calculation of coverage area of microbead. The average coverage of floating microbeads was 85.3%. We could calculate the total number of floating microbeads by measuring the area of a single microbead, the area of clustered floating microbeads, and the average coverage of the floating microbeads with the consideration of the unoccupied space among the microbeads.

3) *Methods in manuscript (page 11)*

Newly added texts

In Fig 4 and 6, the total number of floating microbeads was calculated by measuring the area of a single microbead, the area of clustered floating microbeads, and average coverage of floating microbeads with consideration to the unoccupied space among the microbeads (see Supplementary Fig. 24). The deployed number of CNTY robots for cargo transport was minimized for actuation efficiency.

Q7) In the transportation of semi-submerged cargo (Fig. 5c and Supplementary Movie 8), it seems that the collective robot cannot precisely transport the cargo to a specific point. Dose it because the separation of the robots and cargo is hard to control?

Response to comment

A7) Previously, the CNTY robots were steered by moving the container manually, and cargo transportation was difficult to control delicately. To improve transportation precision, we have devised a motorized stage with two degrees of freedom (2-DoF), coupled with the electromagnetic coils (newly added Supplementary Fig. 29-a). By precisely manipulating the location of motorized stage, the multiple CNTY robots successfully demonstrated transportation of a spherical cargo with rolling resistance (newly added Supplementary Fig. 29-b) as well as a cuboid cargo with sliding resistance (newly added Supplementary Fig. 30) to the desired location (Supplementary Movie 9). Furthermore, we could collect thousands of floating microplastics in an open space, as shown in Fig. 6 and Supplementary Movie 10.

1) Results in manuscript (page 9-10)

Newly added texts

To improve transportation precision of various cargos (Supplementary Movie 9), we devised a motorized stage having two degrees of freedom (2-DoF)³⁶ coupled with the electromagnetic coils. Through manipulating the location of the motorized stage instead of the water container, the spherical cargo with rolling resistance could be delivered with more delicacy to its designated position (Supplementary Fig. 29). The transportation capability was demonstrated even for semi-submerged or underwater cargos with sliding resistance by above-water collective swimming or underwater collective swimming, respectively (Supplementary Fig. 30).

Furthermore, seven 30 vol.% CNTY robots accomplished the collection of thousands of microplastics in an open space by generating a large-magnitude vortex (Fig. 6, Supplementary Movie 10). When the CNTY robots were adjacent to a cluster of floating microplastics, the microplastics became ensnared in the CW vortex. (Fig. 6a). Under manipulation of the motorized stage, CNTY robots transported and merged the scattered clusters of floating polyethylene microbeads ($D= 850 \mu\text{m}$) using assembled rotational swimming (Fig. 6b, c). A total number of 4,630 microplastics was successfully collected within 150 s.

Newly added references

36. Kim, Y. *et al.* Effects of helix geometry on magnetic guiding of helical polymer composites on a gastric cancer model: a feasibility study. *Materials*. **13**, 1014 (2020).

2) Supplementary Fig.29 in Supplementary Information (page 33)

Newly added Figure

Supplementary Fig. 29 Transportation of semi-submerged spherical cargo by manipulating the motorized stage. (a) Motorized stage having two degrees of freedom (2-DoF) coupled with the electromagnetic coils. (b) Delicate positional control of seven 30 vol.% CNTY robots in order to transport the spherical cargo by manipulating the location of the motorized stage. Diameter and weight of the cargo was 4 mm and 52 mg, respectively. The magnetic frequency was 13.3 Hz. Seven 30 vol.% CNTY robots were deployed.

3) *Supplementary Fig.30 in Supplementary Information (page 34)*

Newly added Figure

4) *Fig.6 in manuscript (page 17)*

Newly added Figures

Q8-1) In the transportation of both floating microbeads and semi-submerged cargo, the position of the collective robots is changed by moving the water container. It seems that the general way to change the relative position of the rotational robots is moving the entire surrounding environment. However, in many scenarios, it is difficult to move the environments. How to solve this problem?

Response to comment

A8) Thank you for the constructive comment. Although the relative position of the robots was manually controlled, it was previously intended to propose the proof-of-concept of cargo transport via collective swimming of multiple CNTY robots. However, we agree with the reviewer's point. For entirely contactless steering of the robots and improvement of transportation precision, we have devised a motorized stage with two degrees of freedom (2-DoF), coupled with the electromagnetic coils. We have addressed cargo transportation using the motorized stage in the response to comment (Q7) and newly added Fig. 6.

Q8-2) Moreover, how can this robotic system be used for drug delivery?

Response to comment

A8) In abstract and outlook, drug delivery by the polymer-composited-based robots was mentioned for potential applications (Figure R2 and R3).

Figure R2. Demonstration of drug transport by the multilegged soft robot in a stomach model under wet environment. (H. Lu et al, *Nature Communications*, **9**, 3944, 2018)

Figure R3. Carrying the oblong pharmaceutical pill by rolling motion of the soft robot. (Y. Kim et al, *Nature*, **558**, 274–279, 2019)

The CNTY robots enabled to transport various cargos by above-water swimming as well as underwater swimming (newly added Supplementary Fig. 30 and Supplementary Movie. 9), showing the potential for transportation of pharmaceuticals. The underwater swimming was proceeded after the robots were intentionally submerged in water. According to the reviewer's concerns, drug delivery can be described as referring to therapeutic agents (H. Lee et al., *ACS Applied Materials and Interfaces*, **13**, 19633–19647, 2021; T.-Y. Huang et al., *Advanced Materials*, **27**, 6644–6650, 2015). To avoid misunderstandings with respect to loading of the therapeutic agents, 'drug delivery' was changed to 'transportation of pharmaceuticals' as follows:

1) Supplementary Fig.30 in Supplementary Information (page 34)

Newly added Figure

Supplementary Fig. 30 Transportation of cargos with sliding resistance by above-water swimming and underwater swimming. Multiple 30 vol.% CNTY robots were manipulated by moving the electromagnetic coils coupled with the motorized stage having two degrees of freedom (2-DoF). (a) Transportation of semi-submerged cuboid cargo with a weight of 42 mg by above-water swimming of 19 CNTY robots. The increased number of CNTY robots

resulted in a larger magnitude of vortex due to the larger hydrodynamic volume of the magnetic modular assembly. Therefore, the 19 CNTY robots with assembled rotational swimming transported cuboid cargo with sliding resistance. (b) Transportation of submerged asymmetric cargo with a weight of 39 mg by underwater swimming of two CNTY robots. The underwater swimming was proceeded after the robots were intentionally submerged in water. The magnetic frequency was 8.3 Hz in (a) and (b).

2) Abstract in manuscript (page 1)

Previous texts

The collective actuation of biomimetic nanocomposite robots is anticipated to provide practical robotic applications for microplastic removal^{39,40}, drug delivery⁴¹, and vortex control in microfluidic platforms⁴².

Revised texts

The controllable collective actuation of these biomimetic nanocomposite robots can lead to versatile robotic functions, including microplastic removal, microfluidic vortex control, and transportation of pharmaceuticals.

3) Outlook in manuscript (page 10)

Previous texts

The collective actuation of biomimetic nanocomposite robots is anticipated to provide practical robotic applications for microplastic removal^{39,40}, drug delivery⁴¹, and vortex control in microfluidic platforms⁴².

Revised texts

The collective actuation of biomimetic nanocomposite robots is anticipated to provide versatile robotic applications for microplastic removal^{45,46}, vortex control in microfluidic platforms⁴⁷, and transportation of pharmaceuticals^{48,49}.

Newly added references

48. Lu, H. *et al.* A bioinspired multilegged soft millirobot that functions in both dry and wet conditions. *Nat. Commun.* **9**, 3944 (2018).
49. Kim, Y., Yuk, H., Zhao, R., Chester, S. A. & Zhao, X. Printing ferromagnetic domains for untethered fast-transforming soft materials. *Nature* **558**, 274–279 (2018).

Q9) What does the triangle symbol in Fig. 2e mean? Should it probably be a square symbol?

Response to comment

A9) We thank the reviewer for the detailed comment. We have corrected the triangle symbol to the square symbol indicating rectilinear translational swimming (AR-21.5/30-vol.% at 3.3 Hz)

in Fig. 2d.

1) Fig. 2d in manuscript (page 13)

Previous Figure

Revised Figure

Q10-1) The color and symbols are not matched in Fig. 3e, which is easy to cause confusion. Please correct it.

Response to comment

A10-1) We thank the reviewer for the considerate comment. The circular symbols were revised to be diamond symbols of fluctuating rotational swimming in Fig. 3 e.

1) Fig. 3e in manuscript (page 14)

Previous Figure

Revised Figure

Q10-2) The second sentence of Supplementary Table 2 (“The vol.% notation ...”) should belong to Supplementary Table 1.

Response to comment

A10-2) We thank the reviewer for the detailed comment. We revised the position of the sentence belong to Supplementary Table 1.

1) Legend of Supplementary Table 1 and 2 in Supplementary Information (page 2–3)

Previous texts

Supplementary Table 1. Dimensions, body masses, and saturation magnetization (M_s) values of CNTY robots.

Supplementary Table 2. Swimming velocities of living swimmers and untethered miniaturized artificial swimmers. The vol.% notation refers to the concentration of iron particles dispersed in the PDMS prepolymer.

Revised texts

Supplementary Table 1. Dimensions, body length, body masses, and saturation magnetization (M_s) values of CNTY robots. The vol.% notation refers to the concentration of iron particles dispersed in the PDMS prepolymer.

Reviewer #2

Summary:

The paper describes a methodology of targeted drug delivery where superparamagnetic beads are aerosolized and deployed using a nebulizer. Once released using a nebulizer, the authors apply rotational magnetic fields to induce rolling motion of the bead agglomerates formed in an aqueous film, described as microwheels, which can translate to a desired spot. As an application, the authors show a 3-D printed lung model whereby they attempt targeted delivery of these beads to branches using combined application of a permanent magnet (to create a bolus), and thereon rotational magnetic field (for subsequent localization).

Response to comment

We believe that there was some confusion. We did not utilize aerosolized superparamagnetic bead and the nebulizer. Also, we did not demonstrate a 3D-printed lung model in any Figures in this manuscript. Our polymer-composite-based robots were prepared by dip-coating of nanoporous CNTY framework in a mixture of thermo-curable prepolymer and ferromagnetic iron microparticles. Furthermore, we did not show a permanent magnet. Pulsed electromagnetic field was induced in order to generate rotational magnetic field. Under the control of the electromagnetic field, the CNTY robots performed vortex-induced transportation of floating and semi-submerged cargos. We would like to request further review to ensure that the reviewer's summary comment was properly mentioned.

General comments:

The manuscript describes CNTY robots, nanocomposite robots that mimic the musculoskeletal system, and hold the ability for multimodal collective motion and manipulation at air-water interface. These robots comprise of light and stiff network of MWCNTs and are coated with a magnetic microparticles using PDMS as a binder material. Further, the motion of these CNTY robots is demonstrated for a single robot and multi-robot systems. Finally, some applications are presented where collective motion of CNTY robots are used for cargo manipulation. Overall, the manuscript is well written and has an extensive description of fabrication and characterization of CNTY robots.

Response to comment

We thank the reviewer for the positive evaluation on our manuscript.

Q1) However, there are several concerns regarding the swimming motion of these robots, particularly with regards to collective motion of multiple such robots. Moreover, while many applications for targeted manipulation are presented, the inability of these CNTY robots to move underwater does not justify the proposed application of targeted drug delivery as described by the authors.

Response to comment

Although we are not sure which several concerns the reviewer specifically mentioned with regards to collective motion of multiple such robots, we have discussed (1) assembly and disassembly of multiple CNTY robots and (2) cargo transport in divided spaces and open spaces by responding to the reviewer's comments (Q8, Q9, and Q10). However, we agree with the reviewer's point regarding the underwater drug delivery. We newly demonstrated transportation capability of a submerged cargo (newly added Supplementary Fig. 30-b and Supplementary Movie. 9). Collective swimming of multiple CNTY robots enabled semi-submerged as well as underwater cargos with sliding resistance to transport by above-water swimming or underwater swimming, respectively, showing the potential for transportation of pharmaceuticals (Figure R1 and R2). The underwater swimming was proceeded after the robots were intentionally submerged in water.

Additionally, according to the reviewer's concerns, drug delivery can be described as referring to therapeutic agents (H. Lee et al., *ACS Applied Materials and Interfaces*, **13**, 19633–19647, 2021; T.-Y. Huang et al., *Advanced Materials*, **27**, 6644–6650, 2015). To avoid misunderstandings with respect to loading of the therapeutic agents, 'drug delivery' was changed to 'transportation of pharmaceuticals'.

1) Supplementary Fig.30 in Supplementary Information (page 34)

Newly added Figure

Supplementary Fig. 30 Transportation of cargos with sliding resistance by above-water swimming and underwater swimming. Multiple 30 vol.% CNTY robots were manipulated by moving the electromagnetic coils coupled with the motorized stage having two degrees

of freedom (2-DoF). (a) Transportation of semi-submerged cuboid cargo with a weight of 42 mg by above-water swimming of 19 CNTY robots. The increased number of CNTY robots resulted in a larger magnitude of vortex due to the larger hydrodynamic volume of the magnetic modular assembly. Therefore, the 19 CNTY robots with assembled rotational swimming transported cuboid cargo with sliding resistance. (b) Transportation of submerged asymmetric cargo with a weight of 39 mg by underwater swimming of two CNTY robots. The underwater swimming was proceeded after the robots were intentionally submerged in water. The magnetic frequency was 8.3 Hz in (a) and (b).

Figure R1. Demonstration of drug transport by the multilegged soft robot in a stomach model under wet environment. (H. Lu et al, *Nature Communications*, **9**, 3944, 2018)

Figure R2. Carrying the oblong pharmaceutical pill by rolling motion of the soft robot. (Y. Kim et al, *Nature*, **558**, 274–279, 2019)

2) Abstract in manuscript (page 2)

Previous texts

The collective actuation of biomimetic nanocomposite robots is anticipated to provide practical robotic applications for microplastic removal^{39,40}, drug delivery⁴¹, and vortex control in microfluidic platforms⁴².

Revised texts

The controllable collective actuation of these biomimetic nanocomposite robots can lead to versatile robotic functions, including microplastic removal, microfluidic vortex control, and transportation of pharmaceuticals.

3) Outlook in manuscript (page 9)

Previous texts

The collective actuation of biomimetic nanocomposite robots is anticipated to provide practical robotic applications for microplastic removal^{39,40}, drug delivery⁴¹, and vortex control in microfluidic platforms⁴².

Revised texts

The collective actuation of biomimetic nanocomposite robots is anticipated to provide versatile robotic applications for microplastic removal^{40,41}, vortex control in microfluidic platforms⁴², and transportation of pharmaceuticals^{43,44}.

Newly added references

43. Lu, H. *et al.* A bioinspired multilegged soft millirobot that functions in both dry and wet conditions. *Nat. Commun.* **9**, 3944 (2018).

44. Kim, Y., Yuk, H., Zhao, R., Chester, S. A. & Zhao, X. Printing ferromagnetic domains for untethered fast-transforming soft materials. *Nature* **558**, 274–279 (2018).

These above mentioned considerations have been summarized below as technical (scientific concerns) and detailed (addresses readability) comments.

Technical comments:

Q2) Overall, my main criticism of CNTY robots is that they could only operate on the air-water interface. How do you address applications such as drug delivery as cited in the abstract/outlook? A majority of drug delivery applications require therapeutic agents to be submerged in water.

Response to comment

A2) We addressed the underwater swimmability of CNTY robots and drug delivery application in abstract and outlook according to the response to comment (Q1). The magnetic modular assembly of high-AR CNTY robots could transport floating, semi-submerged, and underwater cargos. However, we would like to emphasize that the main focus of this manuscript was agile above-water swimming of stiff yet lightweight CNTY robots in order to generate a water vortex and transport multiple cargos via the vortex. As shown in revised Supplementary Table 3, enhancement of swimming speed is challenged for the miniaturized magnetic robots. The previous magnetic robots exhibited only a few body lengths per second ($0.55\sim 17 \text{ BL s}^{-1}$) in the single robot systems and their aspect ratios were ranged from 1 to 5. Swimming speed of AR-2.5, AR-6, and AR-13 CNTY robots reached 180, 66, and 40 BL s^{-1} , respectively. We analyzed the swimming speed for single magnetic robot systems. The body mass was revised to be body

length since the mass information rarely reported as well as we would like to note the agile swimmability of CNTY robots despite high aspect ratios.

In designing the magnetic robot capable of vortex generation, a lightweight robot body was needed to impart above-water swimmability and agility, yet a stiff body frame was also required to overcome drag force and swim. Although robots with high Young's modulus can result in considerable vortex generation, increase in Young's modulus indicates that high density materials are necessary according to the well-known Ashby plots. We conceived a magnetic robot with high aspect ratio (AR) and their magnetic assembly as a more effective strategy for generating the vortex. As the magnetic assembly of high-AR robots had a large hydrodynamic volume, rotational swimming of the magnetic assembly was expected to exert high rotational force and induce a large-magnitude vortex. Hence, the magnetic assembly of the multiple robots required larger magnetic attractive force among robots than drag force to preserve the assembled structure even during swimming rapidly. In order to enhance the magnetic attractive force, we planned to bind magnetic particles on the robot surface instead of embedding the particles inside the robot since magnetic force between the particles drastically increases by decreasing the interparticle distance (A. Mehdizadeh et al., *Acta Mechanica Sinica*, **26**, 921–929, 2010).

To meet the requirements, we introduced a biomimetic hierarchical nano/microstructure using nanoporous CNTY framework, PDMS, and magnetic microparticles, which emulate the spongy bone, connective tissue, and skeletal muscle, respectively. First, stiff yet lightweight nano-framework was prepared by synthesizing multi-walled carbon nanotube (MWNT), followed by twisting the MWNT to form the nanoporous CNTY framework (newly added Supplementary Fig. 1, 2, and 3). Then, dip-coating process was implemented using a mixture of PDMS prepolymer and magnetic microparticles to bind highly concentrated magnetic particles on the robot surface. The nanoporous CNTY framework was immersed in the mixture of PDMS prepolymer and magnetic microparticle, followed by drainage of excessive mixture. Because the high concentration of magnetic particles could cause a drastic increase in the viscosity of the composite mixture, we handled the composite mixture by utilizing the prepolymer. Moreover, during the dip-coating process, the nanoporous CNTY framework operated as a membrane to permeate the liquid-phase PDMS prepolymer through capillary forces, whereas the 5- μm -sized magnetic microparticles were excluded from the core region. The magnetic microparticles at the CNTY surface were bound through thermal curing of PDMS via hydrosilylation polymerization. The PDMS acted as a binder for the magnetic particles in a fashion similar to the deep fascia which binds muscle fibers. Absence of the PDMS binder could hinder the adhesion of magnetic particles onto the CNTY framework due to the weak cohesive energy between the magnetic particles. We also could varied the coating thickness of the magnetic composite layer by changing the concentration of magnetic particles (Supplementary Fig. 8).

As a result, the biomimetic structure of CNTY robot contributed to improvement of above-water swimming velocity. The AR-2.5/30-vol.% robot accomplished swimming velocity of 180 BL s^{-1} . We previously reported polymer-composite-based robot with AR-2, showing only above-water swimming velocity of 3.8 BL s^{-1} (S. Won et al., *Nature Communications*, **10**, 4751, 2019). The low swimming velocity and low-AR could not generate vortex. Maximum AR was 4 for magnetic actuation of the polymer-composite-based robots. The magnetic modular

assembly of the high-AR CNTY robots demonstrated on-demand cargo transportation of thousands of microbeads and a semi-submerged millimeter-scale bead via the vortex generation.

As it seems that our intended design principle and significance of work were not fully delivered from the previous manuscript, we extensively revised the manuscript as follows:

1) Supplementary Table 2 and 3 in Supplementary Information (page 3–4)

Previous table

Supplementary Table 2. Swimming velocities of living swimmers and untethered miniaturized artificial swimmers.

Swimmer	Swimming type	Maximum swimming velocity (BL s ⁻¹ ; BL: body length)	Body mass (g)	Supplementary reference
Water striders (Gerridae)	Above water	10.0	1.0 × 10 ⁻²	1
Basilisk lizard (Basiliscus plumifron)	Above water	2.7	20	2
Yellow-bellied sea snake (Pelamis platura)	Above water	0.75	47	3
Mallard (Anas platyrhynchos)	Above water	2.2	1160	4
Living swimmers Copepods (Metridia pacifica)	Underwater	7.0	5.1 × 10 ⁻⁵	5
Brown shrimp (Crangon crangon)	Underwater	18.8	2.3	6
Ghost crab (Ocypode quadrata)	Underwater	11	25	7
Bottlenose dolphin (Tursiops gilli)	Underwater	4.3	8.9 × 10 ⁴	8
Killer whale (Orcinus orca)	Underwater	1.7	1.6 × 10 ⁶	9
Present study	Above water	180	2.6 × 10⁻³	-
Solvent-driven	Above water	52	7.9 × 10 ⁻⁴	10
Untethered artificial swimmers Solvent-driven	Above water	64	4.1 × 10 ⁻²	11
Solvent-driven	Above water	10	0.12	12
Light-driven	Underwater	5.4 × 10 ⁻³	3.2 × 10 ⁻²	13
Light-driven	Underwater	0.3	4.8 ×	14

				10^{-3}	
Light-driven	Underwater	9.2×10^{-2}	1.0×10^{-2}	15	
Battery-driven	Underwater	0.25	16.2	16	
Battery-driven	Underwater	0.51	1.6×10^3	17	
Battery-driven	Underwater	0.45	315	18	
Magnetic-field-driven	Underwater	0.36	1.2×10^{-3}	19	
Magnetic-field-driven	Above water	3.8	1.8×10^{-4}	20	
	Underwater	19			

Revised table

Supplementary Table 2. Swimming speeds of living swimmers

Living swimmer	Swimming type	Body length (mm)	Swimming speed (BL s^{-1} ; BL: body length)	Supplementary reference
Water strider (Gerridae)	Above water	10	10.0	1
Mallard (Anas platyrhynchos)	Above water	330	2.2	2
Basilisk lizard (Basiliscus plumifron)	Above water	600	2.7	3
Yellow-bellied sea snake (Pelamis platura)	Above water	705	0.75	4
Ciliate (Paramecium)	Underwater	0.35	5.7	5
Roundworm (Caenorhabditis elegans)	Underwater	1	0.4	6
Copepod (Metridia pacifica)	Underwater	2.5	7.0	7
Brown shrimp (Crangon crangon)	Underwater	40	18.8	8
Ghost crab (Ocyrode quadrata)	Underwater	76	11	9
Bottlenose dolphin (Tursiops gilli)	Underwater	1910	4.3	10
Toothed whale (odontocete cetaceans)	Underwater	4550	1.4	11

Supplementary Table 3. Swimming speeds of miniaturized artificial magnetic

swimmers in single magnetic robot systems.

Artificial magnetic swimmer	Swimming type	Body length (mm)	Aspect ratio (body length per body width)	Maximum swimming speed (BL s ⁻¹ ; BL: body length)	Supplementary reference
Present study (musculoskeletal system-mimetic robot)	Above water	1.2	2.5	180	-
	Above water	2.8	6	66	-
	Above water	6	13	40	-
Elastically linked ferromagnet	Above water	0.068	1.4	1.4	12
Pentagram-shaped robot	Above water	0.45	1	2.2	13
Helical robot	Above water	0.6	2	4	14
Spermatozoid-like robot	Above water	10	3.3	1.1	15
Janus dimer particle	Underwater	0.01	2	13.3	16
Magnetotactic bacteria-mimetic robot	Underwater	0.05	1	3.2	17
Cuboid-shaped robot	Underwater	0.1	5	2.8	18
Helical robot	Underwater	0.6	2	56	14
Rectangular-shaped robot	Underwater	0.8	2	17	19
Spherical robot	Underwater	0.977	1	7.23	20
Helical robot	Underwater	2	2	0.55	21
Cross-shaped robot	Underwater	7	1	2	22

Newly added references

5. Ghanbari, A. Bioinspired reorientation strategies for application in micro/nanorobotic control. *J. Micro-Bio Robot.* **16**, 173–197 (2020).
6. Rezai, P., Siddiqui, A., Selvaganapathy, P. R. & Gupta, B. P. Electrotaxis of *Caenorhabditis elegans* in a microfluidic environment. *Lab Chip* **10**, 220–226 (2010).
12. Bryan, M. T. *et al.* Microscale magneto-elastic composite swimmers at the air-water and water-solid interfaces under a uniaxial field. *Phys. Rev. Appl.* **11**, 044019 (2019).
13. He, Y. *et al.* Design, analysis and experiments of a magnetic microrobot capable of locomotion and manipulation at water surfaces. *J. Micromechanics Microengineering* **29**,

025010 (2019).

15. Lum, G. Z. *et al.* Shape-programmable magnetic soft matter. *Proc. Natl. Acad. Sci. U. S. A.* **113**, E6007–E6015 (2016).

16. Yu, S. *et al.* Self-propelled janus microdimer swimmers under a rotating magnetic field. *Nanomaterials* **9**, 1672 (2019).

17. Xie, M. *et al.* Bioinspired soft microrobots with precise magneto-collective control for microvascular thrombolysis. *Adv. Mater.* **32**, 2000366 (2020).

18. Ji, S., Li, X., Chen, Q., Lv, P. & Duan, H. Enhanced locomotion of shape morphing microrobots by surface coating. *Adv. Intell. Syst.* **3**, 2000270 (2021).

19. Bi, C., Guix, M., Johnson, B. V., Jing, W. & Cappelleri, D. J. Design of microscale magnetic tumbling robots for locomotion in multiple environments and complex terrains. *Micromachines* **9**, 68 (2018).

20. Chen, W. *et al.* Triple-configurational magnetic robot for targeted drug delivery and sustained release. *ACS Appl. Mater. Interfaces* **13**, 45315–45324 (2021).

21. Hunter, E. E., Brink, E. W., Steager, E. B. & Kumar, V. 3D micromolding of small-scale biological robots. in *2018 International Conference on Manipulation, Automation and Robotics at Small Scales (MARSS)* 1–6 (IEEE, 2018). doi:10.1109/MARSS.2018.8481196

22. Li, C. *et al.* Fast and programmable locomotion of hydrogel-metal hybrids under light and magnetic fields. *Sci. Robot.* **5**, eabb9822 (2020).

2) Introduction in manuscript (page 2)

Previous texts

Bioinspired polymeric structures are anticipated to improve the actuation efficiencies of miniaturized robots while minimizing the number of robots required. The locomotion of aquatic and terrestrial living creatures is a noteworthy source for realizing miniaturized polymer-composite-based robots^{16,17}. Appropriate robotic architectures have been constructed using photo-curable^{18,19}, thermo-curable^{20,21}, and thermo-processable²² polymers. Moreover, magnetic materials have been randomly dispersed²³, anisotropically aligned²⁴, or selectively deposited²⁵ in these polymeric structures to realize magnetic actuation. However, studies on polymer-composite-based magnetic robots have primarily described the independent locomotion of a single robot. Controllable collective actuation of multiple polymer-composite-based robots is required to achieve functionalities beyond those offered by single-robot systems.

Revised texts

Geometric changes in miniaturized robots are anticipated to improve actuation efficiencies of the robots for cargo transportation while minimizing the number of robots required. Polymeric robots are particularly promising since diverse structures, composed of multi-material, can be constructed through polymer processing techniques. For example, polymer-composite-based robots can be fabricated into meshes¹⁹, needles²⁰, jellyfishes²¹, cilia²², and helices²³ by patterning, printing, molding, or thermal fixation of photo-curable, thermo-curable, and thermo-processible polymers. A single polymeric robot in capsule²⁴ or cross²⁵ shape could transport a miniscule cargo to arbitrary destinations. To control these

polymeric robots magnetically, a magnetic metal-layer was deposited on the capsule-shaped photoresist²⁴, and magnetic nanowires were embedded in the cross-shaped hydrogel with anisotropic alignment²⁵. However, cargo transportation has primarily focused on a single cargo smaller than the body length of polymeric robots. Recently, tens of robots cooperated to carry an object that was larger and heavier than the individual robots²⁶, yet further challenges remain to deliver multiple cargos rapidly and simultaneously. Controllable collective actuation of multiple polymer-composite-based robots is required to achieve functionalities beyond those offered by single-robot systems.

Newly added references

19. Nguyen, K. T. *et al.* A Magnetically guided self-rolled microrobot for targeted drug delivery, real-time x-ray imaging, and microrobot retrieval. *Adv. Healthc. Mater.* **10**, 2001681 (2021).
20. Lee, S. *et al.* A needle-type microrobot for targeted drug delivery by affixing to a microtissue. *Adv. Healthc. Mater.* **9**, 1901697 (2020).
21. Ren, Z., Hu, W., Dong, X. & Sitti, M. Multi-functional soft-bodied jellyfish-like swimming. *Nat. Commun.* **10**, 2703 (2019).
22. Gu, H. *et al.* Magnetic cilia carpets with programmable metachronal waves. *Nat. Commun.* **11**, 2637 (2020).
23. Won, S., Je, H., Kim, S. & Wie, J. J. Agile underwater swimming of magnetic polymeric microrobots in viscous solutions. *Adv. Intell. Syst.* **4**, 2100269 (2022).

3) Results in manuscript (page 3-4)

Previous texts

Biomimetic hierarchical nano/microstructures were prepared using a nanoporous carbon nanotube yarn (CNTY) framework, polydimethylsiloxane (PDMS), and magnetic microparticles, which emulate the spongy bone, connective tissue, and skeletal muscle, respectively, in brachial anatomy (Fig. 1a and b-(i)). The human arm exerts a strong force in proportion to the thickness of muscle bundles by means of the skeletal muscle surrounding lightweight yet stiff bones. The spongy bone component of the long compact bones articulated in the elbow comprises hierarchically arranged collagen fibers and porous networks, and remains lightweight despite containing a packed bone marrow. Carbon nanotubes (CNTs) with an exceptional stiffness (Young's modulus ~ 1 TPa) and low density²⁶⁻²⁸ were twisted to construct the spongy-bone-mimicking CNTY framework. One hundred strands of 7-nm-thick CNTs were twisted to prepare a 340- μ m-thick nanoporous CNTY framework (Fig. 1b-(ii) and (iii)).

Revised texts

For a magnetic robot to be capable of vortex generation at the air-water interface as well as cargo transportation, a lightweight robot body was essential to impart agile above-water swimmability. A stiff body frame was also required to overcome surface tension and drag force in swimming even though the two variables of lightweightness and rigidity are considered trade-offs according to Ashby plots²⁷ showing Young's modulus versus density.

We further conceived a magnetic robot with high aspect ratio (AR) and collective actuation of the high-AR robots as a more effective strategy for generating the vortex. When the high-AR robots assembled magnetically to have large hydrodynamic volume, rotational swimming of the magnetically assembled robots was expected to exert high rotational force and induce a large-magnitude vortex. Magnetic assembly of the multiple robots required larger magnetic attractive force among robots than drag force to preserve the assembled structure even through rapid swimming. To enhance inter-robot magnetic attractions, we considered a core-shell-like robot by binding highly concentrated magnetic particles only on the robot surface, instead of embedding particles inside the robot since the magnetic force between particles drastically increases by decreasing interparticle distances^{28,29}. The robot design that met these requirements was inspired by brachial anatomy (Fig. 1a, b-(i)). Movements of the human arm are produced by forces generated from skeletal muscles surrounding the stiff long bones. Along the interior toward the end of the long bones, the spongy bone shows a hierarchical composition of collagen fiber bundles followed by a porous network while the bone marrow fills in the bone. The porosity of spongy bone offers lightness to the long bones while surrounding skeletal muscles are bounded by the connective tissue known as deep fascia. We introduced a biomimetic hierarchical nano/microstructure with nanoporous carbon nanotube yarn (CNTY) framework, polydimethylsiloxane (PDMS), and magnetic microparticles, emulating the spongy bone, connective tissue, and skeletal muscle, respectively. First, the spongy-bone-mimicking nano-framework was prepared using multi-walled carbon nanotube (MWNT) fibers synthesized through floating catalyst chemical vapor deposition (Supplementary Fig. 1). The synthesized 36- μm -thick MWNT fiber was composed of ten-walled 14 nm-thick MWNTs. The MWNT fiber provided specific strength of 1.19 N tex^{-1} (equivalent to $1.19 \text{ GPa g}^{-1} \text{ cm}^3$) and specific modulus of 46.9 N tex^{-1} (Supplementary Fig. 2). For lightweight nanoporous structure, high densification was intentionally restrained in the MWNT fiber, confirmed by Raman spectra (Supplementary Fig. 3, Fig. 1b-(ii), (iii)). To build the stiff yet lightweight nano-framework, 100 MWNT fibers were twisted to form the 340- μm -thick CNTY (Supplementary Fig. 4).

Then, a dip-coating process was implemented using a mixture of PDMS prepolymer and iron microparticles to bind highly concentrated magnetic particles on the robot surface. The nanoporous CNTY framework was immersed in the mixture of PDMS prepolymer and magnetic microparticles, followed by drainage of excessive mixture (Supplementary Fig. 5). During the dip-coating process, the nanoporous CNTY framework operated as a membrane to permeate the liquid-phase PDMS prepolymer through capillary forces³⁰, whereas the 5- μm -sized iron microparticles³¹ were excluded from the core region, as shown in Fig. 1b-(i). The magnetic microparticles at the CNTY surface were bound through thermal curing of PDMS via hydrosilylation polymerization. The PDMS acted as a binder for the magnetic particles in a fashion similar to the deep fascia and mimicked the binding of muscle fibers. Absence of the PDMS binder could hinder the adhesion of magnetic particles onto the CNTY framework due to the weak cohesive energy between the magnetic particles. By spatially selective coating of the magnetic microparticles onto the surface of the nanoporous CNTY framework, the coating thickness of the magnetic composite layer could be varied with the concentration of magnetic particles dispersed in the PDMS prepolymer (Supplementary Fig. 6, 7, 8).

Q3) Regarding the proposed application to manipulate semi-submerged objects, can authors propose a method where such CNTY robots can be made less buoyant so that they submerge in water instead? Please justify.

Response to comment

A3) The underwater swimming of CNTY robots and transportation of semi-submerged cargo could be demonstrated after the robots were intentionally submerged in water, which is addressed in response to comment Q1.

Q4) The Introductory paragraphs describes the state-of-the-art well and in a succinct manner albeit has scope for some more clarifications:

Q4-1) “Reconfigurable swarms of magnetic microparticles have been recently realized by varying the axes and frequencies of an alternating magnetic field”, motion differentiation in microrobots is also observed for different field precession, reference below:

Mohanty, S., Jin, Q., Furtado, G. P., Ghosh, A., Pahapale, G., Khalil, I. S., ... & Misra, S. (2020). Bidirectional Propulsion of Arc-Shaped Microswimmers Driven by Precessing Magnetic Fields. *Advanced Intelligent Systems*, 2(9), 2000064.

Response to comment

A3-1) We thank the reviewer for the positive and constructive comment. We have revised Introduction to describe various external magnetic fields for actuating microrobots as follows:

1) Introduction in manuscript (page 2)

Previous texts

A time-varying magnetic field can enable programmable magnetomotility of microparticles through magnetic torque occurring in a magnetic body owing to the mismatched directions between the magnetization of the body and the applied magnetic field. Reconfigurable swarms of magnetic microparticles have been recently realized by varying the axes and frequencies of an alternating magnetic field^{13,14}. However, most studies on collective behavior have focused on colloidal particle systems¹⁵. Although particle swarms demonstrate micro-cargo transportation capabilities, numerous robots are required to achieve them; moreover, the long-distance transport of multitudinous cargos in a short duration is difficult.

Revised texts

Time-varying magnetic fields^{13–15} including rotating, oscillating, and precessing offer programmable magnetomotility of microrobots through magnetic torque occurring in a magnetic body. Reconfigurable swarms of magnetic microparticles have been realized by varying the axes and frequencies of the alternating magnetic fields^{17,18}. However, most studies on collective behavior have focused on colloidal particle systems¹⁹. Although particle swarms demonstrate micro-cargo transportation capabilities, numerous robots are required to achieve them; moreover, the long-distance transport of multitudinous cargos in a short

duration is difficult.

Newly added references

13. Yu, J. *et al.* Active generation and magnetic actuation of microrobotic swarms in bio-fluids. *Nat. Commun.* **10**, 5631 (2019).
14. Mohanty, S. *et al.* Bidirectional propulsion of arc-shaped microswimmers driven by precessing magnetic fields. *Adv. Intell. Syst.* **2**, 2000064 (2020).
15. Zhang, H. *et al.* Dual-responsive biohybrid neutrobots for active target delivery. *Sci. Robot.* **6**, eaaz9519 (2021).

Q4-2) “Although particle swarms demonstrate micro-cargo transportation capabilities, numerous robots are required to achieve them; moreover, the long-distance transport of multitudinous cargos in a short duration is difficult”, Why is it necessary to rely on swarms of microrobots e.g. Ren et al. also demonstrates such manipulation reliably albeit one at a time:

Ren, L., Nama, N., McNeill, J. M., Soto, F., Yan, Z., Liu, W., ... & Mallouk, T. E. (2019). 3D steerable, acoustically powered microswimmers for single-particle manipulation. *Science advances*, 5(10), eaax3084.

Response to comment

A4-2) When multiple polymer-composite-based robots are cooperated, larger and heavier cargoes than the each robot can be transported (S. Won et al., *Nature Communications*, **10**, 4751, 2019). Although positional control of a single cargo could be achieved by a single polymer-composite-based robot, simultaneous delivery of numerous cargos has been challenged by using a single robot. Moreover, the previous studies described a single cargo with smaller size than the body length of the each robot. Herein, we would like to emphasize a novel transportation strategy of thousands of cargos in an open space as well as a heavier cargo than the CNTY robots. The number of deployed CNTY robots was only seven for the cargo transportation of floating microbeads and semi-submerged millimeter-scale beads in Fig. 4 and 5. Furthermore, owing to collective swimming of the multiple CNTY robots, > 4,000 microplastics could be quickly collected as shown in newly added Fig. 6. We have revised Introduction to describe the purpose of collective motions for the polymer-composite-based robots and have added Fig. 6 as follows:

1) Introduction in manuscript (page 2)

Previous texts

Bioinspired polymeric structures are anticipated to improve the actuation efficiencies of miniaturized robots while minimizing the number of robots required. The locomotion of aquatic and terrestrial living creatures is a noteworthy source for realizing miniaturized polymer-composite-based robots^{16,17}. Appropriate robotic architectures have been constructed using photo-curable^{18,19}, thermo-curable^{20,21}, and thermo-processable²² polymers. Moreover, magnetic materials have been randomly dispersed²³, anisotropically aligned²⁴, or selectively deposited²⁵ in these polymeric structures to realize magnetic

actuation. However, studies on polymer-composite-based magnetic robots have primarily described the independent locomotion of a single robot. Controllable collective actuation of multiple polymer-composite-based robots is required to achieve functionalities beyond those offered by single-robot systems.

Revised texts

Geometric changes in miniaturized robots are anticipated to improve actuation efficiencies of the robots for cargo transportation while minimizing the number of robots required. Polymeric robots are particularly promising since diverse structures, composed of multi-material, can be constructed through polymer processing techniques. For example, polymer-composite-based robots can be fabricated into meshes¹⁹, needles²⁰, jellyfishes²¹, cilia²², and helices²³ by patterning, printing, molding, or thermal fixation of photo-curable, thermo-curable, and thermo-processible polymers. A single polymeric robot in capsule²⁴ or cross²⁵ shape could transport a miniscule cargo to arbitrary destinations. To control these polymeric robots magnetically, a magnetic metal-layer was deposited on the capsule-shaped photoresist²⁴, and magnetic nanowires were embedded in the cross-shaped hydrogel with anisotropic alignment²⁵. However, cargo transportation has primarily focused on a single cargo smaller than the body length of polymeric robots. Recently, tens of robots cooperated to carry an object that was larger and heavier than the individual robots²⁶, yet further challenges remain to deliver multiple cargos rapidly and simultaneously. Controllable collective actuation of multiple polymer-composite-based robots is required to achieve functionalities beyond those offered by single-robot systems.

Newly added references

19. Nguyen, K. T. *et al.* A Magnetically guided self-rolled microrobot for targeted drug delivery, real-time x-ray imaging, and microrobot retrieval. *Adv. Healthc. Mater.* **10**, 2001681 (2021).
20. Lee, S. *et al.* A needle-type microrobot for targeted drug delivery by affixing to a microtissue. *Adv. Healthc. Mater.* **9**, 1901697 (2020).
21. Ren, Z., Hu, W., Dong, X. & Sitti, M. Multi-functional soft-bodied jellyfish-like swimming. *Nat. Commun.* **10**, 2703 (2019).
22. Gu, H. *et al.* Magnetic cilia carpets with programmable metachronal waves. *Nat. Commun.* **11**, 2637 (2020).
23. Won, S., Je, H., Kim, S. & Wie, J. J. Agile underwater swimming of magnetic polymeric microrobots in viscous solutions. *Adv. Intell. Syst.* **4**, 2100269 (2022).
24. Ren, L. *et al.* 3D steerable, acoustically powered microswimmers for single-particle manipulation. *Sci. Adv.* **5**, eaax3084 (2019).
25. Li, C. *et al.* Fast and programmable locomotion of hydrogel-metal hybrids under light and magnetic fields. *Sci. Robot.* **5**, eabb9822 (2020).

2) Fig.6 in manuscript (page 17)

Newly added Figures

We would like to inform the reviewer that the total number of floating beads was miscalculated in the previous manuscript. In Fig. 4, the number of floating beads has been modified from 830 to 3,350. Previously, the total number of floating microbeads was calculated by measuring the area of a single microbead and the area of floating microbeads from the digital camera images. In that method, the low resolution of the digital image prevented accurate

measurement of the area for a single microbead with a small diameter of 255 μm . The overestimated area of a single microbead caused the miscalculation. To determine the number of floating microbeads more precisely, we measured the area of a single particle and the coverage of floating microbeads from the optical microscope images (newly added Supplementary Fig. 24). After converting the microscope image of floating microbeads into a binary image, we measured the average coverage of floating microbeads (black area) per counted number of floating microbeads. As light reflection and multi-focus issue of spheres hindered the image processing, the center of the microbead areas (white area) was excluded from the calculation of coverage area of microbead.

3) *Supplementary Fig.24 in Supplementary Information (page 28)*

Newly added Figure

Supplementary Fig. 24. Binary image of floating microbeads. The unoccupied space among the floating microbeads was considered when calculating the number of floating microbeads. First, we confirmed the optical microscope image of the floating microbeads. After converting the microscope image of floating microbeads into a binary image, we measured the average coverage of floating microbeads (black area) per counted number of floating microbeads. Black and white areas represented microbeads and non-coverage areas, respectively. As light reflection and multi-focus issue of spheres hindered the image processing, the center of the microbead areas (white area in the particle) was excluded from the calculation of coverage area of microbead. The average coverage of floating microbeads was 85.3%. We could calculate the total number of floating microbeads by measuring the area of a single microbead, the area of clustered floating microbeads, and the average coverage of the floating microbeads with the consideration of the unoccupied space among the microbeads.

4) *Methods in manuscript (page 11)*

Newly added texts

In Fig 4 and 6, the total number of floating microbeads was calculated by measuring the area of a single microbead, the area of clustered floating microbeads, and average coverage of floating microbeads with consideration to the unoccupied space among the microbeads (see Supplementary Fig. 24). The deployed number of CNTY robots for cargo transport was minimized for actuation efficiency.

Q5) In the description of synthesis procedure of CNTY robots, it could be clarified explicitly that MWCNTs were used to prepare the yarn. It is only specified in Supplementary Information (SI) and may be useful to defend the physical properties described in the main body of the manuscript.

Response to comment

A5) We thank the reviewer for helpful comment. We have described physical properties, the specific strength, specific modulus, crystallinity, and orientation of the multi-walled carbon nanotube (MWNT) fibers, contributing to preparation of stiff yet lightweight framework (Supplementary Fig. 2 and 3).

1) Results in manuscript (page 3–4)

Previous texts

Carbon nanotubes (CNTs) with an exceptional stiffness (Young's modulus ~ 1 TPa) and low density^{26–28} were twisted to construct the spongy-bone-mimicking CNTY framework. One hundred strands of 7-nm-thick CNTs were twisted to prepare a 340- μm -thick nanoporous CNTY framework (Fig. 1b-(ii) and (iii)).

Revised texts

First, the spongy-bone-mimicking nano-framework was prepared using multi-walled carbon nanotube (MWNT) fibers synthesized through floating catalyst chemical vapor deposition (Supplementary Fig. 1). The synthesized 36- μm -thick MWNT fiber was composed of ten-walled 14 nm-thick MWNTs. The MWNT fiber provided specific strength of 1.19 N tex^{-1} (equivalent to $1.19 \text{ GPa g}^{-1} \text{ cm}^3$) and specific modulus of 46.9 N tex^{-1} (Supplementary Fig. 2). For lightweight nanoporous structure, high densification was intentionally restrained in the MWNT fiber, confirmed by Raman spectra (Supplementary Fig. 3, Fig. 1b-(ii) and (iii)). To build the stiff yet lightweight nano-framework, the 100 MWNT fibers were twisted to form the 340- μm -thick CNTY (Supplementary Fig. 4).

2) Supplementary Fig. 1 in Supplementary Information (page 5)

Newly added figures

Supplementary Fig. 1 Physical properties of multi-walled carbon nanotube (MWNT). Scattering electron microscope (SEM) images of 36- μm -thick single MWNT fiber in (a) low-magnification (top-down view) and (b) high-magnification (radial section view). (c) Transmission electron microscope (TEM) image (top-down view) of a ten-walled 14 nm-thick MWNT composing of the MWNT fiber. (d) Obtained the MWNT fiber with high purity, confirmed by thermogravimetric analyzer (TGA) with a heating rate of $10\text{ }^{\circ}\text{C min}^{-1}$ in an air atmosphere. The weight loss of the MWNT fiber was 86% and residual weight was 6%.

3) *Supplementary Fig. 2 in Supplementary Information (page 6)*
Newly added figure

Supplementary Fig. 2 Stress-strain curve of the single MWNT fiber. Tensile test was performed at 3 mm min^{-1} and gauge length was 1 cm. Specific strength and specific modulus reached up to $1.19 \pm 0.09 \text{ N tex}^{-1}$ (equivalent to 1.19 GPa/SG) and $46.9 \pm 4.5 \text{ N tex}^{-1}$, respectively ($n=10$). SG denotes specific gravity, the density of the materials divided by the density of water. The bulk density of MWNT fiber was 2.6 g cm^{-3} .

4) Supplementary Fig. 3 in Supplementary Information (page 7)
Newly added figures

Supplementary Fig. 3 Raman spectra (inset: polarized Raman spectra) of synthesized MWNT. The intensity ratio of the D-band to G-band (I_D/I_G) in the Raman spectra, which states the crystallinity of CNT, was approximately 0.43 ± 0.11 ($n=5$). The G peak intensity ratio ($I_{G||}/I_{G\perp}$) of the parallel direction to the perpendicular direction along to the CNTY axis in the polarized Raman spectrum, which states the orientation of CNT, was 2.54 ± 0.77 ($n=5$). For lightweight nanoporous structure, the bundling effect was intentionally restrained in the MWNT fiber, confirmed by the crystallinity and orientation^{23,24}.

Newly added references

23. Choi, J. *et al.* Flexible and robust thermoelectric generators based on all-carbon nanotube yarn without metal electrodes. *ACS Nano* **11**, 7608–7614 (2017).
24. Lee, J. *et al.* Direct spinning and densification method for high-performance carbon nanotube fibers. *Nat. Commun.* **10**, 2962 (2019).

We also have revised the manuscript and Supplementary Information to describe the synthesis of MWNT fiber and CNTY more clearly.

5) Methods in manuscript (page 10–11)

Previous texts

Synthesis of CNTY

CNTY was prepared using a floating catalyst chemical vapor deposition method⁴⁹. Methane (Singin Gastech Co., Ltd.; 99.999%), thiophene (Sigma Aldrich; $\geq 99\%$), and ferrocene

(Sigma Aldrich; 98%) were used as the carbon source, promoter, and catalyst precursor for multi-walled CNT (MWCNT) synthesis at 1200 °C, respectively. The MWCNTs were constantly drawn into threads in the reactor by passing through a tap-water bath followed by an acetone bath (Daejung Chemical and Metals). The 100-thread-containing CNTY was directly prepared on a roller at a winding speed of 5 m min⁻¹. These hundred strands were twisted to form a yarn, forming a helical feature and a porous interior between the strands.

Revised texts

Synthesis of CNTY

The CNTY was prepared using a floating catalyst chemical vapor deposition method⁴⁶. Methane (Singin Gastech Co., Ltd.; 99.999%), thiophene (Sigma Aldrich; ≥99%), and ferrocene (Sigma Aldrich; 98%) were used as the carbon source, promoter, and catalyst precursor for MWCNT synthesis at 1200 °C, respectively. Synthesized MWNT fibers were constantly drawn into threads in the reactor by passing through a tap-water bath followed by an acetone bath (Daejung Chemical and Metals). A thread of MWNT fibers was directly prepared on a roller at a winding speed of 5 m min⁻¹. The one hundred fibers of MWNTs were twisted to form a yarn, forming a helical feature and a porous interior between the MWNT fibers.

Q6) Why are the specific aspect ratios of CNTY robots - 2.5,6,13 and 20 chosen for the final experiments?

Response to comment

A5) First, in the manipulation of a single CNTY robot, the object of employing the CNTY robots was to see a clear tendency of swimming modes by increasing ARs to a logarithmic scale. We found that AR-2.5, AR-6, and AR-13 could achieve the bimodal swimming (Fig. 2b–d). Then, the three CNTY robots with different ARs were simultaneous actuated to investigate swimming modes in the three-robot system (Fig. 3e). The AR-13/10-vol.% CNTY robot could demonstrate swimming capability in collective behavior, however, the robot was ineffectual in that the robot was unable to synchronize the rotation frequency at high magnetic frequencies (Supplementary Fig. 16 and 23). Moreover, at low magnetic frequencies to perform assembled rectilinear translational swimming, control of AR-13/10-vol.% CNTY robot was difficult. When the magnetic modular assembly collided with an obstacle, the assembled CNTY robots were disassembled and disassembled AR-13/10-vol.% CNTY robot resulted in rotational swimming, not rectilinear translational swimming. Assembled rectilinear translational swimming of CNTY robots was required for physical blockage of floating microbeads. Hence, we deployed five similar-sized CNTY robots for cargo transport excepting ineffectual AR-13 robots.

Q7) Please mention the magnitude of magnetic fields used for the two swimming modes presented in subsequent swimming modes of CNTY robots.

Response to comment

A7) The maximum magnitude of magnetic field was measured to be 13 mT and only magnetic frequency was varied in order to change swimming modes of the CNTY robots.

1) Results in manuscript (page 5)

Newly added Texts

The designed CNTY robot swims above water under a pulsed quadrupolar electromagnetic field (<13 mT) (Fig. 1c and Supplementary Movie 1).

2) Methods in manuscript (page 11)

Previous Texts

The CNTY robots were actuated at the air–water interface in a container filled with 6.5 mm of water under a pulsed electromagnetic field (<13 mT).

Revised texts

The CNTY robots were actuated at the air–water interface in a container filled with 6.5 mm of water under a pulsed electromagnetic field. The maximum magnitude of the electromagnetic field was measured to be 13 mT. The swimming modes of CNTY robots were changed by switching the magnetic frequency.

Q8) Regarding the experiments shown in magnetic organization:

Q8-1) How do you ensure that assembly of higher number of CNTY robots (e.g. case of 5 or more robots shown in the movies) does not break itself at high frequencies magnetic fields? My main concern is that the assembly and disassembly of multiple CNTY robots is not very clear. Some basic assumptions/quantifications must be clarified:

Response to comment

A8-1) The assembly and disassembly of multiple CNTY robots depend on force competition between drag force and magnetic attractive force in soft joints of robots (C. E. Sing et al., *Proceedings of the National Academy of Sciences of the United States of America*, **107**, 535-540, 2010). Assembled multiple CNTY robots could be disassembled at dominant drag force than magnetic attractive force in the soft joints.

When the number of CNTY robots increased to be 15, the distance among the robots decreased, leading to lateral assembly in addition to head-to-tail assembly (Figure R4). At 3.3 Hz, the multiple CNTY robots exhibited assembled rectilinear translational swimming in a more complex aspect than three- or five-robot systems (Figure R4-a). The laterally assembled 15 CNTY robots was repetitively disassembled and assembled again. We surmise that drag force applied to the soft joints affected by swimming movements and assembled structures. The drag force could overcome the magnetic attractive force in soft joints during rotational movement at vertex, disassembling the magnetic modular assembly. Particularly, the robot at the head/tail of assembly disassembled more than the laterally assembled robot because the lower contact area decreased magnetic attractive force among the robots. The disassembled

CNTY robots were reassembled as inter-robot distance decreased and consequently magnetic attractive force among the robots increased. The predominance of drag force over magnetic attractive force was repeated in the assembled CNTY robots. Assembled rotational swimming appeared from 6.7 Hz to 20 Hz. At the high magnetic frequencies, the assembled rotational swimming was preserved (Figure R4-b) since the laterally assembled CNTY robots were more difficult to be disassembled than the head-to-tail assembly. We realized that increase in the CNTY robots resulted in larger magnitude of vortex due to the larger hydrodynamic volume of the magnetic modular assembly. Therefore, the 19 CNTY robots with assembled rotational swimming transported cuboid cargo with sliding resistance as shown in Supplementary Fig. 28-a of Supplementary Information (response to comment Q1).

Figure R4. Collective swimming of 15 CNTY robots. (a) Assembled rectilinear translational swimming at 3.3 Hz. (b) Assembled rotational swimming at 20 Hz. The particle concentration was 30 vol.%.

The model study to discuss the number effect and assembly structures of the CNTY robots systemically requires very extensive new data sets and analysis of mass-produced robots with micron-scale uniformity; we would like to avoid diluting the primary points of this manuscript, agile multimodal swimming of the biomimetic robots and their collective behavior. We are planning to cover those matters in detail with a complete set of data in the follow-up paper. We have addressed the competition of drag force and magnetic attractive force in soft joints

affecting the collective swimming in page 7 of manuscript.

1) Results in manuscript (page 7)

Previous texts

However, the 10 vol.% CNTY robots with low M_s were disassembled even at a lower magnetic frequency of 10 Hz owing to a weak magnetic attractive force. Fluctuating rotational swimming, which was observed instead of assembled rotational swimming (Fig. 3c, Supplementary Fig. 17), featured engagement and disengagement of multiple CNTY robots. The repetition of engagement and disengagement at 16.7 Hz led to a fluctuation in the distance between the AR-2.5/10-vol.% and AR-13/10-vol.% robots.

Revised texts

However, the 10 vol.% CNTY robots with a low M_s disassembled even at a lower magnetic frequency of 10 Hz due to weak magnetic attractive force. Instead of assembled rotational swimming, the 10 vol.% CNTY robots showed fluctuating rotational swimming at 16.7 Hz (Fig. 3c, Supplementary Fig. 22), which featured engagement and disengagement of multiple CNTY robots. The breakage of magnetic modular assembly occurred when drag force overcame magnetic attractive force³⁴ among soft joints. The repetition of engagement and disengagement led to a fluctuation in the distance between the AR-2.5/10-vol.% and AR-13/10-vol.% robots.

Newly added reference

34. Sing, C. E., Schmid, L., Schneider, M. F., Franke, T. & Alexander-Katza, A. Controlled surface-induced flows from the motion of self-assembled colloidal walkers. *Proc. Natl. Acad. Sci. U. S. A.* **107**, 535–540 (2010).

2) Results in manuscript (page 8)

Previous texts

Unlike in the scenario involving the accompanying AR-13 robots, the magnetic modular CNTY robots could be disassembled owing to the viscous forces, reducing the maximum magnetic frequency at which the assembled rectilinear translational swimming or assembled rotational swimming was permitted.

Revised texts

Fluctuating rotational swimming was observed under the conditions of high magnetic frequency and low particle concentration, where drag force became dominant over magnetic attractive force applied to soft joints of the CNTY robots. Unlike in the scenario involving the accompanying AR-13 robots, the magnetic modular assembly with lower ARs could disassemble due to drag force applied to the soft joints, reducing the maximum magnetic frequency at which the assembled rectilinear translational swimming or assembled rotational swimming was permitted.

Q8-2) At what frequency is the disassembly of robots triggered?

Response to comment

A8-2) As we mentioned in response to comment (Q8-1), fluctuating rotational swimming depended on the competitive relationship of magnetic attractive force and drag force. An increase in particle concentration or decrease in magnetic frequency led to higher magnetic attractive force applied to the soft joints. The breakage of the magnetic modular assembly is correlated with Mason number representing the competition relation between viscous and magnetic attractive force (C. E. Sing et al., *Proceedings of the National Academy of Sciences of the United States of America*, **107**, 535-540, 2010) and we also initially considered this analysis. However, assembly and disassembly behaviors of the CNTY robots are more complicated than the colloid particle systems due to following reasons. First, Mason number is derived from Stokes' law which assume spherical geometry of the particle whereas CNTY robots featured high AR cylindrical morphology. In addition, the CNTY robots swam with not only rotational movement but also translational movement whereas the spherical particles were typically considered to have a motion of rotating or unidirectional dragging. To avoid misleading with inaccurate numbers for governing parameters, we would like not to affirm the effects of Mason number in the manuscript specifically.

Due to complex collective swimming behavior of multiple CNTY robots and difficult quantification of cases inducing disassembly, swimming modes were qualitatively classified on the basis of the experimental data according to magnetic frequency and particle concentration. For assembled rectilinear translational swimming, dissipative magnetic interactions in soft joints was confirmed using deconvoluted harmonic sine functions of the trajectory (Supplementary Fig. 20). The assembled rectilinear translational swimming was a result of dominant inertia force and magnetic attractive force over drag force. Assembled rotational swimming appeared when drag force overcame inertia force of robot yet magnetic attractive force remained dominantly than the drag force. The trajectory of CNTY robots was fitted using the single sine waveform of y-axis coordinate without deconvolution of wave functions (Supplementary Fig. 21). For fluctuating rotational swimming, the distance between CNTY robots fluctuated due to the engagement and disengagement (Supplementary Fig. 22), implying that drag force is higher than both inertia force and magnetic attractive force.

- Supplementary Fig. 20, 21, and 22 in Supplementary Information (page 24–26)

Supplementary Fig. 21 Assembled rotational swimming of multiple CNTY robots. Representative assembled rotational swimming of the AR-2.5/30-vol.%, AR-6/30-vol.%, and AR-13/30-vol.% robots at a magnetic frequency of 13.3 Hz. (a) Time-lapse images. (b) y-axis coordinates of the AR-2.5/30-vol.% robot as a function of time. The assembled rotational swimming data were fitted using a sinusoidal waveform of the y-axis.

Supplementary Fig. 22 Fluctuating rotational swimming of multiple CNTY robots. Representative fluctuating rotational swimming of the AR-2.5/10-vol.%, AR-6/10-vol.%, and AR-13/10-vol.% robots at a magnetic frequency of 16.7 Hz. (a) Time-lapse images

showing engagement and disengagement during the fluctuating rotational swimming mode. (b) Trajectory of fluctuating rotational swimming of the AR-2.5/10-vol.% robot fitted using a sinusoidal waveform of the y-axis. (c) Fluctuation of the distance between the AR-2.5/10-vol.% and AR-13/10-vol.% robots that was modulated by engagement and disengagement, as shown in Fig. 3c.

Q8-3) What number of robots are optimal for collective swimming?

Response to comment

A8-3) The optimized number of CNTY robots could not be assured because deployed number of CNTY robots depended on the shapes, weights, and buoyancy of cargo. The seven robots could transport semi-submerged spherical cargo (52 mg) and 19 robots were deployed to deliver semi-submerged cargo (42 mg) with sliding resistance (see newly added Supplementary Fig. 29 and 30). With respect to floating microbeads, only seven CNTY robots transported thousands of microbeads. The number of CNTY was optimized to be seven in Figures considering vortex generation and actuation efficiency. In Fig. 4 and 5, a too-small number of CNTY robots (< 5) was adverse to generating water vortex whereas a higher number of CNTY robots (> 7) decreased the actuation efficiency. We would like to mention that the multiple CNTY robots show a proof-of-concept of miniaturized magnetic robots for collective behavior that cannot be achieved using a single robot. The model study to scrutinize the effect of the number of CNTY robots on cargo weights requires tens to hundreds of robots. We plan to contrive mass-production process of hundreds of uniform microrobots in the follow-up paper.

Q9) Regarding the fluctuating rotational swimming mode discussed in the manuscript:
Q9-1) It seems that in this case (SI Movie 3) the assembly of CNTY robots breaks at high frequencies?

Response to comment

A9-1) Supplementary Move 3 represents assembled rectilinear translational swimming and assembled rotational swimming of CNTY robots. In the equilibrium state, the five 30 vol.% CNTY robots swam with assembled rotational swimming, not fluctuating rotational swimming.

In the intermediate state from 3 Hz to 10 Hz, the CNTY robots were repeatedly engaged and disengaged in order that the opposing magnetic dipoles were encountered and then were magnetically articulated to form the chain-like assembly.

1) Supplementary Fig. 19-a in Supplementary Information (page 23)

Previous Figure

Revised Figure and newly added texts

Q9-2) What is the utility of fluctuating rotational swimming?

Response to comment

A9-2) As shown in Fig. 3d, e, and Supplementary Fig.20–23, three collective swimming modes and transition tendency were qualitatively and extensively investigated according to parameters of magnetic frequency, particle concentration, and aspect ratio. We figured out that competition of drag force and magnetic attractive force affected the three collective swimming modes. Fluctuating rotational swimming was the resultant mode representing that drag overcame magnetic attractive force in soft joints.

Q9-3) Besides, defining the range where CNTY robots disassemble, it is not clear how this mode contributes to the utility of collective CNTY motion?

Response to comment

A9-3) As we mentioned in response to comment (Q8-1), the assembly and disassembly of multiple CNTY robots depends on force competition between drag force and magnetic

attractive force in soft joints of robots. Assembled multiple CNTY robots could be disassembled at dominant drag force over magnetic attractive force in the soft joints. Fluctuating rotational swimming was also induced when the multiple CNTY robots pushed cargo directly and the magnetic attractive force was dissipated (Fig. 5d). Although the 30 vol.% CNTY robots were disassembled, significant vortex generation by the 30 vol.% robots attributed transportation of semi-submerged cargo.

Q10) Regarding the vortex generation experiments:

Q10-1) Although the demonstration looks promising, the direction in which payload of microbeads can be carried looks subjective to the presence of an obstacle. Case in point, beads go into the confined area when CNTY robots are rotating next to the bottom green edge whereas they come outside the confined area when CNTY robots are rotating next to the top green edge. How could CNTY robots manipulate objects in absence of any edges or obstacles? Please justify.

Response to comment

A10-1) We thank the reviewer for the complimentary comment. In an open space, floating microbeads could be trapped along the vortex (newly added Fig. 6). The multiple CNTY robots with assembled rotational swimming achieved a collection of > 4,000 microplastics as follows:

1) Results in manuscript (page 9-10)

Newly added texts

Furthermore, seven 30 vol.% CNTY robots accomplished collection of thousands of microplastics in an open space by generating large-magnitude vortex (Fig. 6 and Supplementary Movie 10). When the CNTY robots were adjacent to a cluster of floating microplastics, the microplastics initiated to be ensnared in the CW vortex, thereby being trapped in the vortex (Fig. 6a). Under the manipulation of the motorized stage, CNTY robots transported and merged the scattered clusters of floating polyethylene microbeads ($D= 850 \mu\text{m}$) using the assembled rotational swimming (Fig. 6b and c). The total number of 4,630 microplastics was successfully collected within 150 s.

2) Fig.6 in manuscript (page 17)

Newly added Figures

The divided space was designed for blocking or transporting capability of numerous floating microbeads through assembled rectilinear translational swimming or assembled rotational swimming, respectively. After the CNTY robots rotated at the vertex of the trajectory, the CNTY robots initiated rectilinear translational swimming, which could physically block the microbeads from entering. We confirmed that the physical blockage by analyzing high-speed

camera (newly added Supplementary Fig. 25). Importantly, the agility of rectilinear translational swimming contributed to this physical blockage. Over the course of 1 min, the CNTY robots successfully prevented the influx of floating microbeads.

3) Supplementary Fig.25 in Supplementary Information (page 29)

Newly added Figures

Q10-2) In SI Movie 8, the motion of CNTY carrying cargo is triggered by manual movement of the container, hence such manipulation cannot take place in absence of an external stimulus. How do authors explain this application for inaccessible or confined workspaces? Secondly, the rolling motion of spherical cargo assisted in the manipulation which may not otherwise work for irregular/non-spherical geometries.

Response to comment

A10-2) We thank the reviewer for the constructive comment. Previously, the CNTY robots were steered by moving the container manually, and cargo transportation was difficult to control delicately. Although the relative position of the robots was manually controlled, it was previously intended to propose the proof-of-concept of cargo transport via collective swimming of multiple CNTY robots. However, we agree with the reviewer's point. For entirely contactless steering of the robots and improvement of transportation precision, we have devised a motorized stage with two degrees of freedom (2-DoF), coupled with the electromagnetic coils.

By manipulating the motorized stage, the multiple CNTY robots enabled not only spherical cargo with rolling resistance but also cuboid cargo with sliding resistance to the desired location (newly added Supplementary Fig. 28 and 29). Furthermore, thousands of floating microplastics could be collected as shown in Fig. 6 (response to comment Q10-1).

1) Results in manuscript (page 9-10)

Newly added texts

To improve transportation precision of various cargos (Supplementary Movie 9), we devised a motorized stage having two degrees of freedom (2-DoF)³⁶ coupled with the electromagnetic coils. Through manipulating the location of the motorized stage instead of the water container, the spherical cargo with rolling resistance could be delivered with more delicacy to its designated position (Supplementary Fig. 29). The transportation capability was demonstrated even for semi-submerged or underwater cargos with sliding resistance by above-water collective swimming or underwater collective swimming, respectively (Supplementary Fig. 30).

Newly added references

36. Kim, Y. *et al.* Effects of helix geometry on magnetic guiding of helical polymer composites on a gastric cancer model: a feasibility study. *Materials*. **13**, 1014 (2020).

2) Supplementary Fig.29 in Supplementary Information (page 33)

Newly added Figure

Supplementary Fig. 29 Transportation of semi-submerged spherical cargo by manipulating the motorized stage. (a) Motorized stage having two degrees of freedom (2-DoF) coupled with the electromagnetic coils. (b) Delicate positional control of seven 30 vol.% CNTY robots in order to transport the spherical cargo by manipulating the location of the motorized stage. Diameter and weight of the cargo was 4 mm and 52 mg, respectively. The magnetic frequency was 13.3 Hz. Seven 30 vol.% CNTY robots were deployed.

3) *Supplementary Fig.30 in Supplementary Information (page 34)*

Newly added Figure

Q11) In SI Fig. 7 - at what applied magnetic field does saturation M_s occur?

Response to comment

A11) We have added the measurement condition of saturation magnetization as follows:

1) Legend of Supplementary Fig. 11 in Supplementary Information (page 23)

Previous Texts

Supplementary Fig. 7 Magnetization properties of the ternary nanocomposite. (a) Magnetization of bulk iron particles and the ternary nanocomposites measured using a vibrating-sample magnetometer. (b) Saturation magnetization (M_s) of bulk iron particles and the ternary nanocomposites. The M_s of bulk powder of iron particles was estimated to be 237 emu g^{-1} , and those of the ternary nanocomposites were found to be 14, 22, 40, 69, 118, and 152 emu g^{-1} at magnetic particle concentrations of 1, 2, 5, 10, 20, and 30 vol.% in the PDMS prepolymer, respectively.

Revised Texts

Supplementary Fig. 11 Magnetization properties of the ternary nanocomposite. (a) Magnetization of bulk iron particles and the ternary nanocomposites measured using a vibrating-sample magnetometer. (b) Saturation magnetization (M_s) of bulk iron particles and the ternary nanocomposites, **measured at 70 kOe**. The M_s of bulk powder of iron particles was estimated to be 237 emu g^{-1} , and those of the ternary nanocomposites were found to be 14, 22, 40, 69, 118, and 152 emu g^{-1} at magnetic particle concentrations of 1, 2, 5, 10, 20, and 30 vol.% in the PDMS prepolymer, respectively.

Detailed comments:

Q12) Within Section: Biomimetic hierarchical structure of robots:

“The human arm exerts a strong force in proportion to the thickness of muscle bundles”, this claim sounds vague, could you please provide numbers here?

Response to comment

A12) We thank the reviewer for the helpful comment. Generally, increase in the cross-sectional area of muscle bundles results in high muscle force (R. J. Maughan et al, *The Journal of*

Physiology, **333**, 37–49, 1983). However, with the respect to force production of skeletal muscles, various factors are complexly correlated, including muscle design, fiber length, contraction velocity of muscles. To avoid the misunderstanding of muscle force, we have revised the manuscript, focusing on the description of biomimetic hierarchical structure as follows:

1) Results in manuscript (page 3)

Previous Texts

The human arm exerts a strong force in proportion to the thickness of muscle bundles by means of the skeletal muscle surrounding lightweight yet stiff bones. The spongy bone component of the long compact bones articulated in the elbow comprises hierarchically arranged collagen fibers and porous networks, and remains lightweight despite containing a packed bone marrow.

Revised Texts

Movements of the human arm are produced by forces generated from skeletal muscles surrounding the stiff long bones. Along the interior toward the end of the long bones, the spongy bone shows a hierarchical composition of collagen fiber bundles followed by a porous network while the bone marrow fills in the bone. The porosity of spongy bone offers lightness to the long bones while surrounding skeletal muscles are bounded by the connective tissue known as deep fascia.

Q13) Within Section: Agile multimodal swimming:

Q13-1) “CW rotational swimming by increasing the magnetic frequency to 6.7 Hz.”, please refer SI Fig. 8, and first part of SI Movie 2a.

Response to comment

A13-1) **Fig. 1b** represents swimming of **AR-6/10 vol.% CNTY robot** while the **first section in Movie 2** is the case of **AR-2.5/20 vol.% CNTY robot**. Rotational swimming of the AR-6/10 vol.% and AR-2.5/20 vol.% CNTY robots appeared at 6.7 Hz and 10 Hz, respectively.

Supplementary Fig. 12 (previous Supplementary Fig. 8) explains the mechanism of bimodal swimming by using the AR-2.5 robot. The AR-2.5 robot was drawn in order to distinguish between rectilinear translational swimming and rotational swimming clearly. The magnetic frequency exhibiting CW rotational swimming depends on AR and particle concentration of the robots. The diagram regarding bimodal swimming modes is summarized in Fig. 2b–d.

Q13-2) “When the magnetic frequency was switched from 3.3 Hz to 6.7 Hz, the centroid–vertex distance”, define centroid-vertex distance. I assume this is related to the square-shaped trajectory, but the term is not well defined in any of the figures.

Response to comment

A13-2) We thank the reviewer for the detailed comment. We supplemented Fig. 1d to describe centroid–vertex distance more precisely.

1) Fig. 1d in manuscript (page 12)

Previous Figure

Revised Figure

2) Supplementary Fig. 18-e and f in Supplementary Information (page 22)

Previous Figures

Revised Figures

Q14) Within Section: Magnetic organization

Q14-1) “the viscous force applied to the robots causes dissipation of the magnetic attractive force in the soft joints”, the sentence is unclear.

Response to comment

A14-1) We thank the reviewer for the constructive comment. During vertex rotation, the magnetic modular assembly experienced viscous torque. Drag force was perpendicularly applied to the axially assembled robots, attenuating the magnetic attractive force among the soft joints. The magnetic attractive force was recovered during subsequent rectilinear translational swimming. We revised the manuscript as follow:

1) Results in manuscript (page 6)

Previous texts

During vertex rotation, the viscous force applied to the robots causes dissipation of the magnetic attractive force in the soft joints. Consequently, the soft joints exhibit rotational degrees of freedom and periodically undulate to overcome the viscous force of water.

Revised texts

During rotation at the vertex of swimming trajectory, the magnetic modular assembly experienced viscous torque. Drag force was perpendicularly applied to the axially assembled robots, attenuating the magnetic attractive force among the soft joints. Magnetic attractive force was recovered during subsequent rectilinear translational swimming. As a result, the soft joints exhibited rotational degrees of freedom and periodically undulated to endure drag force.

Q14-2) In SI Fig 15, is it deconvoluted sine curves of only one AR i.e., 2.5 case?

Response to comment

A14-2) Deconvolution of the AR-2.5/20-vol.% CNTY robot represented the effect of magnetically interacting dynamic joints. As the AR-2.5, AR-6, and AR-13 CNTY robots magnetically articulated, the three CNTY robots mutually interacted during assembled rectilinear translational swimming. As a result, the AR-6/20-vol.% and AR-13/20-vol.% CNTY robots also showed the deconvoluted harmonic sine curves (Supplementary Fig. 20; previous Supplementary Fig. 15).

1) Supplementary Fig. 20-d and e in Supplementary Information (page 24)

Newly added Figures

Q15) Fig. 5 caption: 13.3Hz - description in SI Fig. 20 suggest this to be camera imaging frequency. Please clarify.

Response to comment

A15) We thank the reviewer for the detailed comment. We corrected the description of magnetic frequency in the legend as follows:

1) Legend of Supplementary Fig. 27 in Supplementary Information (page 30)

Previous Texts

Revised Texts

Q16) Please check the angular frequency described in SI Note 1, they have inconsistent units.

Response to comment

A16) We thank the reviewer for the detailed comment. We revised the angular frequency unit from Hz to rad s^{-1} in Supplementary Note 1.

2) Supplementary Note 1 in Supplementary Information (page 35)

Previous Texts

The complex viscosities of the PDMS–iron-particle mixtures were measured using a parallel-plate rheometer (MCR302, Anton Paar) with a plate gap of 5 mm. The experiments were conducted at 23.8 °C and angular frequencies of 1–100 Hz. The complex viscosity was calculated at an angular frequency of 3 rad s⁻¹.

Revised Texts

The complex viscosities of the PDMS–iron-particle mixtures were measured using a parallel-plate rheometer (MCR302, Anton Paar) with a plate gap of 5 mm. The experiments were conducted at 23.8 °C and angular frequencies of 1–100 rad s⁻¹. The complex viscosity was calculated at an angular frequency of 3 rad s⁻¹.

REVIEWER COMMENTS

Reviewer #1 (Remarks to the Author):

The authors have made many detailed revisions according to the reviewer's comments. The quality of this manuscript is significantly improved. Although most of the questions have been well addressed, there are some explanations that I think are unclear.

1. Previous question 1: The authors imply in the introduction that requiring numerous single particles is one of the disadvantages of colloidal particle swarms. I don't think the number of robots is a key property of the swarming system, especially particle swarms. Besides, I agree that the CNTY robot swarm generates water vortex efficiently with fewer robot numbers. But when considering the swarm as a whole, it doesn't seem to matter how many individual robots it consists of.

2. Previous question 7: In the transportation of a heavy sphere cargo, the cargo is "captured" by the CNTY robots rotating around it at a relatively high frequency. The authors devise a 2-DoF motorized stage that improves the transportation precision of the system significantly. However, the way to release the cargo (i.e., the separation of the sphere from the robots) is unclear. I think there should be a specific method (such as modulating field parameters) to quickly separate the robots from the cargo after the transportation is completed.

3. Previous question 8: The authors demonstrate the transportation of a submerged cargo by underwater swimming of the CNTY robots, which is achieved by intentionally submerging the robots in water. Can you describe the process in more detail? For example, the principle of the lightweight robot sinking to the bottom of the water. How long does it take to submerge for the robot to completely sink? Can the sunken robot perform above-water swimming again? Will the CNTY robots sink to the bottom after swimming above the water for a period of time?

Reviewer #1

The authors have made many detailed revisions according to the reviewer's comments. The quality of this manuscript is significantly improved. Although most of the questions have been well addressed, there are some explanations that I think are unclear.

Response to comment

We thank the reviewer for the valuable evaluation and the constructive comments.

Q1) Previous question 1: The authors imply in the introduction that requiring numerous single particles is one of the disadvantages of colloidal particle swarms. I don't think the number of robots is a key property of the swarming system, especially particle swarms. Besides, I agree that the CNTY robot swarm generates water vortex efficiently with fewer robot numbers. But when considering the swarm as a whole, it doesn't seem to matter how many individual robots it consists of.

Response to comment

A1) We thank the review for the insightful comment. Considering a swarm as a whole as the reviewer commented, we agree that a slight change in the number of robots is not a significant factor for cargo transport. We would like to emphasize that the difference between the CNTY robot and the particle robot is agile swimming speed of the CNTY robots at the water-air interface (up to 180 body lengths per second). The agile above-water swimmability of CNTY robots was preserved even in the collective behavior of multi-robot systems, enabling vortex-induced transportation. Particle robots can float on the surface of water with the assistance of surface tension of water. However, when an external stimulus is applied, the particle robot easily sinks to the bottom of the water container due to the high density. Eventually, actuation control of particle robot swarms is difficult on the water surface. Lightweight body frame is required to prevent magnetic robots from sinking during the above-water swimming. We prepared lightweight nanoporous framework with a high aspect ratio by twisting low-density CNT fibers. Furthermore, when the density and weight of robots are identical, contact area between the robot and water affects the buoyancy. Rod-like magnetic robots can float on the water more easily than spherical particle-like magnetic robots due to higher surface area of the rod-like magnetic robots. By dip-coating processing of the ternary polymer nanocomposites, we obtained the high-aspect ratio CNTY robots. We have deleted the number effect of the particle robots and revised the manuscript as follows:

1) Introduction in manuscript (page 2)

Previous texts

Reconfigurable swarms of magnetic microparticles have been realized by varying the axes and frequencies of the alternating magnetic fields^{16,17}. However, most studies on collective behavior have focused on colloidal particle systems¹⁸. Although particle swarms

demonstrate micro-cargo transportation capabilities, numerous robots are required to achieve them; moreover, the long-distance transport of multitudinous cargos in a short duration is difficult.

Geometric changes in miniaturized robots are anticipated to improve actuation efficiencies of the robots for cargo transportation while minimizing the number of robots required.

Revised texts

Reconfigurable swarms of magnetic microparticles have been realized by varying the axes and frequencies of the alternating magnetic fields^{16,17}. However, most studies on collective behavior have focused on colloidal particle systems¹⁸. Although particle swarms demonstrate micro-cargo transportation capabilities, they face challenges in the transportation of multitudinous cargos. When the amount of cargo increases, long-distance transport and location control become difficult to achieve in a limited time duration.

Geometric changes in miniaturized robots are anticipated to improve actuation efficiencies of the robots for cargo transportation.

Previous response to comment

Previous Q1) Compared with colloidal particle systems consisting of numerous colloidal particles, polymeric robots can minimize the number of robots required. Is this one of the advantages of the polymeric robot system? Is there a more specific explanation for this?

Previous A1) Polymeric robots are promising in robotics since diverse structures composed of multi-material can be constructed through various polymer processing techniques. Polymer-composite-based robots can be fabricated in arbitrary complex shapes such as meshes, needles, jellyfishes, cilia, and helices by patterning, printing, molding, or thermal fixation of photo-curable, thermo-curable, and thermo-processible polymers. In this study, we fabricated ternary hierarchical nanocomposites by dip-coating and thermo-curing processes with the purpose to realize magnetic robots that can generate a water vortex and transport multiple cargos via the vortex.

In designing the magnetic robot, a lightweight robot body was essential to impart agile above-water swimmability. However, importantly, a stiff body frame was also required to overcome surface tension and drag force for swimming. Considering the well-known Ashby plots (modulus vs density), rigidity and lightweightness are in trade-off relationships. For example, robots with high Young's modulus can result in considerable vortex generation. However, increase in Young's modulus indicates that high density materials are necessary according to the Ashby plots. Furthermore, metal-based particle robots can cause submergence in water due to the high density. Therefore, we employed carbon nanotube yarn (CNTY) as rigid yet lightweight nanoporous framework as well as permeable template for dip-coating of thermo-curable magnetic polymer composites to address this tradeoff relationship.

Moreover, colloidal particles suffer from limited geometric and compositional variation. The low aspect ratios of particle robots are because the colloids are synthesized in a direction that minimizes the surface energy of each particle. However, we conceived a magnetic robot

with high aspect ratio (AR) and their magnetic assembly as a more effective strategy for generating the vortex. As the magnetically assembled high-AR robots had a large hydrodynamic volume, rotational swimming of the magnetically assembled robots was expected to exert high rotational force and induce a large-magnitude vortex. To preserve the assembled structure even during the rapid swimming, magnetic attractive force among robots should exceed drag force. To enhance interrobot magnetic attractions, we designed core-shell-like robot by binding highly concentrated magnetic particles only on the robot surface, instead of embedding the particles inside the robot since magnetic force between the particles drastically increases by decreasing the interparticle distances (A. Mehdizadeh et al., *Acta Mechanica Sinica*, **26**, 921–929, 2010).

To meet the requirements, we introduced a biomimetic hierarchical nano/microstructure using nanoporous CNTY framework, PDMS, and magnetic microparticles, which emulate the spongy bone, connective tissue, and skeletal muscle, respectively. First, stiff yet lightweight nano-framework was prepared by synthesizing multi-walled carbon nanotube (MWNT), followed by twisting the MWNT to form the nanoporous CNTY framework (newly added Supplementary Fig. 1, 2, and 3). Then, dip-coating process was implemented using a mixture of PDMS prepolymer and magnetic microparticles to bind highly concentrated magnetic particles on the robot surface. The nanoporous CNTY framework was immersed in the mixture of PDMS prepolymer and magnetic microparticle, followed by drainage of excessive mixture. Because the high concentration of magnetic particles could cause a drastic increase in the viscosity of the composite mixture, we handled the composite mixture by utilizing the prepolymer. Moreover, during the dip-coating process, the nanoporous CNTY framework operated as a membrane to permeate the liquid-phase PDMS prepolymer through capillary forces, whereas the 5- μm -sized magnetic microparticles were excluded from the core region. The magnetic microparticles at the CNTY surface were bound through thermal curing of PDMS via hydrosilylation polymerization. The PDMS acted as a binder for the magnetic particles in a fashion similar to the deep fascia which binds muscle fibers. Absence of the PDMS binder could hinder the adhesion of magnetic particles onto the CNTY framework due to the weak cohesive energy between the magnetic particles. We also could varied the coating thickness of the magnetic composite layer by changing the concentration of magnetic particles (Supplementary Fig. 8).

As a result, the AR-2.5/30-vol.% robot accomplished swimming velocity of 180 BL s^{-1} . We previously reported polymer-composite-based robot with AR-2, showing only above-water swimming velocity of 3.8 BL s^{-1} (S. Won et al., *Nature Communications*, **10**, 4751, 2019). The low swimming velocity and low-AR could not generate vortex. Maximum AR was 4 for magnetic actuation of the polymer-composite-based robots. Furthermore, for the high-AR CNTY robots of AR-6/30-vol.% and AR-13/30-vol.% CNTY robots, the maximum swimming velocity was measured to be 66 BL s^{-1} and 40 BL s^{-1} respectively.

We also would like to highlight polymer-composite-based robot systems in terms of the actuation efficiency by minimizing the number of robots. Only with a single robot, cargos could be transported to arbitrary destinations (L. Ren et al., *Science Advances*, **5**, eaax3084, 2019; C. Li et al, *Science Robotics*, **5**, eabb9822, 2020). However, these demonstrations were the cases of a single cargo smaller than the robot body length. Although we recently reported that tens of robots could cooperate to carry a larger and heavier cargo than the individual robots (S. Won et al., *Nature Communications*, **10**, 4751, 2019), further challenge remains to deliver numerous

cargos simultaneously and rapidly. In this study, the number of CNTY robots was seven for cargo transportation of thousands of floating microbeads and semi-submerged millimeter-scale bead 52 times heavier than the CNTY robot.

As it seems that our intended design principle and significance of work were not fully delivered from the previous manuscript, we extensively revised the manuscript as follows:

1) Introduction in manuscript (page 2)

Previous texts

Bioinspired polymeric structures are anticipated to improve the actuation efficiencies of miniaturized robots while minimizing the number of robots required. The locomotion of aquatic and terrestrial living creatures is a noteworthy source for realizing miniaturized polymer-composite-based robots^{16,17}. Appropriate robotic architectures have been constructed using photo-curable^{18,19}, thermo-curable^{20,21}, and thermo-processable²² polymers. Moreover, magnetic materials have been randomly dispersed²³, anisotropically aligned²⁴, or selectively deposited²⁵ in these polymeric structures to realize magnetic actuation. However, studies on polymer-composite-based magnetic robots have primarily described the independent locomotion of a single robot. Controllable collective actuation of multiple polymer-composite-based robots is required to achieve functionalities beyond those offered by single-robot systems.

Revised texts

Geometric changes in miniaturized robots are anticipated to improve actuation efficiencies of the robots for cargo transportation while minimizing the number of robots required. Polymeric robots are particularly promising since diverse structures, composed of multi-material, can be constructed through polymer processing techniques. For example, polymer-composite-based robots can be fabricated into meshes¹⁹, needles²⁰, jellyfishes²¹, cilia²², and helices²³ by patterning, printing, molding, or thermal fixation of photo-curable, thermo-curable, and thermo-processible polymers. A single polymeric robot in capsule²⁴ or cross²⁵ shape could transport a miniscule cargo to arbitrary destinations. To control these polymeric robots magnetically, a magnetic metal-layer was deposited on the capsule-shaped photoresist²⁴, and magnetic nanowires were embedded in the cross-shaped hydrogel with anisotropic alignment²⁵. However, cargo transportation has primarily focused on a single cargo smaller than the body length of polymeric robots. Recently, tens of robots cooperated to carry an object that was larger and heavier than the individual robots²⁶, yet further challenges remain to deliver multiple cargos rapidly and simultaneously. Controllable collective actuation of multiple polymer-composite-based robots is required to achieve functionalities beyond those offered by single-robot systems.

Newly added references

19. Nguyen, K. T. *et al.* A Magnetically guided self-rolled microrobot for targeted drug delivery, real-time x-ray imaging, and microrobot retrieval. *Adv. Healthc. Mater.* **10**, 2001681 (2021).
20. Lee, S. *et al.* A needle-type microrobot for targeted drug delivery by affixing to a

microtissue. *Adv. Healthc. Mater.* **9**, 1901697 (2020).

21. Ren, Z., Hu, W., Dong, X. & Sitti, M. Multi-functional soft-bodied jellyfish-like swimming. *Nat. Commun.* **10**, 2703 (2019).
22. Gu, H. *et al.* Magnetic cilia carpets with programmable metachronal waves. *Nat. Commun.* **11**, 2637 (2020).
23. Won, S., Je, H., Kim, S. & Wie, J. J. Agile underwater swimming of magnetic polymeric microrobots in viscous solutions. *Adv. Intell. Syst.* **4**, 2100269 (2022).

2) Results in manuscript (page 3-4)

Previous texts

Biomimetic hierarchical nano/microstructures were prepared using a nanoporous carbon nanotube yarn (CNTY) framework, polydimethylsiloxane (PDMS), and magnetic microparticles, which emulate the spongy bone, connective tissue, and skeletal muscle, respectively, in brachial anatomy (Fig. 1a and b-(i)). The human arm exerts a strong force in proportion to the thickness of muscle bundles by means of the skeletal muscle surrounding lightweight yet stiff bones. The spongy bone component of the long compact bones articulated in the elbow comprises hierarchically arranged collagen fibers and porous networks, and remains lightweight despite containing a packed bone marrow. Carbon nanotubes (CNTs) with an exceptional stiffness (Young's modulus ~ 1 TPa) and low density^{26–28} were twisted to construct the spongy-bone-mimicking CNTY framework. One hundred strands of 7-nm-thick CNTs were twisted to prepare a 340- μm -thick nanoporous CNTY framework (Fig. 1b-(ii) and (iii)).

Revised texts

For a magnetic robot to be capable of vortex generation at the air-water interface as well as cargo transportation, a lightweight robot body was essential to impart agile above-water swimmability. A stiff body frame was also required to overcome surface tension and drag force in swimming even though the two variables of lightweightness and rigidity are considered trade-offs according to Ashby plots²⁷ showing Young's modulus versus density. We further conceived a magnetic robot with high aspect ratio (AR) and collective actuation of the high-AR robots as a more effective strategy for generating the vortex. When the high-AR robots assembled magnetically to have large hydrodynamic volume, rotational swimming of the magnetically assembled robots was expected to exert high rotational force and induce a large-magnitude vortex. Magnetic assembly of the multiple robots required larger magnetic attractive force among robots than drag force to preserve the assembled structure even through rapid swimming. To enhance inter-robot magnetic attractions, we considered a core-shell-like robot by binding highly concentrated magnetic particles only on the robot surface, instead of embedding particles inside the robot since the magnetic force between particles drastically increases by decreasing interparticle distances^{28,29}. The robot design that met these requirements was inspired by brachial anatomy (Fig. 1a, b-(i)). Movements of the human arm are produced by forces generated from skeletal muscles surrounding the stiff long bones. Along the interior toward the end of the long bones, the spongy bone shows a hierarchical composition of collagen fiber bundles followed by a

porous network while the bone marrow fills in the bone. The porosity of spongy bone offers lightness to the long bones while surrounding skeletal muscles are bounded by the connective tissue known as deep fascia. We introduced a biomimetic hierarchical nano/microstructure with nanoporous carbon nanotube yarn (CNTY) framework, polydimethylsiloxane (PDMS), and magnetic microparticles, emulating the spongy bone, connective tissue, and skeletal muscle, respectively. First, the spongy-bone-mimicking nano-framework was prepared using multi-walled carbon nanotube (MWNT) fibers synthesized through floating catalyst chemical vapor deposition (Supplementary Fig. 1). The synthesized 36- μm -thick MWNT fiber was composed of ten-walled 14 nm-thick MWNTs. The MWNT fiber provided specific strength of 1.19 N tex^{-1} (equivalent to $1.19 \text{ GPa g}^{-1} \text{ cm}^3$) and specific modulus of 46.9 N tex^{-1} (Supplementary Fig. 2). For lightweight nanoporous structure, high densification was intentionally restrained in the MWNT fiber, confirmed by Raman spectra (Supplementary Fig. 3, Fig. 1b-(ii), (iii)). To build the stiff yet lightweight nano-framework, 100 MWNT fibers were twisted to form the 340- μm -thick CNTY (Supplementary Fig. 4).

Then, a dip-coating process was implemented using a mixture of PDMS prepolymer and iron microparticles to bind highly concentrated magnetic particles on the robot surface. The nanoporous CNTY framework was immersed in the mixture of PDMS prepolymer and magnetic microparticles, followed by drainage of excessive mixture (Supplementary Fig. 5). During the dip-coating process, the nanoporous CNTY framework operated as a membrane to permeate the liquid-phase PDMS prepolymer through capillary forces³⁰, whereas the 5- μm -sized iron microparticles³¹ were excluded from the core region, as shown in Fig. 1b-(i). The magnetic microparticles at the CNTY surface were bound through thermal curing of PDMS via hydrosilylation polymerization. The PDMS acted as a binder for the magnetic particles in a fashion similar to the deep fascia and mimicked the binding of muscle fibers. Absence of the PDMS binder could hinder the adhesion of magnetic particles onto the CNTY framework due to the weak cohesive energy between the magnetic particles. By spatially selective coating of the magnetic microparticles onto the surface of the nanoporous CNTY framework, the coating thickness of the magnetic composite layer could be varied with the concentration of magnetic particles dispersed in the PDMS prepolymer (Supplementary Fig. 6, 7, 8).

Q2) Previous question 7: In the transportation of a heavy sphere cargo, the cargo is "captured" by the CNTY robots rotating around it at a relatively high frequency. The authors devise a 2-DoF motorized stage that improves the transportation precision of the system significantly. However, the way to release the cargo (i.e., the separation of the sphere from the robots) is unclear. I think there should be a specific method (such as modulating field parameters) to quickly separate the robots from the cargo after the transportation is completed.

Response to comment

A2) We thank the reviewer for the detailed comment. We noticed that the release mechanism of cargo was unclearly described in previous manuscript. The CNTY robots were separated from the cargo by rapidly moving the CNTY robots in a different direction from the cargo.

Since the CNTY robots spontaneously reassembled even right after the disassembly process, the location control of multiple CNTY robots was easy and simple. To clarify the release mechanism of cargo, we have investigated the CNTY robots by a high-speed camera (newly added in Supplementary Figure. 29-(c)).

When the CNTY robots swam with assembled rotational swimming, the CNTY robots were able to push the cargo by elastic energy transfer. As a result, the cargo could be placed on the desired location. For this cargo positioning process, the cargo is not captured by the CNTY robots but transferred by pushing behavior of CNTY robots. To avoid any potential misunderstanding, we added the description for the positioning process in legend of Supplementary Figure 29-(c). Although the CNTY robots disassembled, the CNTY robots instantaneously reassembled by magnetic attractive force among the robots and could swim with the assembled rotational swimming. By simply moving the CNTY robots away from the cargo before pushing the cargo again, all robots could be separated immediately from the cargo. The positioning and separating process of cargo were implemented by simply manipulating the location of the motorized stage without varying the magnetic frequency of magnetic fields.

1) Supplementary Fig.29 in Supplementary Information (page 33)

Newly added Figure and texts

positioning and separating process of the cargo were implemented solely through manipulation of the motorized stage location without varying the magnetic frequency of the magnetic field.

Previous response to comment

Previous Q7) In the transportation of semi-submerged cargo (Fig. 5c and Supplementary Movie 8), it seems that the collective robot cannot precisely transport the cargo to a specific point. Dose it because the separation of the robots and cargo is hard to control?

Previous A7) Previously, the CNTY robots were steered by moving the container manually, and cargo transportation was difficult to control delicately. To improve transportation precision, we have devised a motorized stage with two degrees of freedom (2-DoF), coupled with the electromagnetic coils (newly added Supplementary Fig. 29-a). By precisely manipulating the location of motorized stage, the multiple CNTY robots successfully demonstrated transportation of a spherical cargo with rolling resistance (newly added Supplementary Fig. 29-b) as well as a cuboid cargo with sliding resistance (newly added Supplementary Fig. 30) to the desired location (Supplementary Movie 9). Furthermore, we could collect thousands of floating microplastics in an open space, as shown in Fig. 6 and Supplementary Movie 10.

1) Results in manuscript (page 9-10)

Newly added texts

To improve transportation precision of various cargos (Supplementary Movie 9), we devised a motorized stage having two degrees of freedom (2-DoF)³⁶ coupled with the electromagnetic coils. Through manipulating the location of the motorized stage instead of the water container, the spherical cargo with rolling resistance could be delivered with more delicacy to its designated position (Supplementary Fig. 29). The transportation capability was demonstrated even for semi-submerged or underwater cargos with sliding resistance by above-water collective swimming or underwater collective swimming, respectively (Supplementary Fig. 30).

Furthermore, seven 30 vol.% CNTY robots accomplished the collection of thousands of microplastics in an open space by generating a large-magnitude vortex (Fig. 6, Supplementary Movie 10). When the CNTY robots were adjacent to a cluster of floating microplastics, the microplastics became ensnared in the CW vortex. (Fig. 6a). Under manipulation of the motorized stage, CNTY robots transported and merged the scattered clusters of floating polyethylene microbeads ($D= 850 \mu\text{m}$) using assembled rotational swimming (Fig. 6b, c). A total number of 4,630 microplastics was successfully collected within 150 s.

Newly added references

36. Kim, Y. *et al.* Effects of helix geometry on magnetic guiding of helical polymer composites on a gastric cancer model: a feasibility study. *Materials*. **13**, 1014 (2020).

2) *Supplementary Fig.29 in Supplementary Information (page 33)*

Newly added Figure

Supplementary Fig. 29 Transportation of semi-submerged spherical cargo by manipulating the motorized stage. (a) Motorized stage having two degrees of freedom (2-DoF) coupled with the electromagnetic coils. (b) Delicate positional control of seven 30 vol.% CNTY robots in order to transport the spherical cargo by manipulating the location of the motorized stage. Diameter and weight of the cargo was 4 mm and 52 mg, respectively.

The magnetic frequency was 13.3 Hz. Seven 30 vol.% CNTY robots were deployed.

Q3) Previous question 8: The authors demonstrate the transportation of a submerged cargo by underwater swimming of the CNTY robots, which is achieved by intentionally submerging the robots in water. Can you describe the process in more detail? For example, the principle of the lightweight robot sinking to the bottom of the water. How long does it take to submerge for the robot to completely sink? Can the sunken robot perform above-water swimming again? Will the CNTY robots sink to the bottom after swimming above the water for a period of time?

Response to comment

A3) First, the CNTY robots were grabbed by a tweezer and manually pressed to the bottom of the water container. Because density of the CNTY robots is higher than that of water, the robots could sink immediately and remained at the bottom of the water container. As a result, the CNTY robots could demonstrate transportation of submerged cargo through underwater swimming. Second, the CNTY robots could float on the water when the CNTY robots were relocated to the water surface by the tweezer. The CNTY robots remained afloat due to the surface tension of water. We confirmed that the CNTY robots swam above-water without submerging for a day. By pressing the robot to the bottom of the water container, the floating CNTY robot could be submerged again.

As we mentioned in Response to comment A1, the surface tension-assisted floating behavior can appear for high-density materials, for example, coins. Additionally, when the density and weight of robots are identical, contact area between the robot and water affects the buoyancy. Rod-like magnetic robots can float on the water more easily than spherical magnetic robots due to higher surface area of the rod-like magnetic robots. However, when an external force is applied, high-density materials can easily sink to the bottom of the water container. For high-density magnetic particles, actuation control is difficult on the water surface. Hence, lightweight body frame is required to prevent magnetic robots from sinking during the above-water swimming. We prepared lightweight nanoporous framework with a high aspect ratio by twisting low-density CNT fibers (2.6 g cm^{-3}). Although magnetic particles with a high density (7.9 g cm^{-3}) were coated on the robot surface, the CNTY robots achieved agile above-water swimmability. Furthermore, the above-water swimming modes of the robots were programmable on-demand due to the lightwightness of the CNTY robots.

1) Legend of Supplementary Fig. 30 in Supplementary Information (page 34)

Previous texts

a Above-water swimming of 19 robots

b Underwater swimming of 2 robots

Supplementary Fig. 30 Transportation of cargos with sliding resistance by above-water swimming and underwater swimming. Multiple 30 vol.% CNTY robots were manipulated by moving the electromagnetic coils coupled with the motorized stage having two degrees of freedom (2-DoF). (a) Transportation of semi-submerged cuboid cargo with a weight of 42 mg by above-water swimming of 19 CNTY robots. The increased number of CNTY robots resulted in a larger magnitude of vortex due to the larger hydrodynamic volume of the magnetic modular assembly. Therefore, the 19 CNTY robots with assembled rotational swimming transported cuboid cargo with sliding resistance. (b) Transportation of submerged asymmetric cargo with a weight of 39 mg by underwater swimming of two CNTY robots. The underwater swimming was proceeded after the robots were intentionally submerged in water. The magnetic frequency was 8.3 Hz in (a) and (b).

Revised texts

Supplementary Fig. 30 Transportation of cargos with sliding resistance by above-water swimming and underwater swimming. Multiple 30 vol.% CNTY robots were manipulated by moving the electromagnetic coils coupled with the motorized stage having two degrees of freedom (2-DoF). (a) Transportation of semi-submerged cuboid cargo with a weight of 42 mg by above-water swimming of 19 CNTY robots. The increased number of CNTY robots resulted in a larger magnitude of vortex due to the larger hydrodynamic volume of the magnetic modular assembly. Therefore, the 19 CNTY robots with assembled rotational swimming transported cuboid cargo with sliding resistance. (b) Transportation of submerged asymmetric cargo with a weight of 39 mg by underwater swimming of two CNTY robots. The underwater swimming was proceeded after the robots were intentionally submerged in

water. When the CNTY robots were pressed to the bottom of the water container, the robots sank immediately and remained submerged. When the sunken CNTY robots were relocated on the surface of water, they performed agile above-water swimming without submerging due to water's natural surface tension. The magnetic frequency was 8.3 Hz in (a) and (b).

Previous response to comment

Previous Q8-1) In the transportation of both floating microbeads and semi-submerged cargo, the position of the collective robots is changed by moving the water container. It seems that the general way to change the relative position of the rotational robots is moving the entire surrounding environment. However, in many scenarios, it is difficult to move the environments. How to solve this problem?

Previous A8-1) Thank you for the constructive comment. Although the relative position of the robots was manually controlled, it was previously intended to propose the proof-of-concept of cargo transport via collective swimming of multiple CNTY robots. However, we agree with the reviewer's point. For entirely contactless steering of the robots and improvement of transportation precision, we have devised a motorized stage with two degrees of freedom (2-DoF), coupled with the electromagnetic coils. We have addressed cargo transportation using the motorized stage in the response to comment (Q7) and newly added Fig. 6.

Previous Q8-2) Moreover, how can this robotic system be used for drug delivery?

Previous A8-2) A8) In abstract and outlook, drug delivery by the polymer-composited-based robots was mentioned for potential applications (Figure R2 and R3).

Figure R2. Demonstration of drug transport by the multilegged soft robot in a stomach model under wet environment. (H. Lu et al, *Nature Communications*, **9**, 3944, 2018)

Figure R3. Carrying the oblong pharmaceutical pill by rolling motion of the soft robot. (Y. Kim et al, *Nature*, **558**, 274–279, 2019)

The CNTY robots enabled to transport various cargos by above-water swimming as well as underwater swimming (newly added Supplementary Fig. 30 and Supplementary Movie. 9), showing the potential for transportation of pharmaceuticals. The underwater swimming was proceeded after the robots were intentionally submerged in water. According to the reviewer’s concerns, drug delivery can be described as referring to therapeutic agents (H. Lee et al., *ACS Applied Materials and Interfaces*, **13**, 19633–19647, 2021; T.-Y. Huang et al., *Advanced Materials*, **27**, 6644–6650, 2015). To avoid misunderstandings with respect to loading of the therapeutic agents, ‘drug delivery’ was changed to ‘transportation of pharmaceuticals’ as follows:

1) Supplementary Fig.30 in Supplementary Information (page 34)

Newly added Figure

a Above-water swimming of 19 robots

b Underwater swimming of 2 robots

Supplementary Fig. 30 Transportation of cargos with sliding resistance by above-water swimming and underwater swimming. Multiple 30 vol.% CNTY robots were manipulated by moving the electromagnetic coils coupled with the motorized stage having two degrees of freedom (2-DoF). (a) Transportation of semi-submerged cuboid cargo with a weight of 42 mg by above-water swimming of 19 CNTY robots. The increased number of CNTY robots resulted in a larger magnitude of vortex due to the larger hydrodynamic volume of the magnetic modular assembly. Therefore, the 19 CNTY robots with assembled rotational swimming transported cuboid cargo with sliding resistance. (b) Transportation of submerged asymmetric cargo with a weight of 39 mg by underwater swimming of two CNTY robots. The underwater swimming was proceeded after the robots were intentionally submerged in water. The magnetic frequency was 8.3 Hz in (a) and (b).

2) Abstract in manuscript (page 1)

Previous texts

The collective actuation of biomimetic nanocomposite robots is anticipated to provide practical robotic applications for microplastic removal^{39,40}, drug delivery⁴¹, and vortex control in microfluidic platforms⁴².

Revised texts

The controllable collective actuation of these biomimetic nanocomposite robots can lead to versatile robotic functions, including microplastic removal, microfluidic vortex control, and transportation of pharmaceuticals.

3) Outlook in manuscript (page 10)

Previous texts

The collective actuation of biomimetic nanocomposite robots is anticipated to provide practical robotic applications for microplastic removal^{39,40}, drug delivery⁴¹, and vortex control in microfluidic platforms⁴².

Revised texts

The collective actuation of biomimetic nanocomposite robots is anticipated to provide versatile robotic applications for microplastic removal^{45,46}, vortex control in microfluidic platforms⁴⁷, and transportation of pharmaceuticals^{48,49}.

Newly added references

48. Lu, H. *et al.* A bioinspired multilegged soft millirobot that functions in both dry and wet conditions. *Nat. Commun.* **9**, 3944 (2018).
49. Kim, Y., Yuk, H., Zhao, R., Chester, S. A. & Zhao, X. Printing ferromagnetic domains for untethered fast-transforming soft materials. *Nature* **558**, 274–279 (2018).

REVIEWERS' COMMENTS

Reviewer #1 (Remarks to the Author):

In the resubmitted letter, the authors have addressed all my questions in detail. The quality of this manuscript is further improved with corresponding revisions. I support the publication of this manuscript in Nature Communications.